# 20th century global glacier mass change: an ensemble-based model reconstruction

Jan-Hendrik Malles[1,2] and Ben Marzeion[1,2]

[1]Institute of Geography, Climate Lab, University of Bremen, Bremen, Germany
[2]MARUM - Center for Marine Environmental Sciences, University of Bremen, Bremen, Germany

**Correspondence:** J.-H. Malles (jmalles@uni-bremen.de)

**Abstract.** Negative glacier mass balances in most of Earth's glacierized regions contribute roughly one quarter to currently observed rates of sea-level rise, and have likely contributed an even larger fraction during the 20th century. The distant past and future of glaciers' mass balances, and hence their contribution to sea-level rise, can only be estimated using numerical models. Since independent of complexity, models always rely on some form of parameterizations and a choice of boundary conditions, a need for optimization arises. In this work, a model for computing monthly mass balances of glaciers on the global scale was forced with nine different data sets of near-surface air temperature and precipitation anomalies, as well as with their mean and median, leading to a total of eleven different forcing data sets. The goal is to better constrain the glaciers' 20th century sea-level budget contribution and its uncertainty. Therefore, five global parameters of the model's mass balance equations were varied systematically, within physically plausible ranges, for each forcing data set. We then identified optimal parameter combinations by cross-validating the model results against in-situ annual specific mass balance observations, using three criteria: model bias, temporal correlation, and the ratio between the observed and modeled temporal standard deviation of specific mass balances. These criteria were chosen in order not to trade lower error estimates by means of the root mean squared error (RMSE) for an unrealistic interannual variability. We find that the disagreement between the different optimized model setups (i.e., ensemble members) is often larger than the uncertainties obtained via the leave-one-glacier-out cross-validation, particularly in times and places where few or no validation data are available, such as the first half of the 20th century. We show that the reason for this is that in regions where mass balance observations are abundant, the meteorological data is also better constrained, such that the cross-validation procedure does only partly capture the uncertainty of the glacier model. For this reason, ensemble spread is introduced as an additional estimate of reconstruction uncertainty, increasing the total uncertainty compared to the model uncertainty merely obtained by the cross-validation. Our ensemble mean estimate indicates a sea-level contribution by global glaciers (outside of the ice sheets; including Greenland periphery, but excluding Antarctic periphery) for 1901 - 2018 of 69.2 $\pm$ 24.3 mm sea-level equivalent (SLE), or 0.59 $\pm$ 0.21 mm SLE yr$^{-1}$. While our estimates lie within the uncertainty range to most of the previously published global estimates, they agree less with those derived from GRACE data, which only cover the years 2002 - 2018 though.

# 1 Introduction

Glacier mass loss across most of the world constitutes a major part of the contemporary and projected 21st century sea-level rise (e.g., Slangen et al., 2017; Oppenheimer et al., 2019). Moreover, glaciers are important freshwater reservoirs for some regions of the world, and the vanishing of glaciers is thus likely to induce seasonal water scarcity in regions depending on those reservoirs (Cruz et al., 2007; Huss and Hock, 2018; Wijngaard et al., 2018; Kaser et al., 2010; Small and Nicholls, 2003).

Changes of a glacier's mass are often referred to in terms of *surface mass balance*: the difference between snow/ice accumulation and snow/ice loss (ablation) - mostly due to melting - over the glacier's surface. Dividing this value by the glacier's surface area yields the *specific mass balance*, which is an important variable in attempts to quantify glacier mass changes. Specific mass balances are a function of meteorological conditions at glacier locations and glacier-specific characteristics. The future evolution of the global glacier mass is usually estimated using numerical models (Hock et al., 2019; Marzeion et al.,

2020). This is the case for the last century or even more distant past as well (e.g., Goosse et al., 2018; Parkes and Goosse, 2020), since satellites able to observe the Earth's surface only became available well into the second half of the 20th century. Glaciers also lack comprehensive in-situ mass balance measurements, at least before 1950, since they are mostly situated in remote locations (see Figs. 2.6 and 2.7 in WGMS, 2020). It is therefore important to assess and improve glacier mass balance models used to reconstruct or project glacier mass evolution. An ensemble-based, long-term reconstruction can add to our un-

derstanding of the uncertainties in glacier modeling, which might in turn enhance our ability to make more robust projections of glacier mass change (Hock et al., 2019; Marzeion et al., 2020). Marzeion et al. (2020) have shown that ca. 25 % of global mass change uncertainty in 21st century projections of a glacier model ensemble can be attributed to differences in the output of climate models. About 50 % of the uncertainty in 2020, declining to ca. 25 % in 2100, was attributed to differences between individual glacier models. In this work we show that differences in meteorological reanalysis data add considerably to the

uncertainty of an individual glacier model's reconstruction as well. The modeling approaches to establishing global estimates for the glaciers' mass balances mostly make use of temperature-index melt models to represent the energy available for melting solid precipitation (i.e. snow) and ice (e.g., Huss and Hock, 2015; Radić and Hock, 2011; Hirabayashi et al., 2013). As a glacier's mass balance is interrelated with the glacier's geometric and hypsometric properties, some kind of length-area-volume scaling relation is often incorporated to account for changes in these properties in the models (Bahr et al., 2015) in order to

avoid the computational cost of modeling physical processes involved in glacier dynamics. This is especially relevant for an approach like ours, for which we need to run the model many times. The model used in the work presented here additionally includes a response time scaling to account for the glacier geometries' response lagging climatic forcing, but lacks an explicit representation of ice dynamic processes such as deformation, sliding, or calving/frontal ablation (Marzeion et al., 2012).

Although there are approaches based on solving the energy balance at the ice surface, these models usually either lack ice dynamics or geometric scaling (Shannon et al., 2019), can only be applied to a small number of glaciers and depend on upscaling to obtain global numbers (Giesen and Oerlemans, 2013), or do not perform significantly better than a similar model

without energy balance implementation (Huss and Hock, 2015). Another difficulty for models resolving the energy balance is the introduction of additional parameters that have to be constrained, which in turn adds complexity to the model optimization.

This indicates that implementing ice dynamics and the energy balance simultaneously is a difficult task and therefore not yet done routinely, but might still have the potential to enhance the accuracy of glacier modeling. That is because such models would have the ability to represent the physical mechanisms influencing a glacier in a more detailed, and thus possibly more realistic, fashion.

As mentioned above, for computational limitations, models solving the full equations of motion and thermodynamics individually for each glacier are generally not applied at the global scale. However, the Open Global Glacier Model (OGGM, Maussion et al., 2019) has been applied to compute ice velocity and thickness for each glacier based on a flowline approach, but is not yet able to routinely reconstruct glacier changes for such long time periods as in this work. The initialization of that flowline model for times prior to recorded glacier areas was also just recently explored (Eis et al., 2019, 2021).


  None of the models resolving the energy balance or explicitly calculating ice dynamics have been applied to globally reconstruct the glacier mass change on a century time scale. This implies that a comprehensive analysis determining which modeling approach might be most appropriate is not yet possible; at least not for all global glaciers and the whole 20th century. The need for a robust model evaluation, which can also be used to better understand the glacier model contribution to projection uncer-

tainty (Marzeion et al., 2020), is obvious.

  Uncertainties of numerical models are mainly caused by (i) uncertain boundary and initial conditions, (ii) approximations of the model's equations, and (iii) lack of knowledge about parameters involved in the model set-up (Hourdin et al., 2017). Therefore, optimization of parameters and/or input data is a standard procedure in glacier modeling (Huss and Hock, 2015;

Radić and Hock, 2011; Marzeion et al., 2012). Often, a single metric is chosen to be minimized (e.g., the model's RMSE with respect to observed in-situ mass balances). Rye et al. (2012) suggested multi-objective optimization for a (regional) glacier model, striving for 'Pareto optimality' (Marler and Arora, 2004), to constrain parameters more robustly.

  Models of (parts of) the Earth system are typically evaluated using observations and/or proxy data, usually with the ob-

jective to minimize the model's deviation from observations, e.g. by minimizing the RMSE (Gleckler et al., 2008; Taylor, 2001). Although in the case of glaciers, direct in-situ specific mass balance measurements are sparse and very heterogeneously distributed in space and time, they are essential in assessing the uncertainty, i.e. validation, of mass balance models. Nevertheless, other evaluation methods exist; for example using a combination of satellite gravimetry, altimetry, and glaciological measurement data (Huss and Hock, 2015). Such combined calibration data usually are not available for individual glaciers,

and/or do not have the temporal resolution required to assess the model's ability to capture interannual variability, making them impractical for our calibration and validation approach.

To avoid a confusion of the terms optimization, calibration, and validation, we briefly state our notions of these. Validation means calculating metrics that relate model outputs to observed values in a certain variable and that quantify the estimated model uncertainty. Optimization refers to choosing the best global parameter set with respect to the aims one sets regarding the validation. With calibration, we mean the deduction of glacier-specific model parameters from observational data.

Here, we apply a multi-objective optimization, concerning the five global parameters most relevant in the applied model, for each of nine meteorological forcing data sets (see Table 1), their mean and their median. The term global parameter here refers to parameters that are used in the model formulation (see section 2.2.1) and not varied for each glacier, but applied globally. Since the model is able to hindcast glacier evolution, the aim of this work is to (i) optimize the global model parameters in order to obtain model setups that reproduce in-situ mass balance observations as closely as possible, and (ii) to more robustly estimate model uncertainty, taking into account ensemble spread at times and in regions where observations are sparse. We use the model of Marzeion et al. (2012), but introduce changes to the mass balance calibration routine (see section 2.2.2). Additionally, we incorporate newer boundary and initial conditions as well as reference data, against which the model is validated. We show that the ensemble approach to the reconstruction produces more robust estimates of model uncertainty than taking into account results from a leave-one-glacier-out cross-validation (see section 2.2.1 and green box in Fig. 1) alone.

## 2 Data and Methods

In this section we introduce the data and the modeling as well as the optimization chain we applied in this work. In order to make the whole section more accessible to the reader, we point to the flowchart (Fig. 1), which illustrates how the individual steps described in the text are connected.

### 2.1 Data

#### 2.1.1 Meteorological Data

We conducted the search for an optimal parameter set for the version 4.03 of the CRU TS data (corresponding to an update of Marzeion et al., 2012) and additionally eight reanalysis data sets, as well as the mean and the median of all the data sets (see Table 1). The five of the eleven data sets not extending back to 1901 (see Table 1) were filled with CRU TS 4.03 data, exclusively for the purpose of initialization of glacier areas; the results are only shown (and evaluated) during time periods for which we have forcing data from the respective data set.

Anomalies of temperature and precipitation were calculated with respect to the 1961 to 1990 reference period used in CRU CL 2.0. For those data sets not covering the period 1961 to 1991, these anomalies were obtained by calculating the difference between the 1961 to 1990 and the 1981 - 2010 periods in the CRU TS 4.03 data set, and subsequently subtracting this value from the respective data set's 1981 - 2010 mean.

### 2.1.2 Glacier Data

The glacier model requires information about location, area, terminus and maximum elevation of each glacier at some point in time within the modeled time interval (1901 - 2018), which were taken from the most recent version of the Randolph Glacier Inventory (RGI v6.0, RGI (2017)). The RGI relies mostly on Landsat and other satellite imagery. Distinction of individual glaciers within glacier complexes was realized mostly by semi-automatic algorithms for detecting watershed divides. It includes Greenland's peripheral glacier with a high connectivity level 'CL2 (strongly connected)' (RGI, 2017), which we

exclude from the model results we present. Note that we neglect missing or disappeared glaciers that are not recorded in the RGI. This might lead to an underestimation of global mass changes, especially in the early 20th century (Parkes and Marzeion, 2018). The majority of recorded glacier areas date back to the years between 2000 and 2010, while there are a few early records between 1970 and 1980. The exact distribution is given in Fig. 2 of RGI (2017).

To be able to cross-validate the modeled annual specific mass balances, we use in-situ (glaciological) observations of glacier-wide annual specific glacier mass balances collected by the World Glacier Monitoring Service (WGMS, 2018). For the sake of simplicity and because observational errors of in-situ specific mass balance measurements are not always known, we ignore any uncertainties of these observations (Cogley, 2009) and treat them as the 'true' annual specific mass balance of a glacier in the recorded year. As stated in the introduction, our validation approach is based on a decomposition of the RMSE into the

three statistical measures temporal correlation, interannual variability ratio, and bias. Due to the lack of a comprehensive data set for geodetic measurements comparable to that of the WGMS for in-situ measurements, i.e. with the temporal and spatial resolution necessary for the calculation of the first two aforementioned metrics, it is unfortunately not yet possible to use those in the validation framework we established. Since the calculation of correlations and interannual variabilities requires a time series of data, we only take into account glaciers for which at least three years of in-situ mass balance were recorded. Those

are 299 glaciers with a total of 5977 annual specific mass balance measurements. Before 1950, only 110 annual records of 14 glaciers are contained in this data set. Of those 14 glaciers, 12 are situated in Central Europe and Scandinavia, and one in Alaska and Iceland each (WGMS, 2020).

## 2.2 The global glacier mass balance model

### 2.2.1 Basic equations and parameters

In this section, those features of the mass balance model that are relevant to the optimization procedure are described (see grey box in Fig. 1). A more thorough description is given in Marzeion et al. (2012).

The annual specific mass balance $B(t)$ of each glacier is computed as:

$$B(t) = \left[ \sum_{i=1}^{12} [P_i^{solid}(t) - \mu^* \cdot max(T_i^{terminus}(t) - T_{melt}, 0)] \right] - \beta^* \tag{1}$$

where $B$ is the annual modeled mass balance for an individual glacier in year $t$, $P_i^{solid}$ the amount of solid precipitation in month $i$, $\mu^*$ a glacier-specific temperature sensitivity parameter, $T_i^{terminus}$ the mean temperature in month $i$ at the glacier's terminus elevation, $T_{melt}$ a global threshold temperature for snow and ice melt at the glacier surface, and $\beta^*$ a calibration bias correction parameter. Terminus elevation temperature is calculated as:

$$T_i^{terminus}(t) = T_i^{CRUclim} + \gamma_{temp} \cdot (z_{terminus}(t) - z_{CRUclim}) + T_i^{anom}(t) \tag{2}$$

where $T_i^{CRUclim}$ is the climatological temperature in month $i$ taken from the grid point of the CRU CL 2.0 data set (New et al., 2002) closest to the respective glacier, $\gamma_{temp}$ is an empirically derived temperature lapse rate, $z_{terminus}$ is the elevation of the glacier's terminus, $z_{CRUclim}$ is the elevation of the grid point in the CRU CL 2.0 data set, and $T_i^{anom}(t)$ the monthly temperature anomaly deduced from the forcing data set. Values for $\mu^*$ and $\beta^*$ can theoretically be obtained by assuming an equilibrium state of the glacier in present-day geometry during a 31-year period centered around year $t^*$ when annual specific mass balance measurements are available for that glacier. In contrast to the initial publication of the model, we objectify the selection of $t^*$: while Marzeion et al. (2012) argue that $t^*$ is a function of the regional climatological history, it also depends on the glacier's response time scale, as discussed in Roe et al. (2021), for which there is no reason to assume spatial coherence. This means that we now do not spatially interpolate $t^*$ as in Marzeion et al. (2012), but introduce it as an additional global parameter. In the next section we elaborate further on this point.

The inference of the glacier-specific parameters ($\mu^*$ and $\beta^*$; see section 2.2.2) is assessed in a leave-one-glacier-out cross-validation procedure to determine the out-of-sample uncertainty, which should theoretically be done every time the model setup (i.e., parameter set and/or forcing data) is changed. Leave-one-glacier-out cross-validation means we run the model once for each validation glacier, which are those with at least three recorded annual specific mass balances, treating the respective glacier as if we did not have in-situ mass balance measurements available (see green box in Fig. 1). In other words, $\beta(t^*)$ is spatially interpolated in an inverse-distance weighted manner from the ten closest glaciers for the computation of annual specific mass balances of that glacier. The modeled annual specific mass balances of the individual validation glaciers obtained like this are then compared against the in-situ measurements in the optimization procedure (see section 2.3 and blue box in Fig. 1). Hence, we obtain an estimate of the model's uncertainty attached to the calibration procedure (see section 2.2.2). While values for $\mu^*$ can be computed for each individual glacier based on $t^*$, those for $\beta^*$ are spatially interpolated from the ten closest validation glaciers, using inverse-distance weighting. This will certainly work better in regions with high measurement densities and thus be a major part of our estimates' inaccuracy due to the previously mentioned heterogeneous distribution of in-situ mass balance measurements. Also, it is sensitive to errors or biases in the in-situ observations we use.

One global parameter ($T_{melt}$) was introduced in Eq. 1, but three additional ones are associated with the calculation of the monthly solid precipitation $P_i^{solid}(t)$:

$$P_i^{solid}(t) = (a \cdot P_i^{CRUclim} + P_i^{anom}(t)) \cdot (1 + \gamma_{precip} \cdot (z_{mean} - z_{CRUclim})) \cdot f_i^{solid}(t) \tag{3}$$

where $a$ is a precipitation correction factor, $P_i^{CRUclim}$ is the monthly climatological precipitation sum taken from the grid point of the CRU CL 2.0 data set closest to the respective glacier in month $i$, $P_i^{anom}(t)$ is the monthly total precipitation anomaly deduced from the forcing data set, $\gamma_{precip}$ is a global precipitation lapse rate, $z_{mean}$ is the mean elevation of the glacier, and $f_i^{solid}(t)$ is the fraction of solid precipitation:

$$f_i^{solid}(t) = \left\{ \begin{array}{l} 1 \text{ if } T_i^{terminus}(t) \leq T_{prec\ solid} \\[2mm] 0 \text{ if } T_i^{z_{max}} \geq T_{prec\ solid}, \text{ with } T_i^{z_{max}}(t) = T_i^{terminus}(t) + \gamma_{temp} \cdot (z_{max} - z_{terminus}(t)), \\[2mm] 1 + \frac{T_{prec\ solid} - T_i^{terminus}(t)}{\gamma_{temp} \cdot (z_{max} - z_{terminus}(t))} \text{ otherwise} \end{array} \right\} \tag{4}$$

where $T_{prec\ solid}$ is a global threshold temperature for solid precipitation, and $z_{max}$ the maximum glacier elevation. The amount of solid precipitation a glacier receives is hence estimated by applying an empirical negative temperature lapse rate, and a parameterized positive precipitation lapse rate. The assumption of increasing precipitation with elevation might not hold for some glaciers that are located on the downwind side of a mountain or for ones with very high maximum elevations, but this should be accounted for by treating it as a global parameter subject to optimization.

The four global parameters ($T_{melt}$, $a$, $\gamma_{precip}$, and $T_{prec\ solid}$) introduced in Eq. 1 - 4 are at the core of the model's mass balance computations and hence subject to the optimization presented here. Marzeion et al. (2012) used the CRU TS 3.0 data set to obtain $T_i^{anom}(t)$ and $P_i^{anom}(t)$. Here, we include additional meteorological data sets as well as their mean and median values in the optimization (see section 2.1.1).

The monthly mass balances are subsequently translated into volume, area and length changes by geometric scaling and relaxation (see grey box in Fig. 1). The geometric scalings by means of a power law, reviewed in Bahr et al. (2015), are currently the only option for estimating geometric changes from mass changes without having to resolve actual physical processes as in a flowline or higher order model. From the theory discussed in Bahr et al. (2015) it follows that the exponent in these power law scalings is a constant and the scale factor is a random variable. In the model, we applied the constant exponents for mountain/valley glaciers and ice caps given from that theory and scale factors empirically derived in Bahr (1997) and Bahr et al. (1997). Since there are uncertainties attached to the scale factor, we estimate a 40 % error in the volume-area and a 100 % error in the volume-length scaling for the model's error propagation, as in Marzeion et al. (2012). Theoretically, the scale factors could be treated as global parameters as well, but it is not clear whether an optimization of those would benefit the overall (global) model accuracy, while it would increase the efforts in computation and evaluation. Concerning the relaxation, a response time scale of the volume-length change is estimated by assuming that smaller glaciers and those with higher mass turnover will react faster to volume changes (details in Marzeion et al., 2012).

Initial values for the area of each individual glacier at the start of the model run (e.g., beginning of the 20th century) are found using an iterative approach that minimizes the difference in area between modeled glacier and the RGI record in the year of the respective observation (see grey box in Fig. 1). If this iterative procedure is not successful, the glacier is not included in the reconstruction. For these glaciers, a simple upscaling is applied in the computation of regional and global results. The optimized CRU TS 4.03 model setup was able to initialize glaciers accounting for 98 % of the glacier area recorded in the RGI. This value is roughly the same for the optimized model setups that performed well according to our validation procedure, although it is slightly lower for those forced with the mean/median of the meteorological data ensemble (see Table 2). A failure of the initialization for an individual glacier might occur when, for example, the calibration (see section 2.2.2 and grey box in Fig. 1) results in a very high temperature sensitivity for that glacier. The iterative search of an initial area might then not be able to capture the very large starting area necessary for the implicated strong mass change. The largest fractions of area not successfully modeled with the optimized CRU TS 4.03 model setup are located in the Greenland periphery (ca. 9 %) and Russian Arctic (ca. 5 %).

Note that since the CRU CL 2.0 data set used to obtain $P_i^{CRUclim}$ and $T_i^{CRUclim}$ does not cover Antarctica, we do not consider glaciers in the periphery of Antarctica and subantarctic glaciers here (labeled region 19 in RGI, 2017).

### 2.2.2 Mass balance calibration

As explained above, we treat the parameter $t^*$ as a global one, opposed to a glacier-specific estimation in Marzeion et al. (2012). In order to illustrate the reasoning, we need to discuss the mass balance calibration for an individual glacier in the model in more detail (see grey box in Fig. 1). The calibration is based on the idea of inferring a glacier's temperature sensitivity $\mu^*$ by finding a climatological time period in the forcing data set (centered around $t^*$) which would result in a zero specific annual mass balance of the glacier in present-day geometry. Thus, for each center year $\tilde{t}$ of a 31-year period, we can calculate $\mu(\tilde{t})$ by requiring:

$$B(\tilde{t}) = \sum_{i=1}^{12} [P_{i,clim}^{solid}(\tilde{t}) - \mu(\tilde{t}) \cdot max(T_{i,clim}^{terminus}(\tilde{t}) - T_{melt}, 0)] = 0 \tag{5}$$

where $P_{i,clim}^{solid}(\tilde{t})$, and $T_{i,clim}^{terminus}(\tilde{t})$ are climatological averages of $P_i^{solid}(\tilde{t})$ and $T_i^{terminus}(\tilde{t})$. Note that the calculation is based on a smaller number of years when $\tilde{t} < 1916$ or $\tilde{t} > 2003$. For each of the 299 glaciers that have at least three years of in-situ mass balance observations, we calculate the modeled annual specific mass balance (based on Eq. 1) for each $\tilde{t}$. Then, the associated calibration bias of an individual validation glacier is calculated as

$$\overline{B_M} - \overline{B_o} = \beta(\tilde{t}) \tag{6}$$

where $\overline{B_M}$ is the mean modeled specific annual mass balance of the validation glacier, with $\mu^*$ equal to $\mu(\tilde{t})$ in equation 1, for the years of available mass balance measurements, and $\overline{B_o}$ the mean observed mass balance. Hence, a negative (positive)

$\beta(\tilde{t})$ means that the glacier with its present-day geometry would have presumably gained (lost) mass during the climate period around $\tilde{t}$, applying the inferred $\mu(\tilde{t})$. Accordingly, the glacier would be too (in)sensitive to changes in the forcing and the appli-
250 cation of $\beta(\tilde{t})$ is required to balance this in Eq. 1. The general problem here is to infer $t^*$ for glaciers without available annual in-situ mass balance measurements. Marzeion et al. (2012) chose $t^*$ to be that $\tilde{t}$, for which $|\beta(\tilde{t})|$ was minimal; $\mu^*$ was then calculated from equation 5 applied to $t^*$, and $\beta^*$ taken as $\beta(t^*)$. For glaciers without in-situ observations of mass balances, $t^*$ and $\beta^*$ were interpolated from the ten closest glaciers with observations, using an inverse-distance weighting. Using this method, Marzeion et al. (2012) were able to identify a suitable parameter set in the leave-one-glacier-out cross-validation pro-
255 cedure that did not show a large bias against in-situ measurements, applying CRU TS 3.0 as atmospheric boundary conditions, a previous version of the RGI, and other mass balance validation data. However, this is not generally the case for the data sets applied here, and there is a conceptual shortcoming in the spatial interpolation of $t^*$, which we will illustrate for one exemplary model setup.

Part (a) of Fig. 2 shows the global average of $\beta(\tilde{t})$, weighted by the length of each glacier's in-situ mass balance measurement time series (henceforth, all mentioned averages over different validation glaciers imply such a weighting), using CRU TS 4.03 as boundary condition, applying the optimal parameter set (see section 2.3).

    Figure 2b shows that the distribution of $t^*$ estimated directly is bi-modal, with frequent values either at the beginning or
265 end of the considered period, but the spatial interpolation leads to a more even distribution. This in turn means that, generally speaking, the spatial interpolation moves $t^*$ towards the middle of the considered time period, thereby increasing the value of $\beta^*$ for glaciers with an early $t^*$, and decreasing it for those with a late $t^*$. Figure 2b also shows that there are more valida-tion glaciers with $t^*$ at the beginning of the 20th century than at the end of the 20th century or the beginning of the 21st century.

Furthermore, those glaciers with a directly estimated individual $t^*$ at the beginning of the 20th century tend to have a pos-itive $\beta^*$, implying that even with present-day geometry, those glaciers would have lost mass under climatic conditions of the early 20th century applying $\mu(t^*)$. The zero-crossing of the global average $\beta(\tilde{t})$ is thus found at a period when positively and negatively biased glaciers cancel each other. Since moving the median of $t^*$ towards the middle of the of the modeled period generally goes along with an increase of the globally averaged calibration bias $\overline{\beta(t^*)}$, using the spatial interpolation of $t^*$ tends
to lead to a positively biased model setup, which then becomes apparent in the leave-one-glacier-out cross-validation. That is because there are more validation glaciers with an early than late individual $t^*$, as stated above.

    In order to avoid this effect, and taking into account that neighboring glaciers will have different response times, such that even if they experience a very similar evolution of climate anomalies we cannot expect a close spatial coherence of $t^*$, we do
not spatially interpolate $t^*$ as was done by Marzeion et al. (2012), but treat it as a fifth global parameter instead. Note that $\mu^*$ is still a glacier-specific parameter following Eq. 5, and that $\beta(t^*)$ is still interpolated from the ten closest glaciers in an in-verse distance weighted manner. Although retaining the interpolation of $\beta(t^*)$ seems to contradict the argument about regional

climatology made above, it is the only way to handle the calibration bias for glaciers without validation data, and we expect biases caused by this approach on the scale of an individual glacier to cancel out globally. Ultimately, the leave-one-glacier-out cross-validation will reveal any potential new model errors introduced through this change.

At this point we recapitulate the reasoning behind our changes to the calibration procedure compared to previous studies that applied the same model, since it is an important point of this work: In contrast to Marzeion et al. (2012), we do not rely on the assumption of a steady state for every single glacier using present-day geometry and climate conditions during a glacier-specific period around $t^*$, but rather on a global-mean steady state of glaciers in their present-day geometry with climate conditions during a (globally equal) period around $t^*$. This means that while some glaciers with present-day geometry would gain mass when exposed to the climate around the global $t^*$, others would lose mass. Figure 2a shows the mean bias of glaciers for which glacier-specific values of $t^*$ can directly be obtained based on in-situ observations; once for glaciers with $t^*$ before 1920 and, and once with $t^*$ after 1998. It can be seen that the (global) calibration bias ($\beta(\tilde{t})$) is a function of the center year of the climatology ($\tilde{t}$) we assume glaciers (with present-day geometry) to be in equilibrium with globally. For glaciers with an early individual $t^*$, the calibration bias will be increasingly positive as we depart from the climatology of its $t^*$ to warmer climate periods (i.e., later $\tilde{t}$). This is because $\mu$ will be underestimated using a warmer climate for calibration (see Eq. 5). If there are glaciers with early and late individual $t^*$ in close proximity of each other, the $\beta(\tilde{t})$ we interpolate to a glacier with early individual $t^*$ will often be too low, while it will be too high on glaciers with a late individual $t^*$. Because there are more glaciers with an early individual $t^*$ (before 1920) than a late $t^*$ (after 1998, see Fig. 2b), this then results in an overall positive bias in the global cross-validation result. Interpolating $t^*$ has a similar effect. Overall, the cross-validation shows that this method is able to yield unbiased model setups (see section 3.1.1), disregarding possible biases in the validation data.

## 2.3 Parameter optimization strategy

For the identification of an optimal parameter set, we applied a 'brute-force' approach (see blue box in Fig. 1). This means that we varied each parameter other than $t^*$ (see above) using the following ranges, which are similar to those used in Marzeion et al. (2012):

- threshold temperature for snow/ice melt ($T_{melt}$) [°C]: {-2, -1, 0, 1, 2}
- threshold temperature for solid phase precipitation ($T_{prec.\ solid}$) [°C]: {-1, 0, 1, 2, 3, 4}
- precipitation lapse rate ($\gamma_{precip}$) [%/100 m]: {0, 1, 2, 3, 4, 5}
- precipitation correction factor ($a$): {1, 1.5, 2, 2.5, 3}

We did so for each meteorological data set and performed a leave-one-glacier-out cross-validation for each of the 900 parameter combinations possible within these ranges. This resulted in 9900 validation runs (900 times eleven forcing data sets), which we used to identify parameter sets that yield a zero-crossing of the global average $\beta(\tilde{t})$ in a first step. For all forcing data sets except 20CRV3, those zero-crossings were found with $\tilde{t} < 1920$ (applying 20CRV3, some were found in 1962 and 1976).

We then performed additional cross-validations with the twenty best-performing parameter sets yielding a zero-crossing of the global average $\beta(\tilde{t})$ to fine-tune $t^*$, applying the range 1901 to 1920, except for 20CRV3 where we applied the ranges 1909 - 1918, 1960 - 1964, and 1974 - 1978. Hence, we performed 400 (4400) additional cross-validation runs per data set (in total).

From those cross-validations, three characteristic statistical measures of model performance were computed: model bias (i.e., mean model error) with respect to observations, the temporal correlation with observations, and the ratio of standard deviations of interannual variability between modeled and observed mass balances. We do not include the mean squared error (MSE) as a performance measure, since it is simply a (weighted) combination of the three performance measures:

$$325 \quad MSE = \sigma_M^2 + \sigma_o^2 - 2\sigma_M\sigma_o R + (\overline{M} - \overline{O})^2 \qquad (7)$$

where $\sigma_M$ is the standard deviation of modeled mass balances, $\sigma_o$ the standard deviation of observed mass balances, R the Pearson correlation coefficient, $\overline{M}$ the mean of modeled mass balances, and $\overline{O}$ the mean of observed mass balances (thus, the last term corresponds to the squared bias).

From Eq. 7 it can be inferred that a minimum MSE occurs for a model setup in which the standard deviation ratio equals the correlation coefficient. Hence, in a model setup that is not perfectly (positively) correlated with the observations (i.e., $0 < R < 1$), a more realistic standard deviation ratio (e.g., $1 \geq \frac{\sigma_M}{\sigma_o} > R$) will result in a higher MSE. However, a correlation coefficient equal to one is generally not achievable in models such as the one used in this work. Consequently, minimizing the MSE will lead to preference of parameter sets that underestimate variance. This is problematic, since a correct representation of variance is indicative of correct model sensitivity to changes in the forcing. For example, it is possible to imagine to apply a model setup that yields a low bias and good correlation, but largely underestimates the interannual variation of mass balances. It is therefore beneficial to not only minimize the MSE, but rather to minimize the three statistical coefficients it comprises individually, in order to not trade a realistic model sensitivity for a smaller MSE.

All three performance measures were calculated for each validated glacier, and then averaged over all these glaciers, weighted by the number of available mass balance observations per glacier. This was done for every cross-validation run in order to be able to identify the overall best performing model setups.

Standard deviation ratios were brought to represent the deviation from an optimum value (i.e. one) by:

$$345 \quad SR = \frac{\sigma_M}{\sigma_o} - 1 \qquad (8)$$

To determine for each meteorological data set a model parameter set that, on average, shows the highest skill to represent the behavior of observed glaciers, we normalize the performance measures introduced above such that the individual scores $s$ range from 0 for the worst to 1 for the best validation result by the following equations:

$$s_{i,bias} = \frac{max(|bias|) - |bias_i|}{max(|bias|) - min(|bias|)}, s_{i,SR} = \frac{max(|SR|) - |SR_i|}{max(|SR|) - min(|SR|)}, s_{i,R} = \frac{R_i - min(R)}{max(R) - min(R)} \qquad (9)$$

where i is the individual model setup the score is calculated for. These scores were then added up to identify the 'optimal' model setup as the one with the maximum overall score. If a model setup obtained the single best result for all three performance measures individually, it would thus yield a score of three. Note that the three (or potentially other) performance measures might be weighted differently, based on the objective of the model application. However, as shown below, we do not find substantial trade-offs between the three performance measures, such that any potential weighting would have a very limited influence on the results.

## 3 Results

### 3.1 Cross-validation and uncertainty assessment

#### 3.1.1 Performance measures

Table 2 shows the values obtained for performance measures and optimal global parameters. We differentiate between the mean and median of the forcing data input used as individual boundary conditions (mean/median input) and the mean and median of the ensemble output values (mean/median output). For more than half of the optimized model setups, the global mean bias of the optimal parameter set is smaller than 10 mm w.e. yr$^{-1}$, and the correlation is larger than 0.6, while the amplitude of the interannual variability is estimated correctly within a small range (ca. 5 %). RMSEs lie roughly between 700 and 800 mm w.e. yr$^{-1}$ for most optimized model setups. Only 20CRV3 shows a significantly higher RMSE, caused by some large outliers. Note that the number of glaciers that cannot be initialized also depends on the meteorological data set used as boundary condition. CERA20C, e.g., not only performs the worst (obtaining an overall score of 1.38 using the optimal parameter set), but leads to only 274 of 299 validation glaciers being initialized in the cross-validation, and 180,799 of the 212,795 glaciers in the global reconstruction run, representing 84 % of today's global glacier area outside Antarctica. In contrast, the best performing model setup that covers the whole model period (CRU TS 4.03) is able to initialize 298 validation glaciers and 201,004 glaciers in the global reconstruction run, representing ca. 98 % of the global glacier area. Following our scoring system, we find that the statistically best performing single data set covering the whole model period is CRU TS 4.03, and the overall best performing data set, but only covering 1979 - 2018, is ERA5. Our best estimate for the whole model period is the mean model output.

Independent of the time period considered, the mean output of the ensemble shows the best performance, exceeding not only the best individual ensemble member, but also the result obtained by the mean and median input. The statistically best-performing individual ensemble members vary with the time periods that are covered by the meteorological data sets. For example, during the period 1958 - 2018, JRA55 leads to the best performance; from 1979 onward, it is ERA5. Table 2 also shows that the performance measures attain better values if the averages are weighted by the length of the observation time series than with the non-weighted average, illustrating the need for long-term observations for reliable model validation.

In order to assess the consistency of cross-validation results among the ensemble members, two-sample Kolmogorov-Smirnov-Tests for the similarity of distributions were conducted for all 55 possible unique pairs of the eleven optimized model setups. This was done for modeled annual specific mass balance and model deviation distributions. Model deviation here refers to the differences between each modeled and observed annual specific mass balance value in the cross-validation procedure; its average thus corresponds to the average of the bias weighted by the number of available mass balance observations per validation glacier. The confidence level we require for rejecting the similarity of distributions is at 95 %. Regarding the distributions of modeled mass balances, only 10 (18 %) of the tested pairs are not significantly different; all involving the six best-scored model setups (see Table 2). Model deviation distribution pairs do not significantly differ in 27 (49 %) cases, of which only 1 (2 %) involved 20CRV3, CERA20C, or ERA20C. We conducted Welch's t-test for the similarity of means in the same manner. Here, only the three lowest-scored model setups' means of modeled mass balances are significantly distinguishable from other ensemble members. Concerning the mean model deviation, only that of CERA20C significantly differs from the others. Hence, the similarity tests indicate that the results of model setups with higher scores (see Table 2) tend to be more consistent among each other and to differ from lower scored ones statistically. Model deviation distributions significantly different from those of other ensemble members are to a large degree produced by low-scored model setups, while the mean is only significantly different for CERA20C. The significantly high bias and low score of CERA20C indicate particular issues with this forcing data set and lead us to exclude it from the following ensemble calculations. In the subsequent section we will explore where these issues stem from. In doing so, we try to explain why the temporal and spatial constraints of the validation data hinder us to make assertions over which individual model setup is the most reliable one over the whole temporal and spatial model domain.

### 3.1.2 Spread of the ensemble inconsistent with model uncertainty from cross-validation

The leave-one-glacier-out cross-validation procedure applied here is designed to estimate the uncertainty of model results for glaciers that have no in-situ mass balance observations, and for times where there are no in-situ observations. Therefore, in principle, the results of the individual ensemble members should agree within their corresponding uncertainty estimates. However, there is a strong spatial bias in in-situ mass balance observations towards certain RGI regions; mostly locations where also the past state of the atmosphere is well constrained, since both atmospheric and glaciological observations are denser in easily accessible regions. The majority of glaciers, though, is situated in remote locations where observations of the state of the atmosphere were very sparse, particularly in the first half of the 20th century. Thus, the cross-validation is biased towards times and places where the state of the atmosphere, i.e., the boundary conditions of the glacier model, can be assumed to be exceptionally well constrained.

Figure 3 shows that 66% of the validation data originate from only four RGI regions: Western Canada and USA, Scandinavia, Central Europe, and Central Asia. Panel (b) shows the fraction of mean annual ensemble variance of global mass change rates in the modeled period attributable to each RGI region. Most of the ensemble spread is due to disagreement in sparsely observed regions that contain much glacier ice. Of the mean annual global ensemble spread, nearly 60 % can be attributed to the disagreement in estimates for the regions Alaska, Arctic Canada North, and Greenland periphery. The value for the Green-

415 land periphery increases from 21 to 36 % if we included CERA20C in the calculation. This indicates that peripheral glaciers in Greenland are responsible for a considerable amount of the ensemble spread as well as for most of the large divergence of CERA20C from the other ensemble members. The only region that shows a large spread among ensemble members but does not contain as much glacier ice as the previously mentioned ones, are the Southern Andes.

In Fig. 4a, the issue of temporally biased validation data (all are from the second half of the 20th century or the beginning of the 21st) can be recognized. Mean mass loss rates calculated with forcing data sets that have complete data coverage over the whole model period for the four previously mentioned well-observed regions are shown. Comparing results for the four best-observed regions to global results (Fig. 4b), it can be seen that the disagreement on the global scale is larger than in the well-observed regions, and that the global reconstruction forced by CERA20C is far off the three other ensemble members 425 while it is not so in the well-observed regions. This behavior can be explained by the much more pronounced warming of glacier locations at the global scale in CERA20C until ca. 1960 (Fig. 4d): during the calibration, lower temperatures at $t^*$ will lead to higher temperature sensitivities (see Eq. 5). Similarly, the greater increase of temperature will result in higher mass loss rates.

    Concerning these issues with CERA20C, it is striking that in spite of its large positive specific mass balance bias in the 430 cross-validation, global mass change estimates obtained with it are much larger than those of the other ensemble members. This underlines the fact that even though the cross-validation is crucial in the optimization process, we cannot entirely rely on it for assessing global and long-term reconstruction performance of individual model setups. Therefore, and because, as stated in the previous section, the best-performing data sets do produce statistically quite similar results for the validation glaciers, we will only use estimates based on the ensemble – i.e., not individual members – in the following. We exclude the results 435 of model runs forced with the mean and median input from our ensemble calculations in order not to bias them towards the central value and also because they contain the problematic values of the CERA20C data. If a full ensemble approach as done in this study is not feasible (e.g., due to computational constraints) we still recommend to use a mean/median input data set as the meteorological forcing for reconstructions outside the spatial and temporal domain of validation data, since a single best-performing data set can not be identified conclusively.

    In both the well-observed regions (Fig. 4a) and the global scale (Fig. 4b), the different model setups disagree stronger in the first half of the 20th century, reflecting that uncertainty in the atmospheric conditions during that time is also greater. All in all, we find that the ensemble spread tends to be larger than uncertainty estimates obtained via the cross-validation, and that this is caused by the majority of glacier observations coming from places and times where the uncertainty of the state 445 of the atmosphere is smaller than what can typically be expected in glacierized regions. Furthermore, we assume that the individual glaciers' error estimates are uncorrelated with each other and random, i.e. independent, as we do not have direct model error estimates for every glacier and can thus not account for correlations of individual glaciers' errors. However, the ensemble approach allows to explore if, and to which degree, the cross-validation underestimates the true uncertainty of the reconstruction.

### 3.1.3 Combining model and ensemble uncertainty

To account for both the model error, as calculated in the cross-validation procedure, and the ensemble spread, the total uncertainty of the ensemble average is calculated as follows. First, we calculate the model error of the ensemble average solely determined by the means of the RMSEs obtained from the leave-one-glacier out cross-validation:

$$\epsilon_{model}(t) = \frac{\sqrt{\sum\limits_{i}^{n} \epsilon_i(t)^2}}{n} \tag{10}$$

where $\epsilon_i(t)$ is the model uncertainty computed in the cross-validation procedure for an individual ensemble member $i$ for year $t$. Then we add the ensemble spread as a further uncertainty measure to the model error of the ensemble average:

$$\epsilon_{ensemble}(t) = \sqrt{\epsilon_{model}(t)^2 + \sigma(t)^2} \tag{11}$$

where $\sigma(t)$ is the ensemble standard deviation in year $t$. Here, we treat the individual model setups' errors, obtained from the cross-validation procedure, to be independent from each other and the model error of the ensemble average to be independent from the ensemble spread. This might lead to an underestimation of total uncertainty, since there might be correlations of the individual sources of uncertainty for which we cannot account. Because we model more than 200,000 glaciers and assume their errors to be independent as well, thereby assuming that their errors partly cancel each other out, the true uncertainty is probably higher than our estimate. However, we do account for interannual covariances of the ensemble when estimating the uncertainty of mean values over periods longer than one year.

Figure 5 shows the temporal evolution of total uncertainty ($\epsilon_{ensemble}$) as well as the aggregated model uncertainty ($\epsilon_{model}$) and ensemble spread ($\sigma_{ensemble}$) of the ensemble mean mass change rate estimate. The total uncertainty of the ensemble mean estimate is high in earlier years, with a sharp decrease after the first twenty years. This is due to the decrease in the high model error of the ensemble average, especially during the first decade of the 20th century, which is produced by very high mass losses of a few glaciers in some model setups during that period. The ensemble spread is also greater during the first half of the 20th century compared to later years, which can be attributed to less agreement between meteorological data sets in earlier years. Note that the further back in time we go, the fewer meteorological data sets are available, since not all reanalysis products provide data for the whole period.

### 3.2 Global glacier mass change

In the following we present our modeled estimates of global glacier mass change. Note that we express those in units of sea-level contribution. This means positive values indicate a contribution to sea-level rise and thus a mass loss of the glaciers. Figure 6 shows the temporally accumulated mass change estimates, relative to 1980 (the year from which onward all meteorological data sets have data coverage), and their uncertainties. The part (a) shows the estimates for each individual ensemble

member as well as their model uncertainties $\epsilon_{model}$. Especially in the first half of the 20th century, ensemble members diverge, with CRU TS 4.03 showing the lowest and ERA20C, next to CERA20C, the highest mass loss during that period. The ensemble average mass change estimate over the whole model period is $69.2 \pm 24.3$ mm SLE, which translates to an average mass change rate of $0.59 \pm 0.21$ mm SLE yr$^{-1}$, and a mass loss of roughly 18 % relative to 1901.

Table 3 displays the regional and global mass loss rates for different reference periods. Mass change rates estimates for more recent periods are increasing across most regions, reaching $0.90 \pm 0.12$ mm SLE yr$^{-1}$ accumulated globally in the most recent period (2006 - 2018). The only time and region for which an increase in glacier mass is estimated are the Southern Andes during 1901 - 1990, although with a relatively high uncertainty due to ensemble spread (see Fig. 3).

To explore the period of decelerated mass loss between roughly 1940 and 1980 visible in Fig. 7, the periods 1901 to 1940 and 1941 to 1980 are shown in Table 3. For most regions, the mass change rate estimates are substantially smaller in the latter period; only New Zealand exhibits a significantly larger mass loss. Regarding the global estimate, most of the mass loss deceleration took place in the Greenland periphery and the North American continent (i.e., RGI regions 1 to 5). Thus, after increasing mass loss rates until around 1930 (see Fig. 7), glaciers started to lose less mass until around 1980, possibly caused by atmospheric cooling induced by increasing aerosol concentrations (Ohmura, 2006; Ohmura et al., 2007; Wild, 2012). From then on, the glaciers' contribution to sea-level rise accelerated again until the end of the modeled period (2018). Figure 8 shows the drivers of this behavior: the global ensemble mean temperature (lower panel) and precipitation anomalies as well as total amount of solid precipitation (upper panel; see Eq. 3 and 4; all weighted by glacier area). From ca. 1980 on, heat available for ice and snow melt, i.e. the temperature anomaly, increased monotonously. While precipitation at the glacier locations tended to increase over time, the amount of solid precipitation at glacier locations decreases from roughly 1980 on, implying that not only ablation increased, but also accumulation decreased. In contrast to that, the increase in total precipitation between ca. 1930 to 1950 was accompanied by a similar increase in solid precipitation, indicating that the warm anomaly at the same period was too weak to reduce accumulation as much. In order to get an impression of the relative importance of precipitation and temperature anomalies, we ran the model with the optimized median input model setup; once holding total precipitation constant at the climatology around $t^*$, and once holding temperature constant in the same way (Fig. A1). Initialization and calibration were done with the full forcing in order to enable meaningful comparisons with the regular model run. Holding the temperature constant resulted in a 65 % lower mass loss, while the constant lower precipitation increased mass loss by only 5 %. This indicates that the temperature increase plays a much larger role for glacier mass change than increased precipitation. This is expected, because increased temperatures do, as mentioned above, not only increase the melt of a glacier's ice mass, but also decrease the amount of solid precipitation it receives. The finding that precipitation changes play a minor role in glacier mass change is consistent with the literature (Van de Wal and Wild, 2001; Leclercq et al., 2011). Moreover, the interannual variability is larger with varying temperatures compared to only varying precipitation (see Fig. A1).

Our results also indicate that the glaciers' retreat to higher altitudes acted as a negative feedback on mass loss in more recent times. This is based on the observation that although the global average temperature anomalies at glacier locations were considerably higher after 2000 than around 1930, and the amount of solid precipitation was lower, the global glacier sea-level contribution rates are not higher according to our model results (see Figs. 7 and 8). However, this result might have been influenced by the applied scaling and relaxation laws (see section 2.2.1), as they control the geometric response to mass changes in our model. Comparing our results of glacier geometry to a publication that estimated contemporary global glacier volumes (Farinotti et al., 2019), though on the basis of modeling results as well, we find that our global volume estimate differs less than 1 % from their result. Another feedback that certainly plays a role here, but cannot be resolved by our model, is the positive mass balance-surface elevation feedback: as a glacier's surface elevation decreases due to mass loss, it experiences higher temperatures, because of the atmospheric temperature lapse rate. This in turn enhances the initial mass loss (Harrison et al., 2001). Since our model is calibrated and validated with data from more recent years, it could be argued that mass change was actually lower in earlier years due to higher surface elevations of glaciers (Huss et al., 2012).

Concerning uncertainty estimates, Table 3 shows that most of the uncertainty stems from the regions Alaska, Arctic Canada (North), and the Greenland periphery in more recent periods (e.g., 1993 - 2018). In the earliest period (1901 - 1940), the Russian Arctic region exhibits a high uncertainty as well, indicating that the large model error in the early 20th century (see Fig. 5) is produced there alongside regions 1 to 5.

## 4   Discussion

Table 4 shows our global results compared to previously published estimates for mass change rates over certain periods. We mostly find good agreement within the respective uncertainty ranges. For the periods 2003 - 2009, 2002 - 2016, and 1992 - 2016 there is a significant disagreement between literature values and our model results. The disagreeing values for 2002 - 2016 from Wouters et al. (2019) were solely derived from gravimetry (GRACE) data. Estimates for 2003 - 2009 from Gardner et al. (2013) and for 1992 - 2016 from Bamber et al. (2018) involve GRACE data as well. Interestingly, we find that when we compare the five pentads Bamber et al. (2018) studied during 1992 - 2016 individually to our estimates, those for the first three pentads (when the GRACE mission had not yet started) agree within uncertainty ranges. Another work based on GRACE data (Jacob et al., 2012) estimated a mass loss of glaciers outside of Antarctica and Greenland for the period 2003 - 2010 of $0.41 \pm 0.08$, while our estimate for that period lies at $0.66 \pm 0.08$ and that of Gardner et al. (2013, for 2003 - 2009) at $0.59 \pm 0.07$. Part of these disagreements might be explained by the storage of meltwater for example in glacial lakes (Shugar et al., 2020), which because of the close proximity to the glaciers cannot be separated from the ice mass in gravimetry data. GRACE will therefore observe lower mass change values than in-situ or geodetic observations. Since these lower values might be closer to the glaciers' actual contribution to sea-level rise, the issue points to the larger problem of distinguishing between glacier mass change, and the corresponding sea-level change, which are not exactly equal. However, Shugar et al. (2020) also point out that glacial lake storage accounts for only about 1 % of glacier melt volume (excluding Greenland and Antarctica), which indicates

that this process is of limited relevance. Other hydrological processes like groundwater flow or human activities (e.g., building of reservoirs) might still induce discrepancies between gravimetric and in-situ/geodetic measurements. Another potential source of underestimating glacier mass loss by using gravimetry data in regions with many marine-terminating glaciers is the presence of discharged icebergs close to the glaciers that GRACE is presumably not able to separate from the actual glacier mass.

Gardner et al. (2013) point to discrepancies between satellite-derived and in-situ estimates of glacier mass changes, suggesting a negative bias in in-situ observations for regions where the density of those measurements is low. They hence only relied on in-situ observations in regions where those have a high density. Zemp et al. (2019) addressed this issue as well by combining glaciological and geodetic measurements. Although our model is calibrated solely using in-situ observations, its estimates are still close to Zemp et al. (2019), in which the uncertainty for some periods is admittedly large (Table 4). Comparing our results to Zemp et al. (2019) in periods where we included the Greenland periphery (2003 - 2009 and 1961 - 2010) we see a slightly lower agreement. Our estimates' uncertainty range also overlaps more with the one of Gardner et al. (2013) for 2003 - 2009 if we exclude that region. This indicates that our mass change estimates for the Greenland periphery might be too large in these time periods. We also included estimates from WGMS (2015) in Table 4, which are merely arithmetic averages of the available in-situ (glaciological) and geodetic mass balance measurements to show that our estimates, although solely calibrated and validated with in-situ measurements, are lower than those and closer to more thorough analyses of mass change data. Our estimates lying close to those of Marzeion et al. (2012) or Marzeion et al. (2015) can not be explained merely by the fact that the same model was used. One reason for this is that we used newer and more validation data and a newer RGI version. Another one is the change in calibration strategy that we applied in this work (see section 2.2.2). Furthermore, driven with other meteorological data, the Marzeion et al. (2015) mass loss estimates for 2003 to 2009 lie lower than ours when using the same model but a different calibration procedure, while those of Marzeion et al. (2012) lie higher (see Table 4). This underlines the influence of boundary and initial conditions on modeling results.

Finally, the global glacier sea-level rise contribution estimates of Frederikse et al. (2020), excluding Greenland and Antarctic periphery, agree well with ours for the more recent time intervals they specify (1957 - 2018 and 1993 - 2018), while our estimates lie at the very low end of the confidence interval given for the whole time interval they studied (1900 - 2018). This is presumably due to the modeling approach that their estimates in early years rely on, which includes estimations of disappeared and missing glaciers that are not included in the RGI. The increase of global glacier mass loss estimates this causes declines throughout the 20th century by roughly 66 % (Parkes and Marzeion, 2018).

Regarding regional values, Table 3 shows that roughly two-thirds of our global mass loss estimate during 2006 - 2018 occurred in the Greenland periphery and the North American continent. A large amount of the global uncertainty originates from these regions as well. Comparing our regional mass change estimates for recent years to those in the literature (Ciracì et al., 2020; Wouters et al., 2019; Zemp et al., 2019), the most obvious discrepancy can be found in estimates for the Southern Andes,

where our ensemble mean is substantially smaller and even positive in earlier periods shown in Table 3 (e.g., 1901 - 1940) , caused mainly by the model setup forced with 20CRV3 reanalysis data. The opposite is true for the regions Arctic Canada (North) and Svalbard, where our estimate is larger than those previously published. This might be caused by the relatively large portion of area draining into marine-terminating glaciers in those regions, since glacier-ocean interactions are not included in the model we applied and the calibration applying solely atmospheric forcing might thus be problematic. Moreover, our regional estimates for the Greenland periphery and Alaska in the most recent period (2006 - 2016) are close to each other, while Alaska lost significantly more mass according to Zemp et al. (2019) or Ciracì et al. (2020) during that time. This could be indicative of our mass loss estimates for Alaska being too small, or those for the Greenland periphery being too high. Two regions for which our estimates are significantly larger than in the previously published literature are Central Asia and South Asia West. This might be caused by a negative bias of the in-situ measurements used for calibration and validation in this region.

Another region of interest is the Russian Arctic, for which no in-situ measurements of annual specific mass balances were available in the data we used for calibration and validation. That is either because the sparse amount of measurements for that region was not covering at least three years for the individual observed glaciers or we did not find a link of the glaciers to the RGI. Also, more than half of the glacierized area in the Russian Arctic region drains into marine-terminating glaciers. For this region we estimate an average mass change rate of $0.08 \pm 0.02$ mm SLE yr$^{-1}$ during 2002 - 2016, while Wouters et al. (2019) estimate $0.03 \pm 0.01$ and Zemp et al. (2019) $0.07 \pm 0.03$ mm SLE yr$^{-1}$. Regarding the whole period that Zemp et al. (2019) provide estimates for in that region (1951 - 2016), the average they find is $0.05 \pm 0.02$ mm SLE yr$^{-1}$. Our average estimate for that period and region lies at $0.06 \pm 0.01$ mm SLE yr$^{-1}$. This shows that our model results do not, as is the case globally, agree well with GRACE-derived data in that region, but they still do with one other previously published estimate although we did not have calibration/validation data at hand in this region.

Thus, while we find an overall good agreement of our global mass change estimates with previously published ones, besides most of those derived with GRACE data, there are still significant differences in certain regional estimates. These require further research into the causes, and hence point to potential model shortcomings as, for example, area initialization, geometric scalings, neglecting frontal ablation, debris cover and radiation, as well as into the calibration and validation procedure applied. Incorporating frontal ablation processes of marine-terminating glaciers in the model and calibration procedure as well as distinguishing between mass loss above and below sea-level would be crucial model developments for enhancing the reliability of modeled global glacier sea-level rise contribution estimates. Generally speaking, the influence of ice-ocean interaction on global glacier mass loss remains elusive, although one study that conducted global glacier mass change projections, applying a simple frontal ablation parameterization, estimated a total of ca. 10 % global glacier mass loss caused by frontal ablation (Huss and Hock, 2015).

Although the largest potential of reducing the global uncertainty relevant to sea-level rise estimates is in strongly glaciated but little observed regions (e.g., Greenland periphery), reducing it in less glaciated regions (e.g., Southern Andes) could still

be valuable concerning hydrological changes and hence water availability. Future studies on mass loss reconstructions could benefit from addressing the above mentioned processes that are neglected thus far and from expanding the validation framework applied here in such a way that it would be able to include geodetic mass balance estimates as well as the uncertainties attached to in-situ/geodetic reference data. This is because in-situ measurements of annual specific mass balances are not only sparse and heterogeneously distributed, but reportedly negatively biased in some regions (Gardner et al., 2013). Since geodetic measurements provide glacier mass change data for much larger areas than in-situ/glaciological measurements, they add considerably to our understanding of glacier mass change,. Unfortunately, they are not yet standardized and readily available as the in-situ data are, making it unpractical to use them in the validation framework we applied.

Concerning the high uncertainty of mass change estimates during the early 20th century, it would be beneficial to have a suite of models that is able to hindcast glacier changes over that period, similar to intercomparison efforts for projections (Marzeion et al., 2020). More reanalysis products covering that time interval and also the Antarctic periphery would certainly help to constrain global estimates and their uncertainty more, although this might be of limited value due to the lack of historical validation data. In order to not only rely on reanalysis data, it would also be possible to run the model with data of climate models' historical experiments. A comparison with results obtained by applying reanalysis data could bring valuable insights into how, why and where reanalysis and climate model forcings of the mass balance model differ. Finally, the application of a robust initialization method (e.g. Eis et al., 2019, 2021) could help to understand if and how inaccuracies of the initialization method propagate through the modeled period.

## 5 Conclusions

A multi-objective optimization of a global glacier mass balance reconstruction model, forced with an ensemble of meteorological data sets, was presented. We demonstrated that it is possible to find statistically well performing sets of model parameters for each forcing data set, but that we cannot robustly identify which model setup is the most reliable when applied outside of the temporal and spatial domain of available in-situ mass balance validation data. However, one data set (CERA20C) can be identified as performing worse than the others. Disagreement between ensemble members is to a large degree attributable to differences in the forcing data in times and at locations where few validation and calibration data are available. The differences in the forcing data result in diverging glacier mass change estimates, especially in the first half of the 20th century, and thus are a major part of our ensemble estimates' uncertainty. Although our estimates lie within the uncertainty range to most of the previously published global estimates, they agree less with those derived from gravimetry (GRACE) data and show significant differences to the literature in individual regions. Our reconstruction ensemble average indicates that around the 1930s mass loss rates from glaciers were comparable to those of today. This finding is weakened by the lack of an explicit mass balance-surface elevation feedback in the model we applied, though, and it might hence be that mass change rates during the 1930s were actually smaller than today. According to our results, the increase in mass loss until the 1930s was followed by a phase of mass loss deceleration until roughly 1980. The glaciers' contribution to sea-level rise has been accelerating again since then,

despite an indication of the their retreat to more favorable climatic conditions, i.e. higher altitudes. Our results also indicate
that this acceleration was partly driven by decreasing amounts of solid precipitation at glacier locations from ca. 1980 onward.
This implies that the enhanced atmospheric warming not only increased ablation rates, but probably lowered the amount of
snow the glaciers received as well, notwithstanding a slight increase in total precipitation.

*Data availability.* The reconstructed, optimized time series of area, mass, and mass change for the RGI regions, except Antarctic and Sub-
antarctic, are available at PANGAEA: https://doi.pangaea.de/10.1594/PANGAEA.931657. Global results on a 0.5° grid, excluding the afore-
mentioned region, can be obtained by contacting the corresponding author.

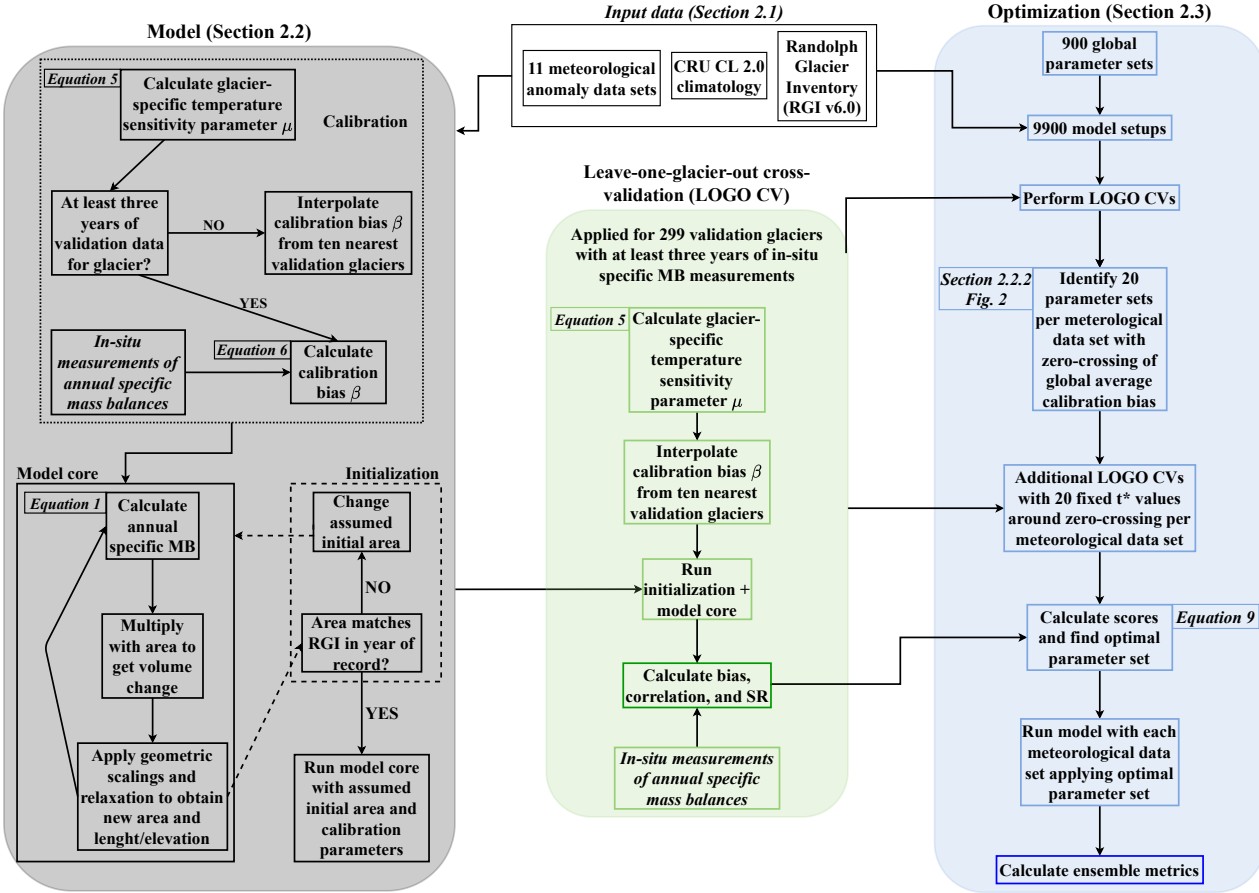

**Figure 1.** Flowchart depicting the modeling and optimization chain.

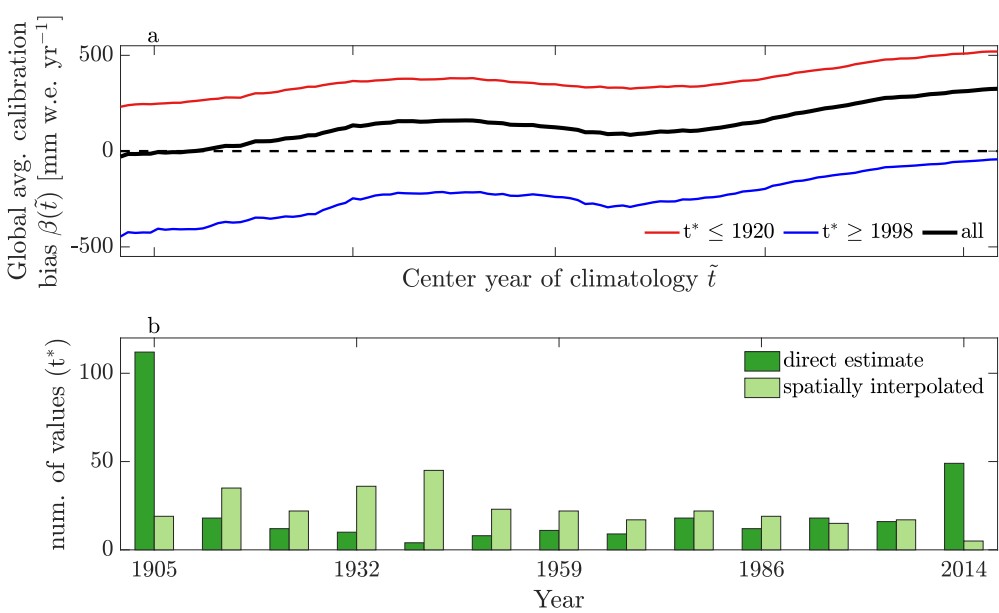

**Figure 2. (a)**: Average calibration bias $\beta$ as a function of the center year of a climatological window around $\tilde{t}$ for validation glaciers showing the lowest calibration bias around the center years $t^* \leq 1920$ (red, n = 132) and $\geq 1998$ (blue, n = 72) as well as the weighted average of all validation glaciers (black, n = 298). **(b)**: Distributions of $t^*$ directly estimated from Eq. 5 and Eq. 6 (green), and spatially interpolated as in Marzeion et al. (2012, light green). Values in both panels are derived from the cross-validation procedure (see section 2.2.2 and 2.3) with the optimized CRU TS 4.03 model setup.

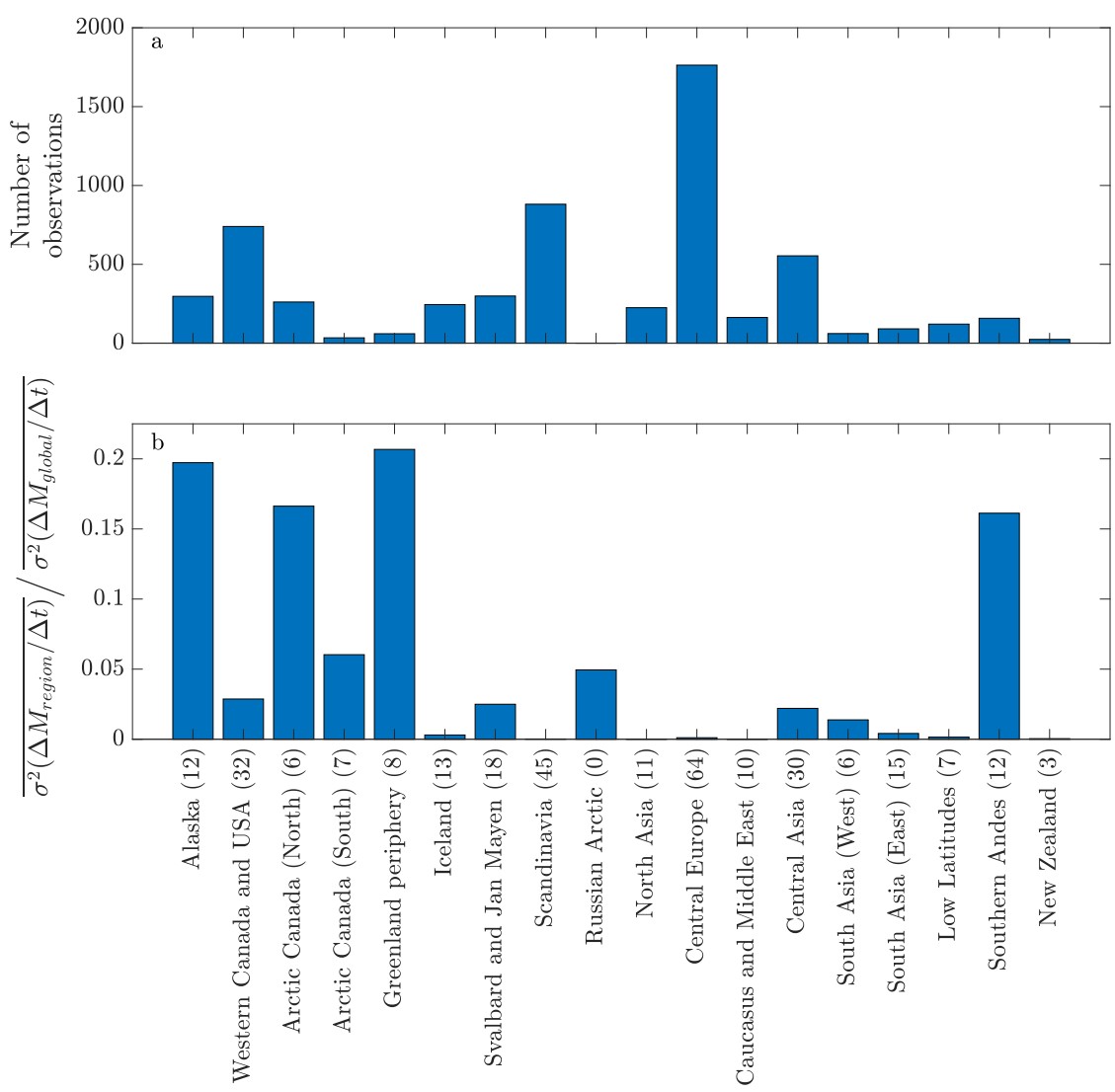

**Figure 3. (a)**: Number of specific annual mass balance observations available for calibration and validation in each RGI region. **(b)**: Fraction of ensemble variance of global mean mass change rate ($\Delta M$ / $\Delta t$) in the modeled period (1901 - 2018) attributable to each RGI region. In brackets is the number of glaciers used for calibration and validation in each region.

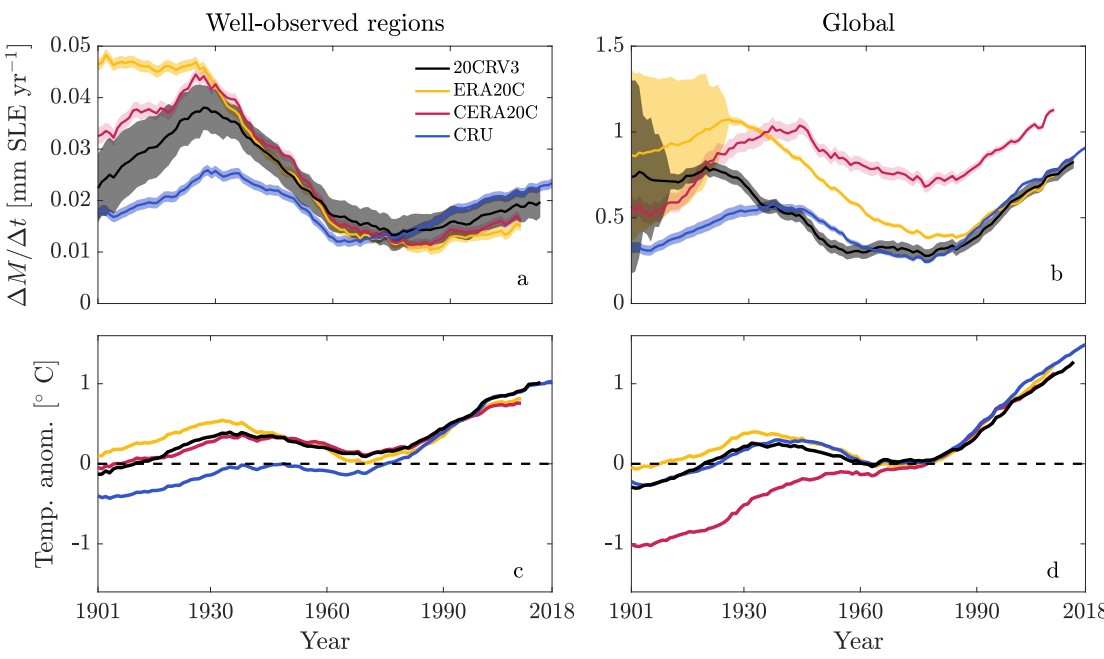

**Figure 4.** Mass loss rate estimates for meteorological forcing data sets with whole 20th century coverage: **(a)** averaged over well-observed regions (Western Canada and USA, Scandinavia, Central Europe, and Central Asia) and **(b)** globally. Average temperature anomalies at glacier tongue locations in **(c)** well-observed regions and **(d)** globally, weighted by glacier area. In all graphs, 31-year moving averages are shown for clarity.

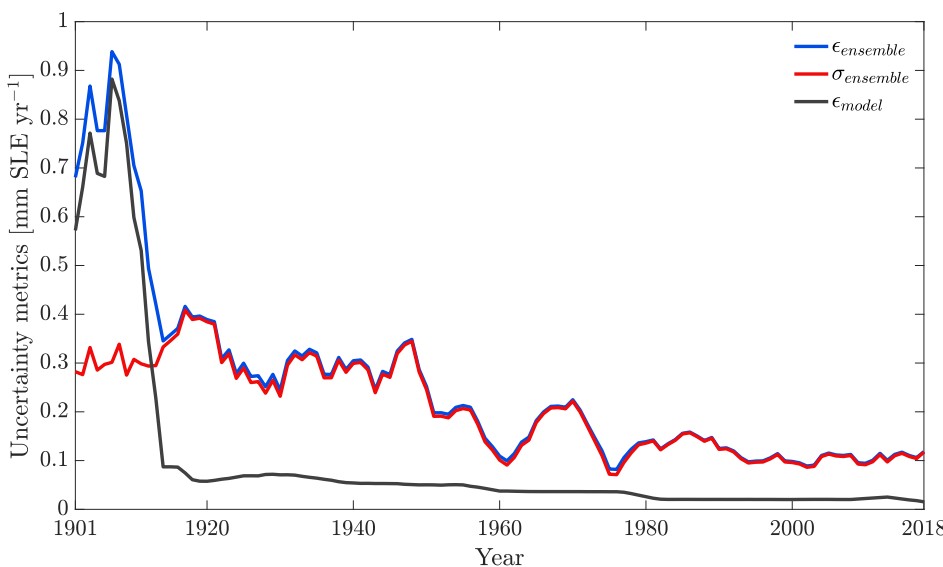

**Figure 5.** 5-year moving average of the temporal evolution of model uncertainty metrics for annual global mass change rates. $\epsilon_{ensemble}$ is the total uncertainty, i.e. combined model uncertainty ($\epsilon_{model}$) and ensemble spread ($\sigma_{ensemble}$; see Eq. 11).

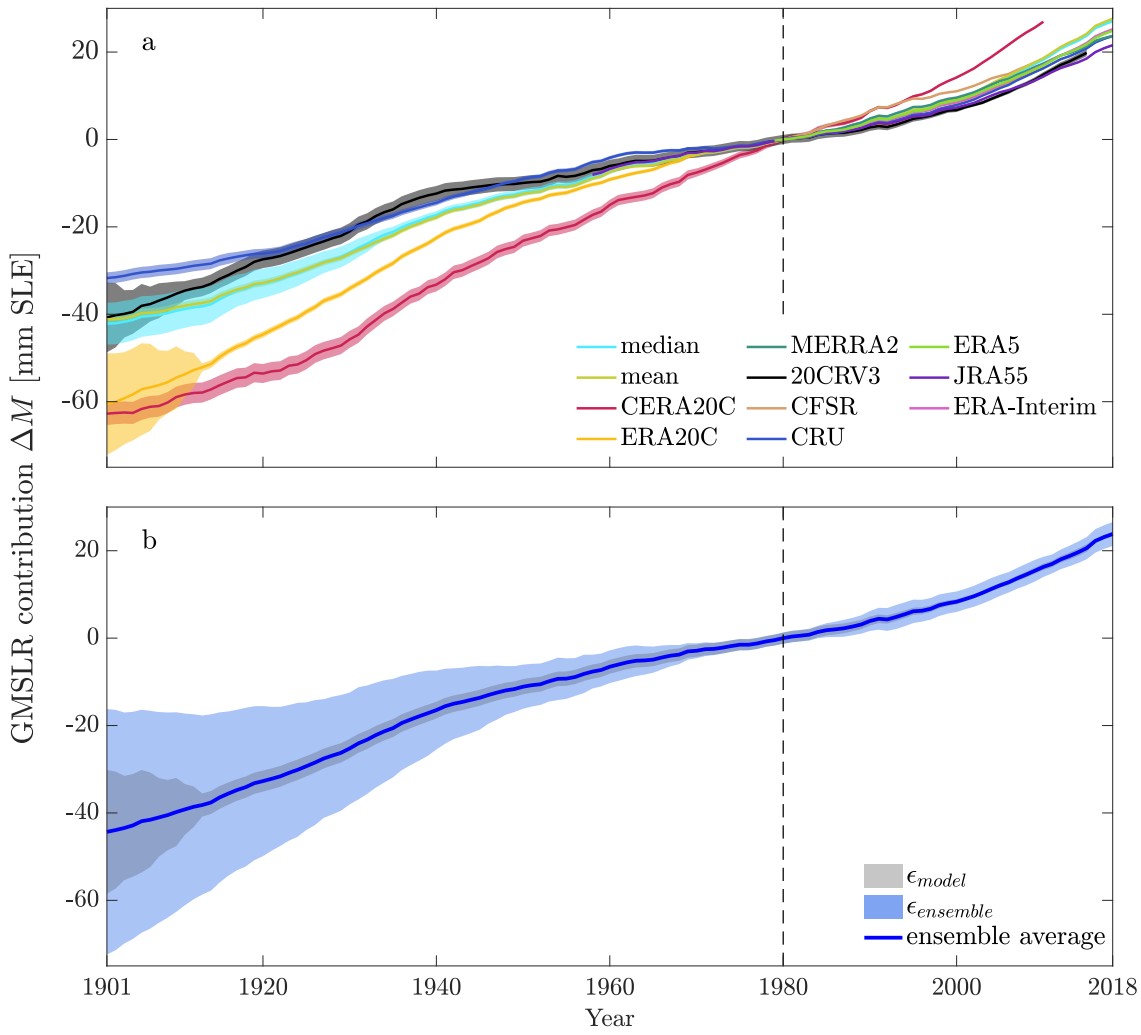

**Figure 6. (a)**: Estimates of temporally accumulated global mean sea-level rise (GMSLR) contribution relative to 1980 for all forcing data sets. Shaded areas are model uncertainties calculated for individual model setups. **(b)**: Ensemble mean output estimate. Shaded areas are the mean model uncertainty (grey, $\epsilon_{model}$; see Eq. 10) and total ensemble uncertainty (blue, $\epsilon_{ensemble}$; see Eq. 11), which are shown at the 90% confidence level. Results of the CERA20C forcing are excluded from the ensemble mean (see section 3.1.1).

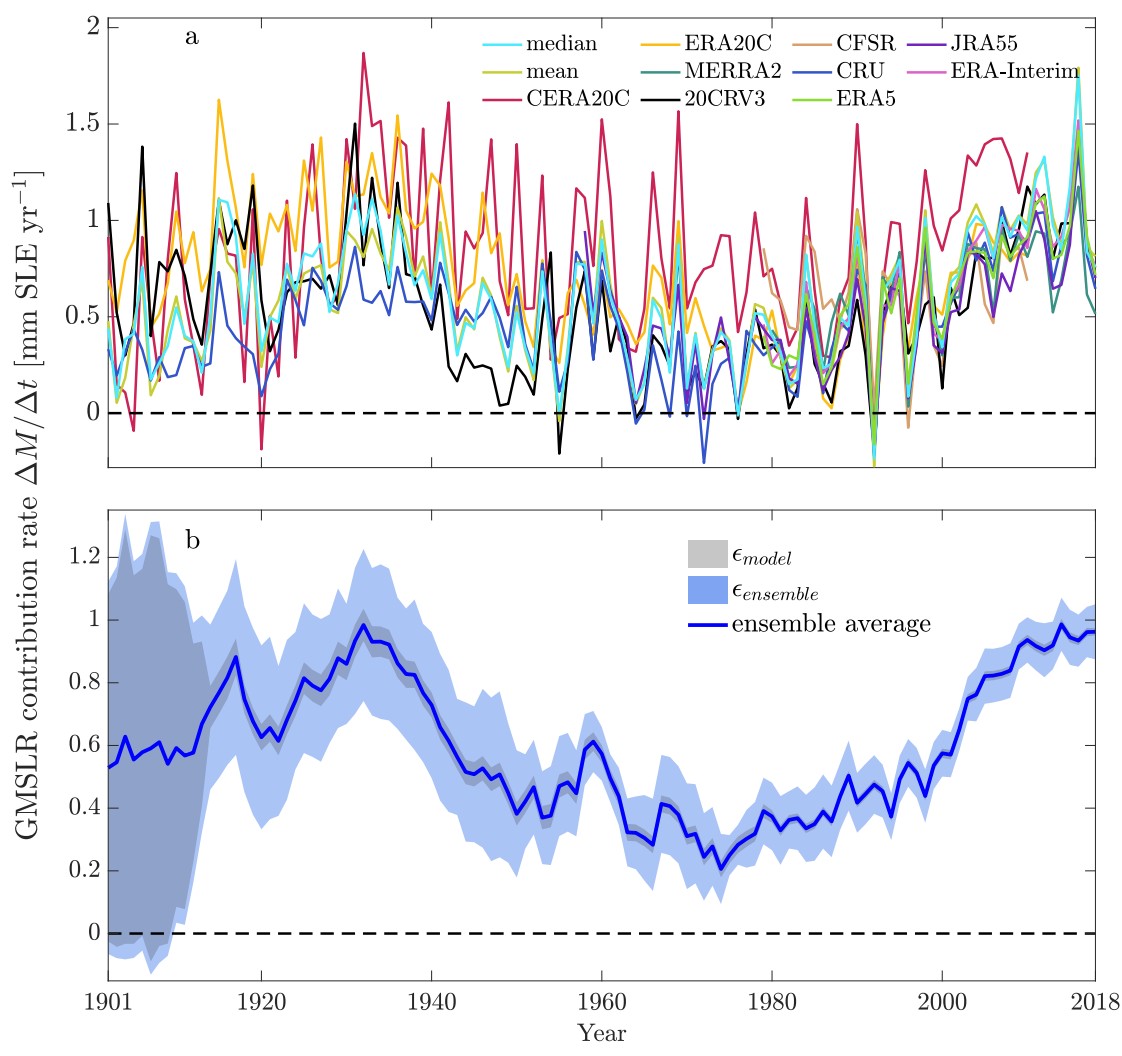

**Figure 7. (a)**: Annual glacier mass change rates expressed in GMSLR contribution rate for all forcing data sets. **(b)**: Mean of ensemble output mass change rates. A 5-year moving average is shown for clarity. Shaded areas are the mean model uncertainty (grey, $\epsilon_{model}$; see Eq. 10) and total ensemble uncertainty (blue, $\epsilon_{ensemble}$; see Eq. 11), which are shown at the 90% confidence level. Results of the CERA20C forcing are excluded from the ensemble mean (see section 3.1.1). Note the different vertical scales of the panels.

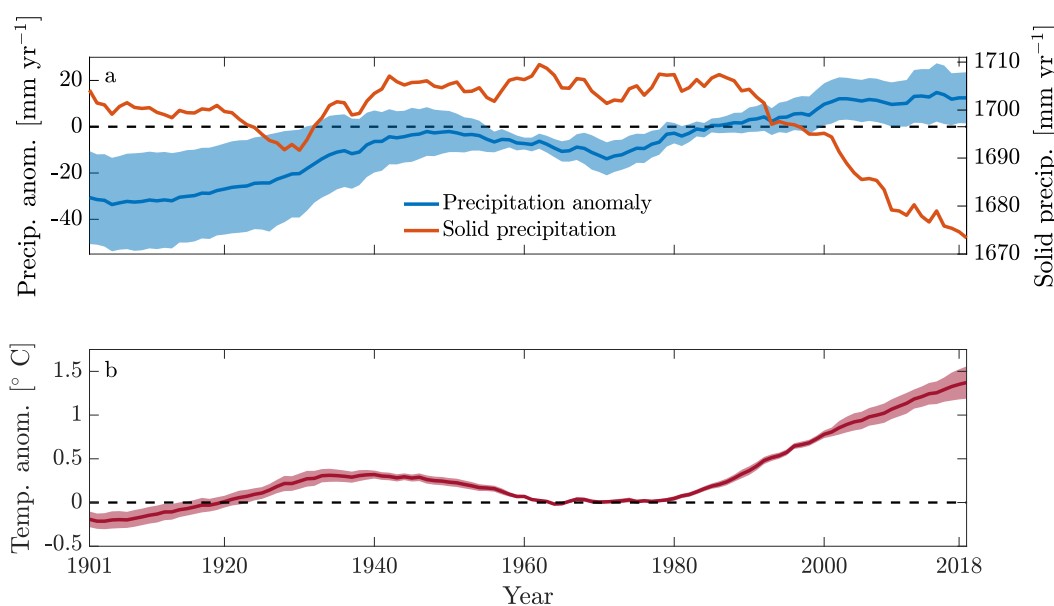

**Figure 8. (a)**: Global mean annual precipitation anomaly relative to 1961 - 1990 and amount of solid precipitation. **(b)**: Global mean annual temperature anomaly relative to 1961 - 1990. The shadings show $\pm 1\,\sigma$, i.e. standard deviation among meteorological forcing data sets. Values in both panels are 31-year moving averages of the ensemble mean at glacier tongue locations and weighted by glacier area, except for the graph of solid precipitation, which is based on the mean forcing input data. Since scales of computed solid precipitation might vary between ensemble members depending on model parameters (see Eq. 3 and 4), the computation of an average, especially with a temporally varying number of ensemble members, is less meaningful. Values of the CERA20C forcing are excluded from the ensemble mean (see section 3.1.1).

**Table 1.** Resolution and time range of the meteorological data sets used as boundary conditions.

| Label used in text & figures | Resolution [°] | Time range | Publication |
|---|---|---|---|
| 20CRV3 | 2 x 2 | 1871 - 2014 | Slivinski et al. (2019) |
| CFSR | 0.5 x 0.5 | 1979 - 2010 | Saha et al. (2010) |
| CRU CL 2.0 | 10' x 10' | 1961-1991 (climatology) | New et al. (1999) |
| CRU TS 4.03 | 0.5 x 0.5 | 1901 - 2018 | Harris and Jones (2020), Harris et al. (2014) |
| CERA20C | 0.28 x 0.28 | 1900 - 2010 | Laloyaux et al. (2018) |
| ERA5 | 0.5 x 0.5 | 1979 - 2018 | Copernicus Climate Change Service (C3S) (2019) |
| ERA20C | 1.13 x 1.13 | 1900 - 2010 | Poli et al. (2016) |
| ERA-Interim | $\sim$0.7 x 0.7 | 1979 - 2018 | Dee et al. (2011) |
| JRA55 | 1.25 x 1.25 | 1958 - 2018 | Kobayashi et al. (2015) |
| MERRA2 | 0.63 x 0.63 | 1980 - 2018 | Gelaro et al. (2017) |

**Table 2.** Values of the performance measures for each tested data set, applying the best-scored parameter set. Values behind coefficients in brackets display non-weighted averages (see text). For the mean and median model output, the score with/without CERA20C is displayed. The total number of cross-validated glaciers used for the respective data set is $n$. $A_M$ is the percentage of glacier area (as recorded in the RGI) covered by the glaciers the model was able to initialize. The last five columns contain the selected global parameters (see Eq. 1 - 5).

| | Bias [mm w.e. yr$^{-1}$] | R | SR | RMSE [mm w.e. yr$^{-1}$] | score | n | $A_M$ [%] | t* | $T_m$ [°C] | $T_{p.\,s.}$ [°C] | $\gamma_p$ [%/100 m] | a |
|---|---|---|---|---|---|---|---|---|---|---|---|---|
| 20CRV3 | 14.1 (57.4) | 0.61 (0.56) | -0.02 (-0.13) | 978.0 (816.9) | 2.30 | 295 | 86.6 | 1978 | 2 | 2 | 1 | 2.5 |
| CERA20C | 79.0 (30.9) | 0.56 (0.52) | 0.07 (0.13) | 747.1 (715.3) | 1.38 | 274 | 83.9 | 1902 | 2 | 0 | 4 | 3 |
| CFSR | 0.0 (-15.1) | 0.60 (0.56) | 0.13 (0.03) | 804.2 (740.9) | 2.13 | 276 | 93.4 | 1917 | 1 | 4 | 3 | 2 |
| CRU TS 4.03 | 0.6 (11.5) | 0.63 (0.59) | 0.01 (-0.05) | 739.6 (695.9) | 2.62 | 298 | 97.8 | 1917 | 0 | 4 | 4 | 3 |
| ERA-Interim | 1.7 (14.7) | 0.64 (0.61) | 0.02 (-0.02) | 715.1 (674.0) | 2.70 | 297 | 97.6 | 1907 | 0 | 4 | 3 | 3 |
| ERA5 | 0.0 (10.4) | 0.67 (0.64) | 0.02 (-0.04) | 714.0 (680.5) | 2.96 | 299 | 97.8 | 1919 | 0 | 4 | 5 | 2.5 |
| ERA20C | 4.6 (-38.9) | 0.58 (0.54) | 0.05 (0.03) | 791.0 (735.8) | 2.04 | 281 | 97.2 | 1902 | 0 | 4 | 1 | 3 |
| JRA55 | -2.0 (18.7) | 0.64 (0.60) | 0.00 (-0.03) | 701.1 (670.5) | 2.66 | 298 | 98.6 | 1915 | -1 | 4 | 5 | 3 |
| MERRA2 | 0.2 (14.8) | 0.64 (0.60) | 0.01 (-0.04) | 719.7 (685.4) | 2.72 | 299 | 97.5 | 1908 | 0 | 4 | 1 | 3 |
| mean in. | 9.1 (-13.6) | 0.66 (0.63) | 0.13 (0.07) | 767.5 (714.5) | 2.59 | 299 | 93.3 | 1901 | 1 | 4 | 3 | 3 |
| median in. | 16.3 (-5.5) | 0.66 (0.62) | 0.02 (-0.03) | 725.9 (679.7) | 2.75 | 299 | 93.8 | 1901 | 1 | 4 | 1 | 3 |
| mean out. | 3.9 (23.7) | 0.68 (0.65) | -0.06 (-0.14) | 704.1 (651.7) | 2.84/2.90 | - | - | - | - | - | - | - |
| median out. | 18.3 (26.3) | 0.67 (0.64) | -0.06 (-0.13) | 680.1 (640.2) | 2.67/2.77 | - | - | - | - | - | - | - |

**Table 3.** Ensemble mean regional mass change rate estimates (in mm SLE yr$^{-1}$) for 18 primary RGI regions over different time periods. Results of the CERA20C forcing are excluded from the ensemble mean (see section 3.1.1). Values in brackets for Southern Andes exclude 20CRV3 results (see section 4).

| | 1901 - 1940 | 1941 - 1980 | 1901 - 1990 | 1971 - 2018 | 1993 - 2018 | 2006 - 2018 |
|---|---|---|---|---|---|---|
| 1 Alaska | 0.08 ± 0.17 | 0.05 ± 0.03 | 0.06 ± 0.07 | 0.09 ± 0.03 | 0.13 ± 0.04 | 0.14 ± 0.04 |
| 2 Western Canada and US | 0.07 ± 0.10 | 0.02 ± 0.01 | 0.04 ± 0.05 | 0.0017 ± 0.006 | 0.020 ± 0.006 | 0.023 ± 0.003 |
| 3 Arctic Canada (North) | 0.10 ± 0.10 | 0.06 ± 0.04 | 0.08 ± 0.05 | 0.09 ± 0.02 | 0.13 ± 0.02 | 0.18 ± 0.03 |
| 4 Arctic Canada (South) | 0.08 ± 0.10 | 0.03 ± 0.03 | 0.06 ± 0.04 | 0.036 ± 0.008 | 0.05 ± 0.01 | 0.07 ± 0.02 |
| 5 Greenland periphery | 0.17 ± 0.13 | 0.05 ± 0.06 | 0.10 ± 0.08 | 0.07 ± 0.04 | 0.12 ± 0.05 | 0.15 ± 0.08 |
| 6 Iceland | 0.01 ± 0.01 | 0.012 ± 0.006 | 0.010 ± 0.008 | 0.012 ± 0.002 | 0.020 ± 0.002 | 0.023 ± 0.004 |
| 7 Svalbard | 0.05 ± 0.02 | 0.03 ± 0.01 | 0.04 ± 0.01 | 0.06 ± 0.01 | 0.08 ± 0.01 | 0.09 ± 0.02 |
| 8 Scandinavia | 0.003 ± 0.002 | 0.0026 ± 0.0009 | 0.003 ± 0.001 | 0.0024 ± 0.0007 | 0.0044 ± 0.0009 | 0.0057 ± 0.0007 |
| 9 Russian Arctic | 0.06 ± 0.10 | 0.05 ± 0.01 | 0.05 ± 0.05 | 0.06 ± 0.01 | 0.07 ± 0.01 | 0.09 ± 0.02 |
| 10 North Asia | 0.002 ± 0.002 | 0.0004 ± 0.0003 | 0.002 ± 0.001 | 0.0016 ± 0.0004 | 0.0025 ± 0.0005 | 0.030 ± 0.004 |
| 11 Central Europe | 0.001 ± 0.002 | 0.001 ± 0.002 | 0.002 ± 0.001 | 0.004 ± 0.002 | 0.006 ± 0.002 | 0.006 ± 0.001 |
| 12 Caucasus and Middle East | 0.002 ± 0.002 | 0.0001 ± 0.0004 | 0.001 ± 0.001 | 0.0008 ± 0.0003 | 0.0016 ± 0.0006 | 0.0021 ± 0.0005 |
| 13 Central Asia (North) | 0.06 ± 0.04 | 0.05 ± 0.01 | 0.05 ± 0.02 | 0.049 ± 0.006 | 0.052 ± 0.005 | 0.057 ± 0.007 |
| 14 Central Asia (West) | 0.04 ± 0.02 | 0.04 ± 0.01 | 0.04 ± 0.01 | 0.034 ± 0.005 | 0.034 ± 0.006 | 0.04 ± 0.01 |
| 15 Central Asia (South) | 0.02 ± 0.02 | 0.02 ± 0.01 | 0.02 ± 0.01 | 0.015 ± 0.005 | 0.017 ± 0.003 | 0.018 ± 0.005 |
| 16 Low Latitudes | 0.01 ± 0.01 | 0.006 ± 0.007 | 0.007 ± 0.008 | 0.003 ± 0.003 | 0.004 ± 0.002 | 0.004 ± 0.002 |
| 17 Southern Andes | -0.03 ± 0.10 | -0.01 ± 0.07 | -0.01 ± 0.08 | 0.01 ± 0.04 | 0.01 ± 0.03 | 0.01 ± 0.02 |
| | (0.007 ± 0.002) | (-0.002 ± 0.002) | (0.004 ± 0.001) | (0.009 ± 0.001) | (0.016 ± 0.002) | (0.022 ± 0.003) |
| 18 New Zealand | 0.002 ± 0.004 | 0.003 ± 0.004 | 0.003 ± 0.003 | 0.0016 ± 0.0005 | 0.0013 ± 0.0004 | 0.016 ± 0.0008 |
| Global (without peripheral Antarctic and Subantarctic) | 0.73 ± 0.46 | 0.42 ± 0.21 | 0.56 ± 0.27 | 0.56 ± 0.06 | 0.75 ± 0.07 | 0.90 ± 0.12 |

**Table 4.** Different estimates for mean annual glacier mass change rates (in mm SLE yr$^{-1}$) over time periods given in the respective literature. Uncertainties at the 90% level for all values except for Wouters et al. (2019) and Zemp et al. (2019), which are at the 95% level. Estimates exclude Greenland and Antarctic periphery; columns marked with * include Greenland periphery. Cells highlighted in bold font are published estimates disagreeing significantly with ours as discussed in section 4. Results of the CERA20C forcing are excluded from the ensemble mean (see section 3.1.1).

| | 1961 - 2016 | 1992 - 2016 | 2003 - '10 | 2002 - '16 | 2002 - '18 | 2006 - '16 | 1902 - '10* | 1961 - 2010* | 2003 - '09* | Data type |
|---|---|---|---|---|---|---|---|---|---|---|
| Ensemble mean | 0.45 ± 0.06 | 0.60 ± 0.06 | 0.66 ± 0.08 | 0.72 ± 0.07 | 0.72 ± 0.07 | 0.76 ± 0.08 | 0.57 ± 0.22 | 0.46 ± 0.08 | 0.83 ± 0.10 | Modeling |
| Ciraci et al. (2020) | - | - | - | - | 0.78 ± 0.08 | - | - | - | - | GRACE |
| Zemp et al. (2019) | 0.4 ± 0.3 | 0.6 ± 0.1 | 0.6 ± 0.3 | 0.7 ± 0.2 | - | 0.7 ± 0.2 | - | 0.4 ± 0.1 | 0.8 ± 0.3 | Glaciological & geodetic |
| Bamber et al. (2018) | - | **0.48 ± 0.05** | - | - | - | - | - | - | - | Synthesis |
| Wouters et al. (2019) | - | - | - | **0.55 ± 0.09** | - | - | - | - | - | GRACE |
| WGMS (2015) | - | - | - | - | - | - | - | 0.57 | 1.12 | Glaciological |
| WGMS (2015) | - | - | - | - | - | - | - | 0.85 | 1.05 | Geodetic |
| Marzeion et al. (2012) | - | - | - | - | - | - | - | - | 0.96 ± 0.12 | Modeling |
| Marzeion et al. (2015) | - | - | - | - | - | - | 0.62 ± 0.05 | 0.49 ± 0.05 | 0.78 ± 0.15 | Modeling |
| Cogley (2009) ° | - | - | - | - | - | - | - | 0.54 ± 0.05 | 0.75 ± 0.07 | Glaciological & geodetic |
| Leclercq et al. (2011) ° | - | - | - | - | - | - | 0.78 ± 0.19 | 0.58 ± 0.15 | 0.87 ± 0.64 | Length observations |
| Gardner et al. (2013) | - | - | 0.59 ± 0.07 | - | - | - | - | - | 0.70 ± 0.07 | Synthesis |
| Jacob et al. (2012) | - | - | **0.41 ± 0.08** | - | - | - | - | - | - | GRACE |

° Updated in Marzeion et al. (2017)

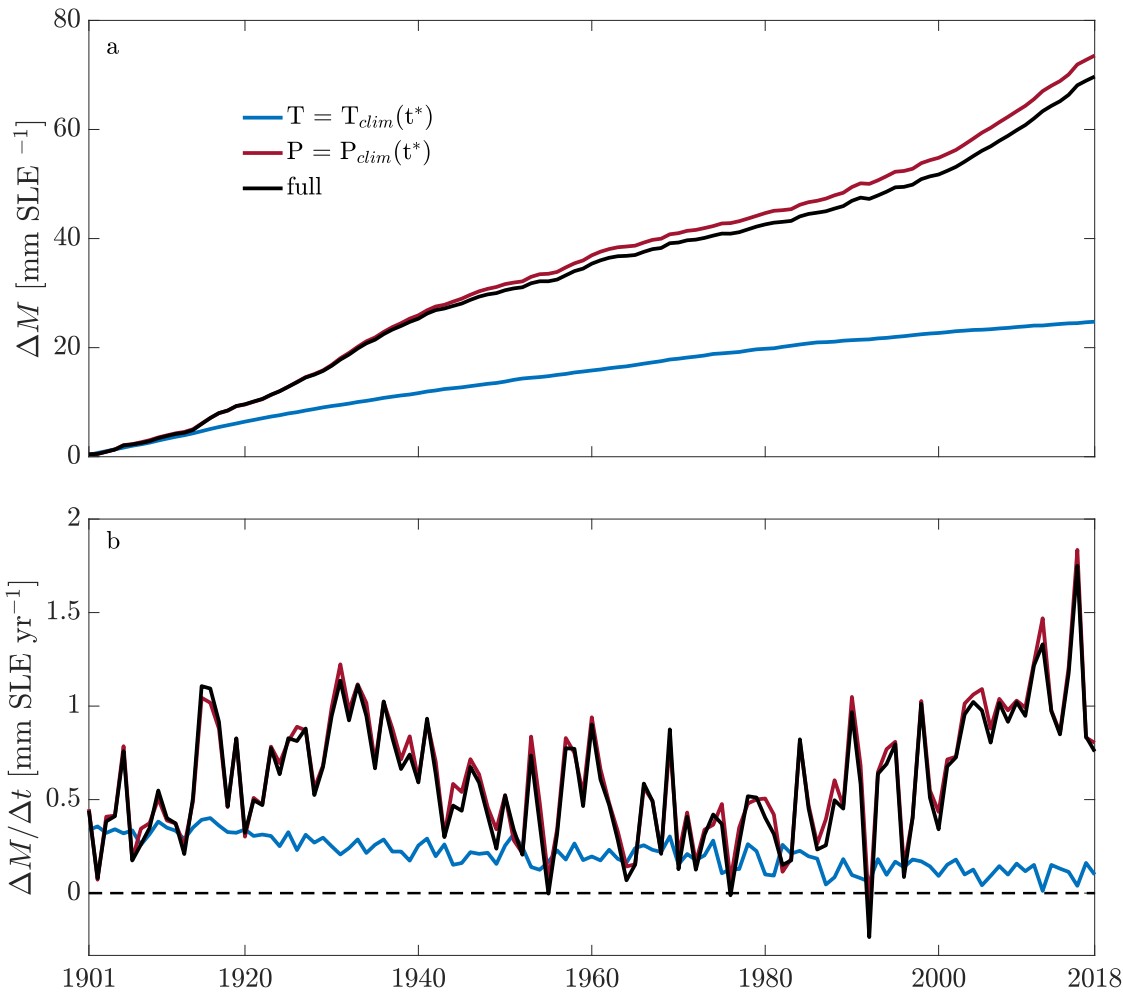

**Figure A1.** Estimated influence of temperature and precipitation anomalies on global glacier mass change. Red lines are modeled glacier mass change with temperature held constant at the climatology around $t^*$. Blue lines are modeled glacier mass change with total precipitation held constant at the climatology around $t^*$. Black lines represent the model run with full forcing. Note that in the case of this forcing data set we found optimal model performance with $t^* = 1901$, which implies the climatology only includes 16 years. **(a)**: Estimates of GMSLR contribution relative to 1901. **(b)**: Estimated annual GMSLR contribution.

*Author contributions.* B. Marzeion designed the research and contributed to the manuscript. J.-H. Malles contributed to designing the research, conducted the simulations and statistical evaluation, and wrote the manuscript.

*Competing interests.* The authors declare no competing interests.

*Acknowledgements.* This work was partly supported through the Sea Level Budget Closure Project (SLBC_cci) as part of the ESA Climate
Change Initiative (CCI) Programme. BM was supported by the Deutsche Forschungsgemeinschaft (DFG), grant no. MA6966/4-1. JHM was
supported by the DFG through the International Research Training Group "Processes and impacts of climate change in the North Atlantic
Ocean and the Canadian Arctic" (IRTG 1904/3 ArcTrain).

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
