# Peer review of "20th century global glacier mass change: an ensemble-based model reconstruction"

_The Cryosphere, 2020_

## Referee Comment (RC1) · Regine Hock (Referee) · 7 Dec 2020

Regine Hock, 5 December 2020

**Malles & Marzeion, Global glacier mass change, TC**

This paper provides a global 20th century glacier mass balance reconstruction driven by an ensemble of climate data sets. Few studies exist that estimate the time series of 20th century global-scale mass changes. It is an important and excellent paper that should be published after some revisions. Overall it is very well written, however, I am not quite sure about the methodology, which is not clear to me (see below).

**General**

1. **Methodology/structure**: The methodology is somewhat confusing in particular calibration and validation. 2.1.2 is about mass balance calibration, then 2.2. is about data and then 2.3 about optimization which I assume is 'calibration'. However the 3. Paragraph in 2.3. talks about validation. All this is confusing. I suggest to avoid that 2.2 Data is sandwiched between calibration and validation and that everything about calibration/optimization is in the same section. I suggest to have 'data' before model description/calibration. It was difficult to follow the description of the model calibration not knowing what data are actually used.

2. **Data**: it is unclear what mass-balance data are used. Are in situ measurements just those derived by the glaciological method, i.e. ignoring all the thousands of geodetic balances? The WGMS is referenced which includes geodetic balances. Also what is the temporal resolution. Are annual balances compared? Seasonal?
   Overall, far too little information is given about the mass balance data. How many glaciers? How many annual measurements? Some info on the temporal distribution? How much is e.g. before 1950 or 1930. All this information is important to evaluate the methodology. Perhaps a figure can illustrate the data density somehow.

3. **Calibration**: It is unclear how the model was calibrated? Please clarify which steps are done for which glacier, which parameters are glacier-specific and what is compared ? I suggest that the model is introduced first entirely independent of how parameters are obtained (section 2.1.1), then a section about calibration that includes all info how the parameters are obtained. This is currently dissected and difficult to follow. This may then allow to avoid repetition of Eq 1 and 4. It is unclear how equation 4 is applied. Where do the parameters come from? Are parameters optimized (matching observations) prior to applying eq 4? Overall this section appears the weakest and hard to grasp what was done how.

4. **Sea level contribution sign convention**: It appears that glacier mass losses expressed in SLE are treated as negative numbers. This is not consistent e.g., with the mass balance glossary (Cogley et al., 2011) and all IPCC reports. When a glacier loses mass, the mass change is negative but the resulting contribution to the ocean is positive. While a loss for the glacier system, it's a gain for the ocean. All SLE figures in the current manuscript can be misinterpreted since a sea level rise in the current paper is shown with a drop. This should be adjusted in all figures and the text for consistency with the literature.

**Details**

1. It should be clearer what the domain is of this paper: all glaciers outside the ice sheets but including those in the periphery of Greenland. I also suggest to replace Greenland by Greenland periphery in all figures/tables to avoid confusion. Also make clear somewhere which glaciers in Greenland you include (RGI connectivity level?)
2. Abstract line 10: What is 'temporal' here? Of annual balances?
3. Intro: Line 26: glacier instead of glaciers'
4. Intro: line 27: rephrase 'more distant past' since this can be misunderstood as thousands of years or even more ago
5. Intro: line 27-28: the statement is a bit odd. If there are no insitu measurements but we had entire 20$^{th}$ century satellite measurements it would be fine, so the logical connection here needs some work.
6. Line 28: in-situ measurement densities → in-situ measurements
7. Line 30: mass loss → mass change
8. Line 41: remove yet
9. Intro: Line 71: better? 'five model parameters'. Unclear what is meant by global?
10. 2.2.1: Line 122: Where do initial values for volumes come from? I assume only glaciers currently existing in RGI6 are considered, i.e. glaciers that have melted are ignored?
11. Line 124: add version of RGI and reference.
12. Line 134: I assume you allow some uncertainty around 0? Important to add that this is specific mass balance (the value depends on the unit): → 'in a zero specific annual mass balance
13. Line 141: add 'annual' to mass balance
14. Line 141 and many other places: the term 'respective' seems to be used a lot in the way it is used in German but not necessarily in English. Can be deleted in most places or in some replace by 'corresponding'
15. 2.2 Data: line 178: can you give some measure how much/often
16. Lines 188-189: perhaps the 3 short sentences can be shortened/combined?
17. Line 191: annual? Glacierwide specific? See general comments. This section would benefit from considerable expansion and clarification.
18. 2.3 Optimization: Line 204: 900? How come?
19. Line 230: what is 'a respective data set'? do you mean 'validated glacier and meteorological forcing data set'?
20. 3.1. Performance: Line 249: what is 'validated meteorological data sets'. Did you validate th climate sets with insitu observations?
21. Line 252, what data sets? You mean: for most forcing data sets?
22. Line 266: a few spelling errors
23. Line 270: 55 unique pairs: unclear? Why 55?
24. Line 271: remove 'respective'
25. Line 281: what is 'low-scored model setups' ? Do you mean the meteo forcing that scored low?
26. 3.2 Difference ….
    The title of the header is confusing

27. Line 290: 'respective' → corresponding
28. Line 329: 'for we do' ???
29. Chapter 4: Line 374: to reduce ?
30. 5 Discussion: Can the first paragraph be broken up in 2 or 3? It's long.
31. Line 387: reformulate 'more correct'. It's either correct or not
32. Line 386-387: this is not necessarily true. You seem to assume that GRACE and the observations are perfect and have no other errors
33. Lines 390ff: lakes are mentioned but the discussion appears not comprehensive. Many other processes can delay the discharge of melt water to the oceans (e.g. groundwater, dams, human uses and transfers, evaporation etc). All difficult to quantity but perhaps the discussion here can be more balanced.
34. Line 391-392: This makes it sound as if Gardner came up with the bias but did not do anything about it, only Zemp did years later. This is not correct. Because of the bias the 'best' method was used in each region and direct glaciological data only used in regions where the measurement density is high.
35. Discussion: The more negative balances compared to others in the period well-studied period 2003-2009 is striking while consistent with Marzeion et al 2012 based on the same model. This is an important discrepancy. Can this be explained in terms of the calibration process? It appears the model was validated/calibration only with direct observations? Why was the model not validated with those thousands of recent geodetic balances covering entire regions in some places.
36. Line 395-400: a number of language issues/convoluted language sometimes difficult to follow.
37. Conclusions: Line 420: what is model setup?

**Figures/Tables**
1. Some multiplot figures are referred to as 'upper/lower' panel, others have labels, a,b, ... Would be good to use the same method across the paper (perhaps the latter).
2. Figure 2: caption: clarify what 1 mass-balance obs is? One annual balance on one glacier? If so N is number of glaciers x number of years per glacier: would be good if that could be distinguished in the figure, or somehow visualizing N for time period classes.
3. Figure 3: can be shortened by: ….. (a) averaged over …. Asia) and (b) globally ….
4. Figure 3: meteorological forcing data sets
5. Figure 5: I suggest to adjust the tick labels to decadal periods, i.e. not 1901 but 1900, not 1921 but 1920 …. Also mark year 1980 somehow (vertical line)
6. Figure 5 and 6: I suggest that the color scheme is adjusted that the median and mean stick out better (line thickness? Color choice?). They are hard to identify.
7. Figure 5: clarify in caption of the ensemble mean is with or without the outlier model (also elsewhere)
8. Figure 7: no need to shorten the legend text in both subplots. There is lots of space, so it can be fully spelled out.
9. Table 1: Resolution: what's the unit?
10. Table 2:
   a) for respective → for each forcing data set ?

b) Last sentence is unclear (A_M). What does it mean? Able to initialize?

c) To make the table stand on its own would be good to spell out what the variables are (at least those that are not obvious) or at least group them somehow, e.g. xxxx are model parameters to determine melt and snow accumulation (Equation xxxx).

11. Table 3: different number of decimals for the same variable is unfortunate.
    - caption: add? 'for 18 primary RGI regions'

12. Table 4: hard to get a grasp on trends. I suggest to order the columns in time chronological order according to the first year of each period.
    - Why is there no entry for Zemp for 2003-2009. They have annual time series for all regions and the period means can easily computed from their supplementary data.
    - Why is Jacob et al. (GRACE) not included?
    - WGMS 2015 is not in reference list? I am not sure this should be included here at all since (I assume) this is just a 'quick' extrapolation of the measurements rather than a thorough analyses as all others. The bias of the in situ measurements is known.
    - For a reader it would be good to get some information how this estimates have been obtained since different methods have (sometimes known) biases. Perhaps a footnote reporting for each model in a few words what the source is (GRACE, extrapolation of observed insitu glacierwide specific balances, mean of insitu mcombination of XXX and XXX, mass balance model …)
    -

---

## Referee Comment (RC2) · Anonymous Referee #2 · 14 Dec 2020

**General comments**

Malles & Marzeion quantify the sea-level contribution from the world's glaciers during the 20[th] century, and its associated uncertainty. They use a published global mass-balance model with some new developments regarding the representation of glacier response time in the mass-balance calibration. The model is applied to all glacierized regions in the world except Antarctica. A number of key model parameters are optimized against in-situ observations of glacier mass balance, and the performance of nine different meteorological forcing datasets ('ensemble members') is assessed. For each region, they also provide uncertainty estimates associated with the model and the ensemble of forcing datasets.

The authors account for total uncertainty as being the combination of the uncertainty in the forcing and the model itself. They find that uncertainty in the atmospheric boundary conditions (the forcing) used may exceed uncertainties related to model parameters. Mass-loss estimates from the different ensemble members diverge more for time periods with lesser known atmospheric conditions (e.g. early 20[th] century), which are found to often coincide with areas where mass-balance observations are scarce (areas outside W. Canada + USA, Scandinavia, Central Europe, Central Asia).

Generally there are few studies that reconstruct global mass loss from glaciers all the way back to the early 20[th] century, let alone provide robust estimates of associated model and forcing uncertainty. This is a well-written manuscript that clearly fits within the scope of TC, and is in my view a timely contribution that furthers our understanding of global glacier mass loss, and provides some excellent strategies for how to deal with uncertainties in mass loss estimates. With this said, I have a number of suggestions for improvement. My major comments are mostly related to a somewhat confusing structure of the paper, an incomplete description of the model, and a limited discussion of the results. The authors should be able to address these comments without running additional model simulations. If comments are properly addressed, I think the paper is highly likely to deserve publication in TC.

**Major comments**

*Structure and model description*
The way the Data and Methods are presented is a bit confusing. First you explain the model equations, before we have an idea of what data you use. I think this makes the model description unnecessarily hard to follow (see below). One option would be to swap 2.1 and 2.2, with some minor rewriting of cross-references. Alternatively, if you keep the current structure, you should make sure that enough information about the data is included for us to understand the model setup.

The description of the mass-balance model would also benefit from more detail. While the most important model equations are included, and you state that more details can be found in Marzeion et al. (2012), I think a bit more rationale behind the model equations and their

underlying assumptions would strengthen this section and help the reader. Right now, the model details are stated without much justification or reasoning behind the model choices. This includes the length-area-volume scalings and relaxation, which currently are only mentioned in one sentence without detail or rationale.

Section 2.1.2 is important, because here you explain how you account for glacier response time, which in my view is an exciting novelty of this study. Unfortunately I found this section a little difficult to follow. Partly this may be because we're not really sure where you're going without knowing the data you use. I think it needs to be articulated more clearly what this new model development regarding response time brings to the model, and why it is needed. It would also be useful with a concrete example illustrating your approach and the underlying logic, perhaps throwing in some typical numbers that help us to better understand what you are doing.

Finally, I think including a new flowchart-type figure of your experimental design would be something that would help the reader to get an overview of your approach. Such a figure could include for example the steps involved in your initialization, geometric scaling, calculation and interpolation of model parameters, calibration, validation and optimization steps, the datasets used and parameters optimized, and the output results.

*Comparison against geodetic mass balance data*
I wonder whether there is a particular reason for only comparing modelled mass balance against in-situ data, when geodetic data is available in many places and indeed also used in some of the previous studies you cite. Even a tentative comparison with some geodetic surveys would strengthen the study. Alternatively, the authors should clearly explain why this was not done, and why it would not be needed for the purposes of the current study.

*Discussion*
The discussion is nicely written, with interesting comparisons with previous studies. Specifically, I commend the authors for including some underlying explanations for the global mass loss evolution simulated (Fig 7). Although the focus of the present study is mainly on quantifying mass loss and estimate uncertainty, it would be interesting with an expanded discussion of the underlying causes of mass loss (relative role of temperature and precipitation, timing, and potential lag of mass loss). This could be based on what you have shown in Fig 7, and should also be put into context with previous literature. To be clear, I think a detailed discussion of the underlying explanations for mass loss over time for each region would be beyond the scope of this paper. Still, findings like "the increase in precipitation between ca. 1930 to 1950 was accompanied by a similar increase in solid precipitation, indicating that the warm anomaly at the same period was too weak reduce accumulation" (note missing 'to') are enticing and warrants further discussion.

Right now you present your results divided into the world's glacierized regions, which makes sense for your purpose. To understand the model performance in more detail and where

potential improvements can be found, I wonder whether it would be possible to also mention if the model performs better/worse for specific glacier types, e.g. alpine valley glaciers, ice caps, cirques etc., or at certain elevation ranges. If it is not possible to disentangle such detail, at least some discussion on what you expect based on the model equations, assumptions and limitations would be useful.

In addition, I would like to see a more thorough discussion on the steps ahead, preferably in light of the limitations of the current study, which themselves are barely discussed at all. The last paragraph (L414-416) hints at this, but could be expanded on. What recommendations do you have for future studies like the current one? How can models be improved? Do we have to live with these higher uncertainties in the early 20th century, or are there ways to improve estimates further? What recommendations do you have for scientists working with meteorological data sets (which you assessed in detail)? What about Greenland periphery, which contributes heavily to the model uncertainty AND contributes highly to global mass loss; can we improve here? And Antarctic glaciers, which you did not have meteorological data for, what should we do with them?

**Minor and technical comments**

*Abstract*
L3. would rather use "estimated" that "calculated" here, since "calculate" to me sounds too precise for what you're talking about.
L11. "The goal is…" I think this sentence needs to come earlier, before you describe the methods.
L13. "ensemble members" not really clear at this stage what you refer to here; are you talking about the different meteorological datasets, or the model parameters? I think you mean the met. data, but you should clarify.
L14-15. "… the availability of mass balance …" I think you can write this sentence in a more direct way, e.g. "where mass balance observations are abundant/in well-observed mass balance regions, forcing data is also better constrained/more known" or similar.
L16. "out-of-sample uncertainty" – what do you refer to here?
L17. "cross validation" vs "cross-validation" check consistency throughout the manuscript

*Introduction*
L21. "is constituting" -> constitutes (change to more direct writing)
L22. "constitute" again – change to are/comprise to avoid word repetition
L23. "their vanishing" – here it is unclear what the word "their" refers to, it could be referring to "regions of the world", implying that some regions of the world may vanish – which is obviously not what you mean. Rewrite.
L26. I think you should define what you mean by mass balance the first time you mention it. Also you need to make clear whether you talk about glacier mass balance, or surface mass

balance throughout the manuscript.

L28. lacking -> lack (more direct)

L28. "comprehensive in-situ measurement densities" – I think you can write this in a less convoluted way

L33. "temperature index melt models" – missing hyphen

L33. "melting precipitation" – do you mean melting of snow?

L34-36. This is useful info, but I think you could explain here why such scalings are used, and emphasize why ice dynamics are not included in these models. I would also add in Discussion how these types of scalings may have influenced your results.

L44. I guess adding additional parameters not only makes model optimization more complex, but also increase model uncertainty?

L48-49. Not clear at this stage how the cited study using OGGM links to your work. Do you follow their approach? On L37-38 you state that you don't include ice dynamic processes, so the reader is left somewhat confused here.

L58-59. "heterogeneously distributed" – I assume you mean within and across geographical regions here? Also I think you can split this sentence into two separate sentences.

L67. I think "metric" would be more appropriate than "parameter" here.

L55-79. The words evaluation, calibration, validation, and optimization are used somewhat interchangeably in this section. Please check that you're consistent, and clarify what you mean by these terms.

*Data and Methods*

L94. "as before" -> as in Marzeion et al. (2012)

L97. Please clarify what you mean by "leave-one-glacier-out" in this context.

L99-100. Beta* is estimated by spatially interpolating from the ten closest glaciers with 3+ years of in-situ observations. This is fine, but because in-situ observations are very unevenly distributed (which is one of your main points in the paper), the constraints on beta* will vary greatly across different regions. In regions with plenty of observed glaciers, the interpolation will likely work better than in sparsely observed regions. I think you should explain this explicitly to the reader, and how this influences your results.

L104-114. If I understand your model correctly, you assume that the fraction of solid precipitation (snow) varies linearly with glacier elevation. If this is the case I think this could be explained in words, in addition to stating the equations. Also you should state why this assumption is sensible for your purposes, and whether this assumption may break down in some glaciological or climatic settings.

L121-125. The scaling procedure and the underlying assumptions should be explained in more detail, see Major comments above. The iterative approach finding the initial area (and volume?) of each glacier needs a bit more elaboration too. For how many glaciers did the iterative procedure work, and in what regions mostly (if any patterns can be seen)?

L138. How many glaciers have 3+ years of in-situ measurements, and are thus included in this calculation?

L147. exactly what is "not generally the case" here?

L151. Change to (a) instead of "upper panel" – and do so throughout the manuscript

L157. mid -> middle

L158. "It" – are you referring to Fig 1b ?

L164. mid -> middle

L169-170. "no longer" I assume you mean as opposed to what was done by Marzeion et al. (2012) ?

L183. they were -> full temporal coverage were

L187. of -> in

L188. the modeled time interval – I think you can state here what this exact time interval is.

L193. Please state why you ignore the uncertainties in the in-situ observations (for simplicity, because of computational constraints, because they don't matter for your purposes?)

L192-194. What about geodetic mass balance estimates? See Major comments.

L196. a -> an

L197-203. How were these ranges selected? Based on previous literature, based on first-principles?

L225. I would try to avoid starting a sentence with "E.g." – replace with "For example" or similar. Otherwise a very useful and nicely written paragraph!

L230. a respective -> each (check use of "respective" throughout manuscript)

*Cross validation and uncertainty assessment*

L264. Avoid starting sentence with "E.g."

L266. be -> by the

L266. than than (double word)

L281. different -> differs

L284-286. I think you can split this sentence in two for readability.

L306. Rewrite to "In Fig. 3a, … " – and check figure cross-refs (use of B vs (b) etc.) and be consistent throughout the manuscript (for example, at L309 and 313 in this section)

L316-319. Great point, well said!

L333. "and for and" – redundant word

L334. A little weird to say that something "grows backward in time" – would rather say that it starts high, then decreases. Similarly on L348, when something "shrinks going back in time". I would rather say that the further back in time we go, the fewer meteorological datasets (ensemble members) are available.

*Global Glacier Mass Loss*

Overall a very interesting section!

L350. Not sure why all words in the section heading here are capitalized, why for other headings, they aren't. Check to be consistent.

L352. Use a/b instead of upper/lower panel when you refer to figures.

L354. "are diverging" -> diverge (more direct)

L355-358. Check the sign of your mass loss estimates (and in Fig 5) – one can get the impression that mass loss decrease over time with the current figure 5 version, which obviously not is what you want to show.

L358. please state explicitly what "the most recent period" is

L359. "earlier years" – a bit vague, please be more concrete

L366. state when "the end of the modeled period" is (2018?)

L369-374. Interesting findings regarding the explanation for the found changes in mass loss. These changes regarding the temperature and precipitation and the resulting changes in mass loss over time are interesting enough to be included in the Abstract, or at least in the Conclusions.

L376. state which periods you refer to by "the more recent periods"

*Discussion*

L383. Gardner et al citation should be without parentheses.

L385-386. I think you can remove the parentheses around "because … glaciers"

L402. Again please remind the reader when "the most recent time" is

L405. Please state when "earlier periods" is in this context

L410. Remind the reader that you're looking at Greenland periphery rather than Greenland

L414. Use of e.g. here breaks the flow of reading, would rather use wording like "for example"

L415. Would add '(e.g. Greenland periphery)' here as an example of such a largely glaciated but less observed region

In general a nicely written section, but missing a few key parts, see Major comments above.

*Conclusions*

L421. Not clear from the context here what "validation data" refers to, please be explicit on what type of data this is

L428. Be explicit on what has been decelerating and accelerating (the mass loss, not the glacier flow)

**Figures**

In general nicely crafted, clear figures! A great addition would be a flowchart-type figure explaining the model setup and experimental design, see Major comments.

*Figure 1*

I would write out what beta is in both the ylabel and the caption.

Add (a), (b) labels, and for subsequent figures.

Also I would write out what t_tilde and t* is in the caption, so that the reader can understand the figure without looking up the variables in the text.

Add cross-reference to the appropriate section after "cross-validation procedure".

*Figure 2*

Would write out the entire ylabel "number of observations" (can be split over two lines to fit the yaxis.

Caption: state when "the modeled period" is.

*Figure 3*

I think you can remove "RGI" in the title for (a), as it is not crucial to understand the figure and is just another abbreviation to check for the non-expert reader.

*Figure 5*
Check sign of mass loss, see Minor comments above.
I think you can also write out "Sea-level contribution" in the ylabel.

*Figure 6*
I would include the legend from Fig 5b also in Fig 6b.

*Figure 7*
You have space to write out temperature anomaly and precipitation anomaly in (a). In (b), you don't really need a legend, as there is only one time-series to show.

**Tables**
*Table 1*
Resolution – missing unit (degrees?)

---

## Referee Comment (RC3) · Anonymous Referee #3 · 22 Dec 2020

**Review Malles and Marzeion 20th Century global glacier mass change…**

This paper is about a method to find out which data set is most appropriate to calculate the volume change of glaciers and to provide a realistic uncertainty estimate on this value. As such the title is misleading. The title suggests that this is going to be the best estimate of the glacier volume loss. At the same time this is probably true but the paper needs to be reframed in that case which I would favor. Part of the methods can go to an appendix to free up some space to discuss results more extensively in context. There is very much emphasis on the results and limited context on other estimates in the literature aiming the same. The abstract for instance has only one sentence (the last) with results and is nearly completely about the methodology.

**Major points**

IF the paper is going to focus more on the result rather than on the method, one needs to address the fact that this paper is based on reconstructions of the climate based on data sets which are partly observational driven whereas projections are based on model results and the question is whether they can be connected. For this reason it would be interesting to use the CMIP ensemble over the historical period as an additional data set to see what that brings applied to the proposed method here.

I would be interested to see which fraction of the volume loss is related to temperature and which to precipitation. You could easily calculate that with your calibrated model by switching off the variation of one of the two.

I cannot understand the assumption on line 90 where you argue that mu_star and beta_star follow from the assumption that the present-day geometry is a steady state. We all know that the majority of the glaciers is not in a steady state? So even if you find that t_start is not present-day this seems a flawed approach. In addition I can buy the argument that t_star is not only a function of regional climatological history, but then you conclude based on that argument that it is allowed to take a global value for t_star??. I would conclude based on the same observation that t_star is not a global value, but requires to be estimated for each individual glacier and in fact you seem to do that around line 145. So I have serious problems with the concept outlined in lines 90-95. Following up you argue that u_star for each glacier follows from a global t_star (line 98) why?? and that b_star maybe spatially interpolated why?? Why not the other way around? or both spatially interpolated or based on t_star. I feel lost on the model description.

You use different data sets for temperature, but for precipitation you only use the anomalies of the different data sets based on the holy CRU CL 2.0 data set. Thereby you ignore all the uncertainty in the CRU data and consider it as the holy truth that doesn't seem to be correct to me. Furthermore you don't stress this at all.

Line 380. Table 4 is biased towards a comparison over the recent period, there is a variety of estimates for the entire century it would be interesting to compare you results also with those.

**Minor points**

Line 12: A little unclear what is meant with the different ensemble members. I presume you imply to say that most variation is caused by the input climatological rather than by the uncertainty in the parameters to constrain each model. Please rephrase to clarify.

Line 16: I don't understand the difference between total uncertainty and reconstruction uncertainty. Is ensemble uncertainty + reconstruction uncertainty the total uncertainty? Please rephrase to clarify. I am also unsure what the difference is between ensemble spread and ensemble uncertainty, both are used in the abstract or are they the same?

Line 19: The total uncertainty yields an uncertainty of only 8%, that is extremely small in my view and I doubt whether that is in line with budget closure studies of sea level over the last century.

Line 22: The paper of Slangen et al 2017 is on attribution that does not seem to be a very logical paper to refer to. Moreover the idea that glacier are important is much older. So an older reference seems more appropriate.

Line 28: densities can be left out.

Line 31: It is not trivial that an ensemble based data reconstruction adds to the uncertainty in modelling the future. I think that is only true for an ensemble based on models for the historical period but that is not what you do. You have to make this separation clearer. Though I would prefer you take such an ensemble CMIP5 or 6 onboard in your reconstruction.

Line 36: The word additionally suggest you take changes in the geometry and hypsometry on board, but do you really do that? And how? Having a response time scale is something different. Explicitly state that the time scale mimics only part of the effect caused by the dynamics, the height and length changes are not captured by a time scale, but normally embedded in the dynamical adjustment.

Line 40: strange sentence whether you resolve the energy balance or the dynamics are different entities. So the fact that an energy balance does not include the dynamics is not an argument to dismiss an energy balance approach.

Line 44: In the context of the paper it is very odd to argue that the additional degrees of freedom of an energy balance models are a limitation. You are aim is to get to an adequate estimate of the uncertainty, so the uncertainty space can much better be explored in an energy balance model than with a temperature-index approach.

Line 46: Computational limitations are not a constraint. The constraint is the lack of data on the geometry and they probably add little due to the large uncertainty in the forcing.

Line 51-55. I would expect here a summary of the outcome of Marzeion 2020 specifying how large model uncertainty is versus forcing uncertainty as discussed in Marzeion 2020.

Line 75. You can not assume the reader to know the Marzeion et al. 2012 paper, please explain in a few sentences the concept or explicitly refer to the next section. Then wrap up in line 84 a bit more the concept of the Marzeion 2012 model.

Line 96. Unclear what you mean with leave-one-glacier-out cross validation. Explain in more detail.

Line 110. Do you include all the uncertainties in parameters around precipitation. What does " with $T\_i\_\_zmax(t)$ " mean. I think you can leave that out the fact that you use (t) already indicates that this parameter is time-dependent.

Line 110: Unclear what to do with line 4 of the equation adding it or multiplying it with line 3? Syntax ambiguous.

Line 125: Why can you not find an initial volume which leads to the RGI volume at the end of your run. It is a pretty linear system, so I would expect that there is always an initial volume leading to the RGI volume. Please explain why this is apparently not the case.

Line 135. Can you explain why there should be a condition of $B(t\_hat)=0$. Many glaciers are not in steady state for the given geometry so why would this condition hold for at least one value of $t\_hat$.

Line 155. Figure 1 is not clear I don't understand the difference between the direct and spatially interpolated method.

Line 190. Mention for which time interval the RGI data are valid.

Line 220. MSE is not minimized if $sigma\_M=sigma\_O$ but in case R=0 then they are independent.

Line 222. I don't buy the argument that this is a complex model. Please provides stronger arguments.

Line 251 margin is not the right English word.

Line 266 twice than

Line 314 part D of *Figure* 3

Line 333 and for and?

Line 340. For completeness you have to mention that you assume $e\_ensemble$ to be independent of $sigma\_t$ such that you can pool them. This is not necessarily true and maybe one of the reasons why you end up with a total uncertainty of only 8%. You need to discuss the later in this context.

Line 358 specify the most recent period 2015-2018?

Line 370 and 371 consider leaving out "is"

Line 394 What is the explanation for the large difference with Zemp et al. 2019.

Line 399. You can easily be more specific as you have the Parkes and Marzeion numbers.

Line 410. I guess you need to take the marine terminating on – board in a final estimate in order to prevent the suggestion that glacier mass loss is extremely accurately known as you suggest with the 0.05 mm/yr value.

Line 426. It is worth bringing this to the abstract.

---

## Author Comment (AC1) · 5 Mar 2021

**Point-by-point response to the reviewers**

We would like to thank the three reviewers for the constructive and detailed feedback on our manuscript. We have done our best to address the concerns raised by the reviewers and here provide a point-by-point reply to each of the reviewers' comments, including an explanation to subsequent changes to the manuscript. After responding to concerns that were mentioned from all reviewers, we provide a point-by-point answer to the individual reviewers' comments. ("RC" stands for "reviewer comment", "AR" for "authors response")

5 "AR" for "authors response")

**Structure of the manuscript/methodology**

We acknowledge that the presentation of our methodology (including data usage) was too brief and not structured in a comprehensible manner. To improve this, we changed the sequence of the sections (data is now the first one in the methodological part of the paper), and extended the description of the model calibration process at those places where we think it was helpful, according to the reviews. We also included a new flowchart (Figure 1 in the new version of the manuscript) to allow a comprehensive overview of the methodology.

**15 Treatment of 'CL2 (strongly connected)' Greenland periphery glaciers**

In the former version of the manuscript we included glaciers classified as 'CL2 (strongly connected)' in the Randolph Glacier Inventory (RGI), i.e. glaciers that have a strong connection to the Greenland ice sheet. Since these are usually excluded from estimates regarding ice mass apart from the Greenland ice sheet, we excluded them in the revised version of the manuscript, for consistency with the literature. We updated the values in text, figures, and tables accordingly.

**Treatment of ensemble standard deviations/uncertainties**

Previously, we did not account for interannual covariances of the ensemble and hence treated its annual standard deviations to be independent from one another. Since there are considerable interannual covariances of the ensemble, this led to underestimated uncertainties for mean values over periods longer than one year. We now take the interannual covariances of the ensemble into account, thereby yielding higher uncertainty estimates. Additionally, we now applied Bessel's correction for the calculation of standard deviations for unknown populations means, which increases uncertainties slightly, too. We updated all values in text, figures, and tables accordingly.

30

**1 Answers to review of Regine Hock**

**1.1 General comments**

**RC:** "*Methodology/structure: The methodology is somewhat confusing in particular calibration and validation. 2.1.2 is about mass balance calibration, then 2.2. is about data and then 2.3 about optimization which I assume is 'calibration'. However*

- 35 the 3. Paragraph in 2.3. talks about validation. All this is confusing. I suggest to avoid that 2.2 Data is sandwiched between calibration and validation and that everything about calibration/optimization is in the same section. I suggest to have 'data' before model description/calibration. It was difficult to follow the description of the model calibration not knowing what data are actually used."
- AR: As stated in the first part of our general answer to the reviewers ("Structure of the manuscript/methodology"), which
  addresses concerns raised by all reviewers, we adjusted the structure and text. In this specific case, we put the data section first in the methodological part of our manuscript. This should make the explanation of the modeling process more coherent. We also added a flowchart figure to the manuscript in order to make the whole methodology clearer. Finally, concerning the differences between optimization/validation/calibration, we added the following to the second last paragraph of the introduction: "To avoid a confusion of the terms optimization, calibration, and validation, we briefly state our notions of these. Validation means
- 45 calculating metrics that relate model outputs to observed values in a certain variable and that quantify the estimated model uncertainty. Optimization refers to choosing the best parameter set with respect to the aims one sets regarding the validation. With calibration we mean the deduction of glacier-specific model parameters from observational data."

RC: "Data: it is unclear what mass-balance data are used. Are in situ measurements just those derived by the glaciological
method, i.e. ignoring all the thousands of geodetic balances? The WGMS is referenced which includes geodetic balances. Also what is the temporal resolution. Are annual balances compared? Seasonal? Overall, far too little information is given about the mass balance data. How many glaciers? How many annual measurements? Some info on the temporal distribution? How much is e.g. before 1950 or 1930. All this information is important to evaluate the methodology. Perhaps a figure can illustrate the data density somehow."

AR: Why we do not use geodetic measurements is stated in a comment below, which brings this issue up as well. We included more information about the data used in the validation procedure in the data section, i.e. that we use annual glaciological measurements and stated the following: "Those are 299 glaciers with a total of 5977 annual specific mass-balance measurements. Before 1950, only 110 annual records of 14 glaciers are contained in this data set. Of those 14 glaciers, 12 are situated in Central Europe and Scandinavia, and one in Alaska and Iceland each." Concerning the exact temporal distribution

60 we refer to WGMS (2020, Fig. 2.6), since the data is taken from the WGMS.

**RC:** "Calibration: It is unclear how the model was calibrated? Please clarify which steps are done for which glacier, which parameters are glacier-specific and what is compared ? I suggest that the model is introduced first entirely independent of how parameters are obtained (section 2.1.1), then a section about calibration that includes all info how the parameters are

65 obtained. This is currently dissected and difficult to follow. This may then allow to avoid repetition of Eq 1 and 4. It is unclear how equation 4 is applied. Where do the parameters come from? Are parameters optimized (matching observations) prior to applying eq 4? Overall this section appears the weakest and hard to grasp what was done how."

**AR:** As stated in the first part of our answer to the reviewers ("Structure of the manuscript/methodology"), which addresses concerns raised by all reviewers, we adjusted the structure and text to address these. In this specific case, we put the section on

- 70 calibration (section 2.2.2) directly after the introduction of the model (section 2.2.1) to make this more coherent. Additionally, we added explanations of the leave-one-glacier-out cross-validation procedure (in section 2.2.1) and of the calibration bias (in section 2.2.2), which is a crucial variable in the calibration process. We hope that this makes the description of the methodology more understandable. We think that Eq. 4 (Eq. 5 in the revised version of the manuscript) should be retained, for it being an modification of Eq. 1 (2) and thus beneficial in explaining the calibration of Eq. 1 (2).
- 75

80

**RC:** "Sea level contribution sign convention: It appears that glacier mass losses expressed in SLE are treated as negative numbers. This is not consistent e.g., with the mass balance glossary (Cogley et al., 2011) and all IPCC reports. When a glacier loses mass, the mass change is negative but the resulting contribution to the ocean is positive. While a loss for the glacier system, it's a gain for the ocean. All SLE figures in the current manuscript can be misinterpreted since a sea level rise in the current paper is shown with a drop. This should be adjusted in all figures and the text for consistency with the literature."

**AR:** We changed the sign of the sea-level contribution everywhere accordingly.

**1.2** Specific comments**

RC: "It should be clearer what the domain is of this paper: all glaciers outside the ice sheets but including those in the
periphery of Greenland. I also suggest to replace Greenland by Greenland periphery in all figures/tables to avoid confusion.
Also make clear somewhere which glaciers in Greenland you include (RGI connectivity level?)"

**AR:** We changed "Greenland" to "(the) Greenland periphery" everywhere. We also stated in section 2.1.2 that we now exclude 'CL2 (strongly connected)' glaciers, opposed to what we did in the previous version of the manuscript.

90 **RC:** "Abstract line 10: What is 'temporal' here? Of annual balances?"

AR: Yes. We changed the sentence leading to this term in order to make this clearer.

**RC:** *"Intro: Line 26: glacier instead of glaciers'"* **AR:** Done.

95

**RC:** "Intro: line 27: rephrase 'more distant past' since this can be misunderstood as thousands of years or even more ago" **AR:** Done.

**RC:** "Intro: line 27-28: the statement is a bit odd. If there are no insitu measurements but we had entire 20th century satellite measurements it would be fine, so the logical connection here needs some work."

**AR:** We added the fact that satellite measurements are lacking as well, and we thus have to rely on modeling for estimating changes outside of the observed domain. Thereby we hopefully close the logical gap.

RC: "Line 28: in-situ measurement densities -> in-situ measurements"

105 **AR:** Done.

RC: "Line 30: mass loss -> mass change" AR: Done.

110 RC: "Line 41: remove yet" AR: Done.

**RC:** *"Intro: Line 71: better? 'five model parameters'. Unclear what is meant by global?"* **AR:** We added one sentence on the meaning of "global" parameter in the introduction.

115

**RC:** "2.2.1: Line 122: Where do initial values for volumes come from? I assume only glaciers currently existing in RGI6 are considered, i.e. glaciers that have melted are ignored?"

AR: Initial volumes are estimated by area-volume scaling from the estimated initial areas. In section 2.2.1 we added some explanations of the scalings and initial area estimation. In the data section we now stated that only glaciers currently existing
120 in RGI v6.0 were used.

**RC:** "Line 124: add version of RGI and reference." **AR:** Done.

125 **RC:** "*I* assume you allow some uncertainty around 0? Important to add that this is specific mass balance (the value depends on the unit): -> 'in a zero specific annual mass balance"

**AR:** Added "specific". The idea is that the error/uncertainty of this assumption should be balanced by applying the calibration bias  $\beta$ . Hopefully, the whole complex of model calibration this refers to is clearer now with the changes to the manuscript outlined in the related responses above.

130

RC: "Line 141: add 'annual' to mass balance" AR: Done.

RC: "Line 141 and many other places: the term 'respective' seems to be used a lot in the way it is used in German but not necessarily in English. Can be deleted in most places or in some replace by 'corresponding'"

**AR:** Corrected throughout the manuscript.

RC: "2.2 Data: line 178: can you give some measure how much/often"

AR: Done. Five of the eleven data sets, or three of nine excluding the mean/median input, do not extend back to 1901.

**140**

**RC:** "*Lines 188-189: perhaps the 3 short sentences can be shortened/combined?*" **AR:** Done.

**RC:** "Line 191: annual? Glacierwide specific? See general comments. This section would benefit from considerable expan-145 sion and clarification."

**AR:** Annual glacierwide specific, yes. We expanded the data section.

RC: "2.3 Optimization: Line 204: 900? How come?"

**AR:** We explicitly state now that this is the number of possible parameter combinations within the given ranges.

150

**RC:** *"Line 230: what is 'a respective data set'? do you mean 'validated glacier and meteorological forcing data set'?"* **AR:** The part was rephrased to make it clear that this was done for every glacier in every cross-validation run.

**RC:** "3.1. Performance: Line 249: what is 'validated meteorological data sets'. Did you validate th climate sets with insitu 155 observations?"

AR: No. Changed to "optimized model setups", because that is what we meant there.

RC: "Line 252, what data sets? You mean: for most forcing data sets?"

**AR:** Also changed to "optimized model setups", because that is what we meant there.

**160**

**RC:** "*Line 266: a few spelling errors*" **AR:** Corrected.

RC: "Line 270: 55 unique pairs: unclear? Why 55?"

165 **AR:** Rephrased the sentence to make it clearer. Eleven model setups yield 55 unique pairs that can be tested for similarity.

RC: "Line 271: remove 'respective'"

AR: Done.

170 **RC:** "Line 281: what is 'low-scored model setups'? Do you mean the meteo forcing that scored low?"

**AR:** Yes. We added a reference to the table showing the scores for the optimized model setups in that sentence for clarification.

**RC:** "3.2 Difference .... The title of the header is confusing"

175 **AR:** Rephrased to improve clarity.

**RC:** "*Line 290: 'respective' -> corresponding*" **AR:** Done.

180 RC: "Line 329: 'for we do' ???" AR: for -> as.

RC: "Chapter 4: Line 374: to reduce ?" AR: Done.

185

**RC:** *"5 Discussion: Can the first paragraph be broken up in 2 or 3? It's long."* **AR:** Done.

RC: "Line 387: reformulate 'more correct'. It's either correct or not"

190 AR: Rephrased.

**RC:** "Line 386-387: this is not necessarily true. You seem to assume that GRACE and the observations are perfect and have no other errors"

**AR:** Connected to the previous comment, we now phrased this as "[...] these lower values might come closer to the glaciers' actual contribution to sea-level rise [...]", making it less absolute.

**RC:** "Lines 390ff: lakes are mentioned but the discussion appears not comprehensive. Many other processes can delay the discharge of melt water to the oceans (e.g. groundwater, dams, human uses and transfers, evaporation etc). All difficult to quantity but perhaps the discussion here can be more balanced."

200 **AR:** We added a Statement on this.

**RC:** "Line 391-392: This makes it sound as if Gardner came up with the bias but did not do anything about it, only Zemp did years later. This is not correct. Because of the bias the 'best' method was used in each region and direct glaciological data only used in regions where the measurement density is high."

205

210

5 **AR:** We rephrased the two sentences to describe that correctly.

**RC:** "Discussion: The more negative balances compared to others in the period well-studied period 2003-2009 is striking while consistent with Marzeion et al 2012 based on the same model. This is an important discrepancy. Can this be explained in terms of the calibration process? It appears the model was validated/calibration only with direct observations? Why was the model not validated with those thousands of recent geodetic balances covering entire regions in some places."

**AR:** We do not think this can be explained merely by using the same model/calibration. One reason for this is that we used newer and more validation data and a newer RGI version. Another one is the change in calibration strategy that we applied in this work. Finally, driven with other meteorological data, the Marzeion et al. 2015 estimates lie lower than ours when using the same model but the "old" calibration procedure. Concerning the comparison with other sources, we added the Zemp et al 2019

215 estimate for 2003 - 2009, which agrees with ours within uncertainty ranges, to Table 4. Secondly, we now excluded strongly connected (CL2) Greenland periphery glaciers from our estimates, which enhances the agreement with Zemp et al 2019 and Cogley 2009. It now even (very slightly) lies within uncertainty bounds of Gardner et al. 2013.

We added an explanation to the text in the Data section (2.1.2) why we did not use geodetic measurements in our calibration/validation framework: This is (i) because we need annual specific mass balance measurements of individual glaciers, which are not widely available, and (ii) where they are available for individual regions and limited time periods, they are far

- 220 which are not widely available, and (ii) where they are available for individual regions and limited time periods, they are far from readily usable for the global scale (i.e., many different formats, hydrological vs. calendar years, access limitations, etc.). We added a statement to the discussion pointing out the value of such observations, but also the need to coordinate formats etc. to improve usability.
- RC: "Line 395-400: a number of language issues/convoluted language sometimes difficult to follow."AR: Changed the segment to a less convoluted phrasing.

RC: "Conclusions: Line 420: what is model setup?"

AR: We introduce what we mean by model setup in the new flow chart.

230

**1.3 Tables/Figures**

**RC:** "Some multiplot figures are referred to as 'upper/lower' panel, others have labels, *a*,*b*, ... Would be good to use the same method across the paper (perhaps the latter)."

**AR:** Changed to labels a, b, ... everywhere.

**RC:** "Figure 2: caption: clarify what 1 mass-balance obs is? One annual balance on one glacier? If so N is number of glaciers x number of years per glacier: would be good if that could be distinguished in the figure, or somehow visualizing N for time period classes."

AR: Added "specific annual" to the caption to clarify the first question in this comment. Added number of validation glaciers
to each region's label. We added how many validation glaciers are available in each region in brackets behind the names. For the exact time coverage of in-situ measurements the reader is referred to WGMS (2020, Fig. 2.6), where those graphs already exist and are easily accessible.

**RC:** "Figure 3: can be shortened by: ..... (a) averaged over .... Asia) and (b) globally ....."

245 **AR:** Done.

**RC:** *"Figure 3: meteorological forcing data sets"* **AR:** Done.

**RC:** "Figure 5: I suggest to adjust the tick labels to decadal periods, i.e. not 1901 but 1900, not 1921 but 1920 .... Also mark year 1980 somehow (vertical line)"

AR: Done.

**RC:** *"Figure 5 and 6: I suggest that the color scheme is adjusted that the median and mean stick out better (line thickness? Color choice?). They are hard to identify."*

**AR:** The point we tried to make in the paper is that median and mean input forcing are "just another ensemble member", since we cannot assert which optimized model setup performs best outside the temporal and spatial validation domain. Therefore, we do not see a benefit in making them stand out, rather the opposite. The mean ensemble output, however, is shown in its own panels as our best estimate (and including uncertainty based on the entire ensemble).

260

265

**RC:** *"Figure 5: clarify in caption of the ensemble mean is with or without the outlier model (also elsewhere)"* **AR:** Added to all figure/table captions for which this applies.

**RC:** *"Figure 7: no need to shorten the legend text in both subplots. There is lots of space, so it can be fully spelled out."* **AR:** Done.

RC: "Table 1: Resolution: what's the unit?"

**AR:** The unit is degrees. Added.

**270 RC:** "Table 2: a) for respective -> for each forcing data set ? b) Last sentence is unclear  $(A_M)$ . What does it mean? Able to initialize? c) To make the table stand on its own would be good to spell out what the variables are (at least those that are not obvious) or at least group them somehow, e.g. xxxx are model parameters to determine melt and snow accumulation (Equation xxxx)."

**AR:** a) Done. b) we added the following to the second last paragraph of section 2.2.1: "A failure of the initialization for an individual glacier might occur when, for example, the calibration (see section 2.2.2) results in a very high temperature sensitivity for that glacier and the iterative search of an initial area is not able to capture the very large starting area necessary for the thereby assumed strong mass change." For glaciers where this is the case, we apply simple statistical upscaling. c) We added in the caption that the last five columns are the selected global parameters and referred to the equations where they are applied.

**280**

**RC:** "Table 3: different number of decimals for the same variable is unfortunate. - caption: add? 'for 18 primary RGI regions'"

**AR:** We chose the different number of decimals for different regions due to the different magnitudes of mass change and associated uncertainties, such that we don't have to give zero uncertainty for some regions. Added 'for 18 primary RGI regions'.

285

**RC:** "Table 4: hard to get a grasp on trends. I suggest to order the columns in time chronological order according to the first year of each period. - Why is there no entry for Zemp for 2003-2009. They have annual time series for all regions and the period means can easily computed from their supplementary data. - Why is Jacob et al. (GRACE) not included? - WGMS 2015 is not in reference list? I am not sure this should be included here at all since (I assume) this is just a 'quick' extrapolation of the measurements rather than a therework analyses as all others. The bias of the in situ measurements is known. For a reader it

290

the measurements rather than a thorough analyses as all others. The bias of the in situ measurements is known. - For a reader it would be good to get some information how this estimates have been obtained since different methods have (sometimes known) biases. Perhaps a footnote reporting for each model in a few words what the source is (GRACE, extrapolation of observed insitu glacierwide specific, mean of insitu mcombination of XXX and XXX, mass balance model ...)"

**AR:** - Changed order of columns.

295 - Added Zemp for all time periods available.

- Jacob et al. (2012) now in the table.

- We chose to keep WMGS 2015 in the table in order to show that our estimates are not merely reproducing those values, even though they are the basis of our calibration.

- Added footnotes to report the sources of the estimates.

**2 Answers to reviewer 2**

**2.1 General comments**

**2.1.1 Structure and model description**

RC: "The way the Data and Methods are presented is a bit confusing. First you explain the model equations, before we have
an idea of what data you use. I think this makes the model description unnecessarily hard to follow (see below). One option would be to swap 2.1 and 2.2, with some minor rewriting of cross-references. Alternatively, if you keep the current structure, you should make sure that enough information about the data is included for us to understand the model setup."

**AR:** As suggested we changed the structure such that the data part now comes first.

- **RC:** "The description of the mass-balance model would also benefit from more detail. While the most important model equations are included, and you state that more details can be found in Marzeion et al. (2012), I think a bit more rationale behind the model equations and their underlying assumptions would strengthen this section and help the reader. Right now, the model details are stated without much justification or reasoning behind the model choices. This includes the length-area-volume scalings and relaxation, which currently are only mentioned in one sentence without detail or rationale."
- **AR:** We added some sentences on the scalings and relaxation, and on the parameterization of solid precipitation in section 2.2.1.

RC: "section 2.1.2 is important, because here you explain how you account for glacier response time, which in my view is an exciting novelty of this study. Unfortunately I found this section a little difficult to follow. Partly this may be because we're not
really sure where you're going without knowing the data you use. I think it needs to be articulated more clearly what this new model development regarding response time brings to the model, and why it is needed. It would also be useful with a concrete example illustrating your approach and the underlying logic, perhaps throwing in some typical numbers that help us to better understand what you are doing."

AR: We do not directly account for response time other than in the rough relaxation approximation we apply, which was
already included in Marzeion et al. (2012). We added the following statement to the manuscript: "Concerning the relaxation, a response time scale of the volume-length change is estimated by assuming that smaller glaciers and those with higher mass turnover will react faster to volume changes". The idea of introducing t\* as a global parameter is to find a climatology with which glaciers globally, or rather the validation/calibration glaciers, would be in (near-)equilibrium, if they had had the present-day extent at that time. In the latter part of section 2.2.2, in combination with Fig. 1, we illustrate this for one exemplary model
setup. Here, the response time indirectly comes into play, for we no longer assume that t\* is only a function of regional climatology, as was done in Marzeion et al. (2012). If we do not follow that assumption anymore, the spatial interpolation of t\*

becomes invalid, and we replace it by the global optimization in section 2.3.

**RC:** "Finally, I think including a new flowchart-type figure of your experimental design would be something that would help the reader to get an overview of your approach. Such a figure could include for example the steps involved in your 335 initialization, geometric scaling, calculation and interpolation of model parameters, calibration, validation and optimization steps, the datasets used and parameters optimized, and the output results."

AR: Thank you for the suggestion, which we believe really should help the readers. We included a flowchart as Figure 1 in the Data and Methods part (Chapter 2) now.

340

**2.1.2 Comparison against geodetic mass balance data**

**RC:** "I wonder whether there is a particular reason for only comparing modelled mass balance against in-situ data, when geodetic data is available in many places and indeed also used in some of the previous studies you cite. Even a tentative comparison with some geodetic surveys would strengthen the study. Alternatively, the authors should clearly explain why this was not done, and why it would not be needed for the purposes of the current study."

**AR:** We extended the explanation regarding geodetic mass balances in the revised data section (2.1.2), there is a lack of a data set for geodetic measurements that would enable us to use those in our calibration/validation approach. If there was one with the homogeneity as well as spatial and temporal resolution of the WGMS one for in-situ measurements, it would be great for incorporating those in future studies.

350

345

**2.1.3 Comparison against geodetic mass balance data**

**RC:** "The discussion is nicely written, with interesting comparisons with previous studies. Specifically, I commend the authors for including some underlying explanations for the global mass loss evolution simulated (Fig 7). Although the focus of the present study is mainly on quantifying mass loss and estimate uncertainty, it would be interesting with an expanded discussion 355 of the underlying causes of mass loss (relative role of temperature and precipitation, timing, and potential lag of mass loss). This could be based on what you have shown in Fig 7, and should also be put into context with previous literature. To be clear, I think a detailed discussion of the underlying explanations for mass loss over time for each region would be beyond the scope of this paper. Still, findings like "the increase in precipitation between ca. 1930 to 1950 was accompanied by a similar increase in solid precipitation, indicating that the warm anomaly at the same period was too weak reduce accumulation" (note missing 'to') are enticing and warrants further discussion."

360

**AR:** We included a figure in the appendix (Fig. A1) based on two additional model runs: one holding temperature constant at the climatology around t\*, and one holding total precipitation constant at the climatology around t\*. We also added some sentences on this in the Discussion (section 4). In summary, it shows that precipitation anomalies play a minor role except for some regions like Scandinavia or New Zealand, which is consistent with previous knowledge.

**RC:** "Right now you present your results divided into the world's glacierized regions, which makes sense for your purpose. To understand the model performance in more detail and where potential improvements can be found, I wonder whether it would be possible to also mention if the model performs better/worse for specific glacier types, e.g. alpine valley glaciers, ice caps, cirques etc., or at certain elevation ranges. If it is not possible to disentangle such detail, at least some discussion on what you expect based on the model equations, assumptions and limitations would be useful."

370

**AR:** Unfortunately, it is not possible to disentangle this in great detail, since the validation data do not cover enough glaciers of different types (or even allow for a clear categorization). We see no significant correlation of the performance measures with, e.g., elevation range. We expanded the discussion of marine-terminating glaciers, for which we expect the model to perform worse based on the model equations, because we neglect frontal ablation.

375

**RC:** "In addition, I would like to see a more thorough discussion on the steps ahead, preferably in light of the limitations of the current study, which themselves are barely discussed at all. The last paragraph (L414-416) hints at this, but could be expanded on. What recommendations do you have for future studies like the current one? How can models be improved? Do we have to live with these higher uncertainties in the early 20th century, or are there ways to improve estimates further? What

380 recommendations do you have for scientists working with meteorological data sets (which you assessed in detail)? What about Greenland periphery, which contributes heavily to the model uncertainty AND contributes highly to global mass loss; can we improve here? And Antarctic glaciers, which you did not have meteorological data for, what should we do with them?"

**AR:** We expanded the discussion on the limitations of this work and recommendations for future studies. The main points are:

385 - Use geodetic mass change data for calibration/validation when readily available.

- Ensembles of different forcings as well as of different models can give valuable insights to where uncertainties, spatially, temporally, and/or in the construction of models, stem from.

- Multitemporal glacier outlines, especially for earlier years than those recorded in the RGI, would be great to constrain models.

**2.2 Minor and technical comments**

390 2.2.1 Abstract

**RC:** "L3. would rather use "estimated" that "calculated" here, since "calculate" to me sounds too precise for what you're talking about."

AR: Agreed, done.

395 RC: "L11. "The goal is..." I think this sentence needs to come earlier, before you describe the methods."AR: Placed the sentence before the description of the methods now.

**RC:** "L13. "ensemble members" not really clear at this stage what you refer to here; are you talking about the different meteorological datasets, or the model parameters? I think you mean the met. data, but you should clarify."

400 **AR:** Phrased it "[...] optimized model setups (i.e. ensemble members) [...]" now.

**RC:** "L14-15. "... the availability of mass balance ... " I think you can write this sentence in a more direct way, e.g. "where mass balance observations are abundant/in well-observed mass balance regions, forcing data is also better constrained/more known" or similar."

405 **AR:** Done.

**RC:** "L16. "out-of-sample uncertainty" – what do you refer to here?"

**AR:** Removed the term "out-of-sample" and phrased it: "[...] such that the cross-validation procedure does only partly capture the uncertainty of the glacier model".

**410**

**RC:** "*L17.* "cross validation" vs "cross-validation" check consistency throughout the manuscript" **AR:** Done.

**2.2.2 Introduction**

415 RC: "L21. "is constituting" -> constitutes (change to more direct writing)" AR: Done.

**RC:** "L22. "constitute" again – change to are/comprise to avoid word repetition"

AR: Done.

**420**

**RC:** "L23. "their vanishing" – here it is unclear what the word "their" refers to, it could be referring to "regions of the world", implying that some regions of the world may vanish – which is obviously not what you mean. Rewrite." **AR:** Done.

425 **RC:** *L26.* "*I* think you should define what you mean by mass balance the first time you mention it. Also you need to make clear whether you talk about glacier mass balance, or surface mass balance throughout the manuscript."

**AR:** Done. We also state more accurately now in the data and model description that our validation/calibration attempts are based on annual specific mass-balance data.

430 **RC:** *L28.* "lacking -> lack (more direct)"

AR: Done.

**RC:** "L28. "comprehensive in-situ measurement densities" – I think you can write this in a less convoluted way"

AR: We reformulated this now as follows: "This is the case for the last century or even more distant past as well (e.g.,
Goosse et al., 2018; Parkes and Goosse, 2020), since satellites able to observe Earth's surface did not yet exist for a large portion of the 20th century. Glaciers also lack comprehensive in-situ mass-balance measurements, at least before 1950, since they are mostly situated in remote locations (see Figs. 2.6 and 2.7 in WGMS, 2020)."

RC: "L33. "temperature index melt models" – missing hyphen"

440 **AR:** Done.

RC: "L33. "melting precipitation" – do you mean melting of snow?"AR: Yes. Clarified this in the text.

445 **RC:** "L34-36. This is useful info, but I think you could explain here why such scalings are used, and emphasize why ice dynamics are not included in these models. I would also add in Discussion how these types of scalings may have influenced your results."

AR: We did so now. In response to your earlier comment on this topic, we included more information about the scaling law in the model description (section 2.2.1). We also stated in the discussion that it might have influenced the modeled
terminus elevation-mass balance feedback, although any detrimental effect would be included in the errors obtained in the cross-validation procedure.

**RC:** "L44. I guess adding additional parameters not only makes model optimization more complex, but also increase model uncertainty?"

**AR:** Adding parameterizations of previously neglected physical processes, in combination with calibration/validation, should decrease the model uncertainty if enough observations are available. Given the scarcity of such observations for glaciers, it seems questionable if introducing additional degrees of freedom (by introducing new parameters) is beneficial. This question needs to (and eventually, should) be addressed in a dedicated, well-designed study applying identical validation methods to models of different complexity.

460

**RC:** "L48-49. Not clear at this stage how the cited study using OGGM links to your work. Do you follow their approach? On L37-38 you state that you don't include ice dynamic processes, so the reader is left somewhat confused here."

**AR:** We added that there is this global model (OGGM) that includes ice dynamics, but that it cannot reconstruct on time scales as in this study yet, although work on this topic has been done (Eis et al. 2019/2021) That is why we used the simpler

**RC:** "L58-59. "heterogeneously distributed" – I assume you mean within and across geographical regions here? Also I think you can split this sentence into two separate sentences."

AR: Added "heterogeneously distributed in space and time". Also split the sentence.

470

**RC:** "*L*67. *I think "metric" would be more appropriate than "parameter" here.*" **AR:** Corrected.

**RC:** "L55-79. The words evaluation, calibration, validation, and optimization are used somewhat interchangeably in this section. Please check that you're consistent, and clarify what you mean by these terms."

**AR:** We tried to make it more consistent and added a little paragraph of our notions of these terms towards the end of the Introduction.

**RC:** "L94. "as before" -> as in Marzeion et al. (2012)"

480 **AR:** Done.

RC: "L97. Please clarify what you mean by "leave-one-glacier-out" in this context."

**AR:** We added the following sentences on that validation method in section 2.2.1: "Leave-one-glacier-out cross-validation means we run the model once for each validation glacier, i.e. glaciers with at least three recorded annual specific mass-balances, treating the respective glacier as if we did not have in-situ mass-balance measurements available. In other words,  $\beta(t^*)$  is spatially interpolated in an inverse-distance weighted manner from the ten closest glaciers for the computation of annual specific mass-balances of that glacier. The modeled mass-balances of the individual validation glaciers thereby obtained are then validated against the in-situ measurements in the optimization procedure (see section 2.3). Hence, we obtain an estimate of the model's uncertainty attached to the calibration procedure (see section 2.2.2)."

490

**RC:** "L99-100. Beta\* is estimated by spatially interpolating from the ten closest glaciers with 3+ years of in-situ observations. This is fine, but because in-situ observations are very unevenly distributed (which is one of your main points in the paper), the constraints on beta\* will vary greatly across different regions. In regions with plenty of observed glaciers, the interpolation will likely work better than in sparsely observed regions. I think you should explain this explicitly to the reader, and how this influences your results."

495

**AR:** We now added that this "[...] will certainly work better in regions with high measurement densities and thus be a major part of our estimates' inaccuracy due to the previously mentioned heterogeneous distribution of in-situ mass-balance measurements". Unfortunately, we are not able to explicitly state how this will influence our global results, since we cannot

**RC:** "L104-114. If I understand your model correctly, you assume that the fraction of solid precipitation (snow) varies linearly with glacier elevation. If this is the case I think this could be explained in words, in addition to stating the equations. Also you should state why this assumption is sensible for your purposes, and whether this assumption may break down in some glaciological or climatic settings."

505

**AR:** Correct, we added the following on this: "The amount of solid precipitation a glacier receives is hence estimated by applying an empirical negative temperature change, and a parameterized positive precipitation change with increasing elevation. The assumption of increasing precipitation with height might not hold for some glaciers that are located on the downwind side of a mountain or for ones with very high maximum elevations, but this should be accounted for by treating it as a global parameter subject to optimization."

510

**RC:** "L121-125. The scaling procedure and the underlying assumptions should be explained in more detail, see Major comments above. The iterative approach finding the initial area (and volume?) of each glacier needs a bit more elaboration too. For how many glaciers did the iterative procedure work, and in what regions mostly (if any patterns can be seen)?"

**AR:** We added a more detailed description of the scaling laws and the initialization method.

- 515 On the scalings: "The geometric scalings by means of a power law, reviewed in Bahr et al. (2015), are currently the only option for estimating geometric changes from mass changes without having to resolve actual physical processes as in a flowline or higher order model. From the theory discussed in Bahr et al. (2015) it follows that the exponent in these power law scalings is a constant and the scale factor is a random variable. In the model we applied the constant exponents for mountain/valley glaciers and ice caps given from that theory and scale factors empirically derived in Bahr (1997) and Bahr et al. (1997). Since there are
- 520 uncertainties attached to the scale factor, we estimate a 40 % error in the volume-area and a 100 % error in the volume-length scaling for the model's error propagation. Theoretically, the scale factors could be treated as global parameters as well, but it is not clear whether an optimization of those would benefit the overall (global) model accuracy, while it would increase the efforts in computation and evaluation. Concerning the relaxation, a response time scale of the volume-length change is estimated by assuming that smaller glaciers and those with higher mass turnover will react faster to volume changes (details in Marzeion et 2012)."

525 al. 2012)."

On the initialization issue: "The optimized CRU TS 4.03 model setup was able to initialize glaciers accounting for 98 % of the glacier area recorded in the RGI. This value is roughly the same for the optimized model setups that performed well according to our validation procedure, although it is slightly lower for those forced with the mean/median of the meteorological data ensemble (see Table 2). A failure of the initialization for an individual glacier might occur when, for example, the calibration

530 (see section 2.2.2) results in a very high temperature sensitivity for that glacier and the iterative search of an initial area is not able to capture the very large starting area necessary for the thereby assumed strong mass change. The largest fractions of area not successfully modeled with the optimized CRU TS 4.03 model setup are located in the Greenland periphery (ca. 9 %) and Russian Arctic (ca. 5 %)."

500

RC: "L138. How many glaciers have 3+ years of in-situ measurements, and are thus included in this calculation?"

535 **AR:** This is now stated in the Data section (2.1). The answer is 299.

**RC:** "L147. exactly what is "not generally the case" here?"

**AR:** Finding model setups that are not positively biased against in-situ data. We now state this in the previous sentence of the manuscript.

**540**

**RC:** *"L151. Change to (a) instead of "upper panel" – and do so throughout the manuscript"* **AR:** Done.

**RC:** "L157. mid -> middle"

545 **AR:** Done.

**RC:** *"L158. "It" – are you referring to Fig 1b ?"* **AR:** Yes. Replaced "it" with "Fig 1b".

550 RC: "L164. mid -> middle" AR: Done.

**RC:** *"L169-170. "no longer" I assume you mean as opposed to what was done by Marzeion et al. (2012) ?"* **AR:** Yes. We wrote that clearer now.

555

**RC:** "*L183. they were -> full temporal coverage were*" **AR:** We changed "they were obtained [...]" to "these anomalies were obtained [...]".

**RC:** "L187. of -> in"

560 **AR:** Done.

**RC:** *"L188. the modeled time interval – I think you can state here what this exact time interval is."* **AR:** Done.

565 **RC:** "L193. Please state why you ignore the uncertainties in the in-situ observations (for simplicity, because of computational constraints, because they don't matter for your purposes?)"

AR: Mainly for simplicity, and because they are not always known. We added this in the manuscript.

**RC: "L192-194. What about geodetic mass balance estimates? See Major comments."**

**AR:** We added to the Data section (2.1.2) why we cannot use them yet in a desirable way. Also see answers related to this point above.

**RC:** *"L196. a -> an"* **AR:** Done.

575

**RC: "L197-203. How were these ranges selected? Based on previous literature, based on first principles?"**

**AR:** Ranges are similar to those in Marzeion et. al. Stated this in the manuscript now as well. Only the range for the threshold temperature of solid precipitation was shifted lower by one degree.

580 **RC:** "L225. I would try to avoid starting a sentence with "E.g." – replace with For example" or similar. Otherwise a very useful and nicely written paragraph!"

AR: Replaced "E.g." with "For example".

RC: "L230. a respective -> each (check use of "respective" throughout manuscript)"

585 **AR:** Done.

**RC:** *"L264. Avoid starting sentence with "E.g.""* **AR:** Replaced "E.g." with "For example".

590 RC: "L266. be -> by the" AR: Done.

**RC:** "L266. than than (double word)"

**AR:** Deleted double word.

**595**

RC: "L281. different -> differs" AR: Done.

RC: "L284-286. I think you can split this sentence in two for readability."

600 **AR:** Done.

**RC:** "L306. Rewrite to "In Fig. 3a, ... " – and check figure cross-refs (use of B vs (b) etc.) and be consistent throughout the manuscript (for example, at L309 and 313 in this section)"

AR: Done.

**605**

RC: "L316-319. Great point, well said!" AR: Thank you! :-)

RC: "L333. "and for and" – redundant word"

610 **AR:** Deleted redundant word.

**RC:** "A little weird to say that something "grows backward in time" – would rather say that it starts high, then decreases. Similarly on L348, when something "shrinks going back in time". I would rather say that the further back in time we go, the fewer meteorological datasets (ensemble members) are available."

615 **AR:** Adjusted accordingly.

**2.2.3 Global Glacier Mass Loss**

**RC:** "Overall a very interesting section!"

**620**

**RC:** "L350. Not sure why all words in the section heading here are capitalized, why for other headings, they aren't. Check to be consistent."

**AR:** Changed it for this heading to be consistent.

625 **RC:** "*L*352. Use a/b instead of upper/lower panel when you refer to figures." **AR:** Done.

RC: "L354. "are diverging" -> diverge (more direct)" AR: Done.

**630**

**RC:** "L355-358. Check the sign of your mass loss estimates (and in Fig 5) – one can get the impression that mass loss decrease over time with the current figure 5 version, which obviously not is what you want to show."

**AR:** Changed the signs for consistency.

635 **RC:** "*L358. please state explicitly what "the most recent period" is*" **AR:** Done. **RC:** "L359. "earlier years" – a bit vague, please be more concrete"

AR: Done.

640

**RC:** "*L366. state when "the end of the modeled period" is (2018?)*" **AR:** Yes. Done.

RC: "L369-374. Interesting findings regarding the explanation for the found changes in mass loss. These changes regarding
the temperature and precipitation and the resulting changes in mass loss over time are interesting enough to be included in the Abstract, or at least in the Conclusions."

**AR:** We added this sentence to the end of our Conclusions: "Our results indicate that this is partly driven by decreasing amounts of solid precipitation at glacier locations, i.e. reduced accumulation due to atmospheric warming, from ca. 1980 on."

650 **RC:** "*L*376. state which periods you refer to by "the more recent periods""

**AR:** Stated one example in the manuscript.

**2.2.4 Discussion**

RC: "L383. Gardner et al citation should be without parentheses."

655 **AR:** Done.

**RC:** "L385-386. I think you can remove the parentheses around "because ... glaciers"" **AR:** Done.

660 **RC:** "*L402. Again please remind the reader when "the most recent time" is*" **AR:** Done.

RC: "L405. Please state when "earlier periods" is in this context"

**AR:** Added one example.

**665**

**RC:** *"L410. Remind the reader that you're looking at Greenland periphery rather than Greenland"* **AR:** Added "periphery" everywhere in the manuscript for consistency.

RC: "L414. Use of e.g. here breaks the flow of reading, would rather use wording like "for example""
670 AR: Done.

**RC:** "L415. Would add '(e.g. Greenland periphery)' here as an example of such a largely glaciated but less observed region" **AR:** Done.

675 RC: "In general a nicely written section, but missing a few key parts, see Major comments above."AR: Thank you! We addressed these now in the revised version of the manuscript.

**2.2.5 Conclusion**

**RC:** *"L421. Not clear from the context here what "validation data" refers to, please be explicit on what type of data this is"* **AR:** Specified "validation data" as "in-situ mass-balance validation data".

**680**

**RC:** "*L428. Be explicit on what has been decelerating and accelerating (the mass loss, not the glacier flow)*" **AR:** Done.

**2.2.6 Figures**

685 **RC:** "In general nicely crafted, clear figures! A great addition would be a flowchart-type figure explaining the model setup and experimental design, see Major comments."

AR: Thank you! We added a flowchart figure to make our methodology clearer.

RC: "Figure 1: I would write out what beta is in both the ylabel and the caption. Add (a), (b) labels, and for subsequent
figures. Also I would write out what ttilde and t\* is in the caption, so that the reader can understand the figure without looking up the variables in the text. Add cross-reference to the appropriate section after "cross-validation procedure"."

**AR:** Changed figures accordingly.

**RC:** "Figure 2: Would write out the entire ylabel "number of observations" (can be split over two lines to fit the yaxis. 695 Caption: state when "the modeled period" is."

AR: Done.

**RC:** "Figure 3: I think you can remove "RGI" in the title for (a), as it is not crucial to understand the figure and is just another abbreviation to check for the non-expert reader."

700 **AR:** Done.

**RC:** "Figure 5: Check sign of mass loss, see Minor comments above. I think you can also write out "Sea-level contribution" in the ylabel."

AR: Changed the sign. Added "GMSLR contribution".

705

RC: "Figure 6*I would include the legend from Fig 5b also in Fig 6b.*"AR: Done.

**710 **RC:** *"Figure 7**

You have space to write out temperature anomaly and precipitation anomaly in (a). In (b), you don't really need a legend, as there is only one time-series to show."

**AR:** Adjusted accordingly.

**715 2.2.7 Tables**

RC: "Table 1 Resolution – missing unit (degrees?)" AR: Added unit.

**720 3 Answers to reviewer 3**

**3.1 Major points**

**RC:** "IF the paper is going to focus more on the result rather than on the method, one needs to address the fact that this paper is based on reconstructions of the climate based on data sets which are partly observational driven whereas projections are based on model results and the question is whether they can be connected. For this reason it would be interesting to use the

725 *CMIP ensemble over the historical period as an additional data set to see what that brings applied to the proposed method here.*"

**AR:** We think the focus of the paper should stay on the methodology, as it became evident from the reviews that this the part that caused the most confusion (probably in part because of our puzzling explanation) and because it contains approaches not previously applied, e.g. the usage of  $t^*$  and the multi-objective optimization with a scoring system.

730 Concerning climate-model forced runs, the main problem is that we cannot apply our optimization approach to such in a meaningful way, since climate models are not built to reproduce internal, unforced variability, and will therefore not be able to attain good values in the error analysis, even if they performed perfectly at long time scales. Although we could just run the climate model with a certain parameter set, e.g. that of the best-scored data set, such a comparison is out of the scope of this paper, because the paper is about uncertainties in "realistic", observation-driven reconstructions. The comparison of reconstructions 735 done with climate models' historical experiments and reanalysis data, respectively, should therefore rather be done in the context of projections.

**RC:** "I would be interested to see which fraction of the volume loss is related to temperature and which to precipitation. You could easily calculate that with your calibrated model by switching off the variation of one of the two."

- 740 AR: We included a figure in the appendix (Fig. A1), based on two additional model runs we conducted: once holding temperature constant at the climatology around t\* and once holding total precipitation constant at the climatology around t\*. We also added some sentences on this in the Discussion (section 4). In summary, it shows that precipitation anomalies play a minor role on the global scale, which is consistent with the literature.
- **RC:** "I cannot understand the assumption on line 90 where you argue that  $mu_{star}$  and  $beta_{star}$  follow from the assumption that the present-day geometry is a steady state. We all know that the majority of the glaciers is not in a steady state? So even if you find that  $t_{start}$  is not present-day this seems a flawed approach. In addition I can buy the argument that  $t_{star}$  is not only a function of regional climatological history, but then you conclude based on that argument that it is allowed to take a global value for  $t_{star}$ ?. I would conclude based on the same observation that  $t_{star}$  is not a global value, but requires to be estimated for each individual glacier and in fact you seem to do that around line 145. So I have serious problems with the
- concept outlined in lines 90-95. Following up you argue that  $mu_{star}$  for each glacier follows from a global  $t_{star}$  (line 98) why?? and that  $b_{star}$  maybe spatially interpolated why?? Why not the other way around? or both spatially interpolated or based on  $t_{star}$ . I feel lost on the model description."

AR: It is true that the majority of glaciers is not in steady-state, and our way of calibrating μ\* and β\*r does not imply the
contrary. Instead, it assumes that it is possible to find climate conditions in the past (the period which we call t\*), under which glaciers *in present-day geometry* would have been in balance *in the global average*. This assumption is not very strong, simply because the glaciers are there - i.e., climate conditions must have been right for them. Note also that this does not even imply that glaciers were in fact in balance during t\*, because they would have had a different geometry. This is also illustrated by the results of the new experiment, shown in the new figure in the appendix: because glaciers during t\* were generally larger
than today, keeping temperatures constant (climatology taken from t\*) nevertheless leads to shrinking glaciers. We agree that ideally, t\* should be a regional variable, because it is a function of the actual evolution of climate, which is impacted by regional variability (among other things). However, the cross-validation shows that it is not beneficial to spatially interpolate t\*. Probably, this would be different if the density of (in-situ) mass balance observations was greater. However, once you have an estimate of t\* (global in our case), an estimate of μ\* follows from Eq. 1 (Eq. 2 in the revised version of the manuscript),

765

The main point of this – admittedly convoluted – calibration routine is that spatially interpolating  $\mu^*$  leads to significantly larger errors. There is no doubt that there may be better ways to do this, but this is the best we have been able to come up with, and its shortcomings are included in the error estimates obtained in the cross validation.

We re-organized the section describing the model setup and hope we have made these things clearer.

and it is glacier-specific, since temperature and precipitation are glacier-specific.

**RC:** "You use different data sets for temperature, but for precipitation you only use the anomalies of the different data sets based on the holy CRU CL 2.0 data set. Thereby you ignore all the uncertainty in the CRU data and consider it as the holy truth that doesn't seem to be correct to me. Furthermore you don't stress this at all."

**AR:** This seems to be a misunderstanding. We use different data sets for both temperature and precipitation. Taking anomalies implies that we trust the climatology of CRU CL 2.0 data set, and we justify this by its higher spatial resolution. This also does not imply that we ignore uncertainties of the CRU CL 2.0 data set, since of course, they are part of the cause for the uncertainties we quantify in the cross-validation.

**RC:** "Line 380. Table 4 is biased towards a comparison over the recent period, there is a variety of estimates for the entire century it would be interesting to compare you results also with those."

**AR:** We are not aware of global, century-long reconstructions apart from Leclercq et al. (2011, and updates) and Marzeion et al. (2015, and previous versions). We would be glad to include other reconstructions if you can give us a hint.

**3.1.1 Minor points**

RC: "Line 12: A little unclear what is meant with the different ensemble members. I presume you imply to say that most
variation is caused by the input climatological rather than by the uncertainty in the parameters to constrain each model.
Please rephrase to clarify."

**AR:** Done. We mean that, for most years and regions, the uncertainty is rather caused by differences between the individual optimized model setups (i.e. ensemble members) than by differences between model results and annual specific mass-balance observations used for validation.

**790**

775

**RC:** "Line 16: I don't understand the difference between total uncertainty and reconstruction uncertainty. Is ensemble uncertainty + reconstruction uncertainty the total uncertainty? Please rephrase to clarify. I am also unsure what the difference is between ensemble spread and ensemble uncertainty, both are used in the abstract or are they the same?"

AR: As we do a reconstruction, the total uncertainty is the same as the reconstruction uncertainty, which is model un-certainty combined with ensemble spread. To avoid this confusion, we no longer use the term "ensemble uncertainty" in the revised version of the manuscript.

**RC:** "Line 19: The total uncertainty yields an uncertainty of only 8%, that is extremely small in my view and I doubt whether that is in line with budget closure studies of sea level over the last century."

800

**AR:** In the revised version of the manuscript we incorporated ensemble covariances between aggregated years for mean estimates over time periods, which increases the uncertainty estimates. As indicated in the Discussion (section 5) the glacier-related uncertainty of the sea-level budget increases considerably when uncharted glaciers are taken into account (Parkes & Marzeion, 2018). In our view, the sea-level budget closure does not conclusively indicate that these uncertainties are uncrealistic.

805

**RC:** "Line 22: The paper of Slangen et al 2017 is on attribution that does not seem to be a very logical paper to refer to. Moreover the idea that glacier are important is much older. So an older reference seems more appropriate."

**AR:** The Slangen et al. (2017) paper here is not on attribution; it is on model-based reconstruction of 20th century sea-level rise (see reference). Potentially, this is a confusion with a Slangen et al. (2016) paper, which is on attribution.

810

RC: "Line 28: densities can be left out." AR: Done.

RC: "Line 31: It is not trivial that an ensemble based data reconstruction adds to the uncertainty in modelling the future. I
815 think that is only true for an ensemble based on models for the historical period but that is not what you do. You have to make this separation clearer. Though I would prefer you take such an ensemble CMIP5 or 6 onboard in your reconstruction."

**AR:** It is expected that differences in forcings will results in different results, i.e. create uncertainty. To which degree this is true for different climate-model forcings is out of the scope of this study (note our explanation above, why it is not possible to address this in a coherent way in the present experimental set-up) and should be investigated in the context of a projection study.

820

835

**RC:** "Line 36: The word additionally suggest you take changes in the geometry and hypsometry on board, but do you really do that? And how? Having a response time scale is something different. Explicitly state that the time scale mimics only part of the effect caused by the dynamics, the height and length changes are not captured by a time scale, but normally embedded in the dynamical adjustment."

- **AR:** We added some sentences explaining the scalings in the model description (section 2.2.1). The model translates volume changes to area, and length, thereby terminus elevation changes by the mentioned scalings, which are based on power laws derived from that theory. "Additionally" here means that the geometry changes estimated by these means are divided by the estimated response time scale, yielding a relaxation of the area and length changes.
- **RC:** "Line 40: strange sentence whether you resolve the energy balance or the dynamics are different entities. So the fact that an energy balance does not include the dynamics is not an argument to dismiss an energy balance approach."

**AR:** The formulation was aimed at indicating that implementing ice dynamics and energy balance simultaneously is a difficult task and that this is therefore not done routinely. We did not dismiss an energy balance approach, but only pointed out that resolving the energy balance on the global scale is not standard, and where it is applied, other important processes, i.e. changes in geometry/ice dynamics, might not be addressed. We added a statement to the corresponding paragraph in the manuscript.

**RC:** "Line 44: In the context of the paper it is very odd to argue that the additional degrees of freedom of an energy balance models are a limitation. You are aim is to get to an adequate estimate of the uncertainty, so the uncertainty space can much better be explored in an energy balance model than with a temperature-index approach."

AR: We did not state that this is a limitation, but simply that it would add parameters, thus increase the degrees of freedom, which would make the optimization harder. It is not clear to us why the uncertainty space can "much better be explored" with an energy balance model.

**RC:** "Line 46: Computational limitations are not a constraint. The constraint is the lack of data on the geometry and they probably add little due to the large uncertainty in the forcing."

**AR:** We would argue both lack of data and lack of computing resources are limiting our ability to apply high-order models on the global scale.

**RC:** "Line 51-55. I would expect here a summary of the outcome of Marzeion 2020 specifying how large model uncertainty as of is versus forcing uncertainty as discussed in Marzeion 2020."

AR: We added a brief discussion on how Marzeion et al. 2020 is relevant in this context.

**RC:** "Line 75. You can not assume the reader to know the Marzeion et al. 2012 paper, please explain in a few sentences the concept or explicitly refer to the next section. Then wrap up in line 84 a bit more the concept of the Marzeion 2012 model."

**AR:** We added more detail to the descriptions of the model in section 2.2.1 (on the geometric scalings and relaxation) and 2.2.2 (on the calibration procedure).

**RC:** "Line 96. Unclear what you mean with leave-one-glacier-out cross validation. Explain in more detail."

**AR:** We expanded this section and re-organized the way the model, its calibration and uncertainty assessment are introduced.

860

**RC:** "Line 110. Do you include all the uncertainties in parameters around precipitation. What does "with  $T_i_zmax(t)$ " mean. I think you can leave that out the fact that you use (t) already indicates that this parameter is time-dependent."

**AR:** The model uses three parameters (all of them global) concerning accumulation on a glacier: precipitation correction factor (a), temperature threshold for solid precipitation ( $T_{prec \ solid}$ ), and a lapse rate of precipitation ( $\gamma_{precip}$ ). All of them are varied in the optimization procedure and uncertainty assessment. We identify a parameter set for each applied meteorological data set that is best able to reproduce measured in-situ specific mass-balances. This means we take into account that the accumulation of glaciers is uncertain, making the parameters involved subject to optimization. " with T\_i\_zmax(t) " means that  $T_i^{zmax}(t)$  is the temperature at the maximum elevation of the glacier. We left out the (t) on the left side of the equation now.

870 **RC:** "Line 110: Unclear what to do with line 4 of the equation adding it or multiplying it with line 3? Syntax ambiguous."

**AR: There was an unfortunally placed line break, which we removed, making the internal structure of the equation clearer.**

RC: "Line 125: Why can you not find an initial volume which leads to the RGI volume at the end of your run. It is a pretty linear system, so I would expect that there is always an initial volume leading to the RGI volume. Please explain why this is apparently not the case."

**AR:** The volume of the glaciers is not well known; the area is a much better constrain (i.e., lower uncertainty). Therefore, we use the RGI area. Additionally, we added one sentence in the manuscript on why the initialization might fail: "A failure of the initialization for an individual glacier might occur when, for example, the calibration (see section 2.2.2) results in a very high temperature sensitivity for that glacier, and the iterative search of an initial area is not able to capture the very large starting area necessary for the thereby assumed strong mass change."

880

**RC:** "Line 135. Can you explain why there should be a condition of  $B(t_hat)=0$ . Many glaciers are not in steady state for the given geometry so why would this condition hold for at least one value of  $t_hat$ ."

**AR:** We do not assume a (near) steady state for every glacier with the climate around  $\tilde{t}$  (as explained above), but rather of the global glacier mass. This means that while some glaciers will gain mass with this climate, others will lose mass and those should roughly cancel each other out. Please note that we refer to glaciers with present-day geometry experiencing the climate around  $\tilde{t}$  – a counter-factual, intermediate step needed for the calibration routine, and not affecting the actual reconstruction.

RC: "Line 155. Figure 1 is not clear I don't understand the difference between the direct and spatially interpolated method." AR: The direct method is to infer it directly from Equations 4 and 5 (Equations 5 and 6 in the revised version of the manuscript) for each glacier, taking t\* as the  $\tilde{t}$  with the lowest  $|\beta(\tilde{t})|$ . Spatially interpolation here refers to the method used in Marzeion et al. 2012, where t\* was interpolated from the ten closest glaciers in an inverse-distance weighted manner. We put it in the caption of Figure 1 (Figure 2 in the revised version) as follows now: "(b): Distributions of t\* directly estimated from Eq. 5 and Eq. 6 (green), and spatially interpolated as in Marzeion et al. (2012, light green)."

**895**

**RC: "Line 190. Mention for which time interval the RGI data are valid."**

**AR:** We put it as follows in the manuscript: "The majority of recorded glacier areas date back to the years between 2000 and 2010, while there are a few early records between 1970 and 1980. The exact distribution is given in Fig. 2 of RGI (2017)."

900 RC: "Line 220. MSE is not minimized if sigma\_M=sigma\_O but in case R=0 then they are independent."
 AR: A model with R=0 is not to be expected (nor would it be included here), so we dismissed that case.

**RC:** *"Line 222. I don't buy the argument that this is a complex model. Please provides stronger arguments."* **AR:** We deleted "complex".

RC: "Line 251 margin is not the right English word."

AR: Replaced "margin" with "range".

RC: "Line 266 twice than"

910 **AR:** Deleted.

RC: "Line 314 part D of Figure 3"AR: Corrected.

915 RC: "Line 333 and for and?" AR: Corrected.

RC: "Line 340. For completeness you have to mention that you assume e\_ensemble to be independent of sigma\_t such that you can pool them. This is not necessarily true and maybe one of the reasons why you end up with a total uncertainty of only
920 8%. You need to discuss the later in this context."

**AR:** Done. We added the following to that paragraph: "Here, we treat the individual model setups' errors, obtained from the cross-validation procedure, to be independent from each other and the model error of the ensemble average to be independent from the ensemble spread. This might lead to an underestimation of total uncertainty, as there might be correlations of the individual sources of uncertainty for which we cannot account. Because we model 200,000 glaciers and assume their errors to be independent as well, i.e. assume that their errors might, partly, cancel each other out, the 'true' uncertainty is probably higher than our estimate."

**RC:** "Line 358 specify the most recent period 2015-2018?"

**AR:** Specified.

930

925

**RC:** "*Line 370 and 371 consider leaving out "is"*" **AR:** Done.

RC: "Line 394 What is the explanation for the large difference with Zemp et al. 2019."

935 **AR:** We took out the 'CL2 (strongly connected)' glaciers from the Greenland periphery now, which decreased the difference.

RC: "Line 399. You can easily be more specific as you have the Parkes and Marzeion numbers."

**AR:** We added:"[The increase of global glacier mass loss estimates this causes declines throughout the 20th century (Parkes and Marzeion, 2018)] by roughly 66 %."

**RC:** "Line 410. I guess you need to take the marine terminating on – board in a final estimate in order to prevent the suggestion that glacier mass loss is extremely accurately known as you suggest with the 0.05 mm/yr value."

AR: We have them "on - board", but we do not calibrate them well, because effectively, the solid mass loss is lumped into the climatic mass balance in our model's approach. If and how this affects the accuracy of our model we do not know, since
945 there is no global estimate of solid glacier mass loss, let alone for the entire 20th century. Note, however, that the other estimates we compare to in Table 4 mostly include geodetic and gravimetric data, which inlcude mass loss through solid discharge. Nevertheless, our reconstruction typically agrees within the repsective error margins with these other estimates, such that it seems unlikely that the effect is very big.

**950 **RC:** "Line 426. It is worth bringing this to the abstract."**

**AR:** We brought this sentence to the Abstract: "While our estimates lie within the uncertainty range to most of the previously published global estimates, they agree less with those derived from GRACE data."

---

## Author Comment (AC2) · 5 Mar 2021

Please find our comments to all three referees in the supplement PDF.

Please also note the supplement to this comment:
https://tc.copernicus.org/preprints/tc-2020-320/tc-2020-320-AC2-supplement.pdf

---

## Referee Report (RR1)

**Review of tc-2020-320, revised manuscript version**

I have read the new revised manuscript and congratulate the authors for their thorough revisions, which has resulted in a clear and compelling manuscript. I believe that this study will be a valuable contribution to the community and well-received by readers of TC. I have a few minor and technical comments, which I think should be addressed before the paper can be published. These are not too substantial in nature, and I will therefore recommend that the manuscript is accepted subject to minor revisions.

l8. "Therefor," – typo

l29-31. Great addition for background. I would put the newly introduced terms "*surface mass balance*" and "*specific mass balance*" in italics

l35. "satellites able to observe Earth's surface did not yet exist for a large portion of the 20th century." Good point, but I think you can be more specific here what you mean by "large portion" (i.e. letting the reader know roughly for how long back in time we have remotely-sensed observations of glaciers)

l180. Do you mean "inaccuracy" or "uncertainty" here?

l199. Would rather use "elevation" than "height" here to be consistent

l200-207: do you mean "scaling factor" not "scale factor" ?

l212: Not sure what you mean by "random variable" here. Does this mean that it has to be empirically derived?

l214-215. This is important information, how do you estimate the 40% and 100% errors in the volume-area and volume-length scalings, respectively? Does these numbers matter at all for the errors derived in the global model?

l307. "parameter combinations/sets" – I think it's enough to write either "parameter combinations" or "parameter sets", you don't need both

l497: Holding the temperature constant resulted in a mass change decrease of 65 %, while the constant lower precipitation increased the glacier mass change by 5 %.

I would change to "Holding the temperature constant resulted in 65 % lower mass loss, while the constant lower precipitation increases mass loss by 5 %." I think this is what you mean? (looking at your new Fig A1)

l508: "... applied scaling and relaxation laws," - perhaps include cross-references to the relevant equations/section here.

l509-510. " is the positive ice-elevation feedback: as a glacier loses mass, it's thickness, and thereby surface elevation decreases, causing it to experience higher temperatures." I think you can write this in a more concise way. I would also call it a "mass balance – elevation feedback" or "surface mass balance – elevation" feedback. Although this feedback is well known, it doesn't hurt to add a reference (e.g. Harrison et al., 2001).

Reference:
Harrison, W. D., Elsberg, D. H., Echelmeyer, K. A., & Krimmel, R. M. (2001). On the characterization of glacier response by a single time-scale. *Journal of Glaciology*, *47*(159), 659-664.

l536-539. Great point, please add a reference if this point has been raised in the literature before.

l575. "to small" – typo

l610. "making it unpractical to use them in validation framework we applied." - missing "the"

l619. I don't think using "e.g." in-text reads well, would change sentence to "… the application of a robust initialization method (e.g. Eis et al., 2019; 2021) … "

l630. "Finally, all ensemble members agree that around the 1930s mass loss rates from glaciers were comparable to those of today." – this is what you find, but something that can easily be misinterpreted by other scientists only reading the Conclusion, as well as by the broader public/news media. I think you should clarify/add the caveat that you discuss at the end of Section 4 (l507-512), specifically the neglected mass balance-elevation feedback, and state that, most likely, mass loss rates are actually greater today than around the 1930s.

l632-633. "They were followed by a phase of mass loss deceleration roughly between 1940 and 1980, which has been accelerating since then" – this could be written more clearly. First, it is not clear what "They" refer to (the mass loss rates?). In the last part, "which has been", this needs to be clarified as well. What has been accelerating (the deceleration? the mass loss?). From reading the paper I know what you mean, but I'm being a bit picky here because these lines will be among the most read in the paper, so it's great if they cannot be misinterpreted.

l633-635. Perhaps a personal preference, but I would reduce/remove the use of i.e. in-text (similar to e.g. above), and try to describe what you mean in words. In my opinion this makes the text flow better.

l634. "… this is partly driven by … " – do you mean here that reduced solid accumulation partly explains the acceleration found since the 1980s? If so this is an important point. Also Fig 8 shows that the amount of precipitation has increased since the 1980s, so I guess Fig 8 suggests that due to air temperatures being warmer, precipitation increasingly falls as rain

instead of snow. So, the main driver is still air temperature, right? You don't want people to confuse your finding that air temperature is by far the main driver of global glacier mass loss (cf. your new Fig. A1). I think you can end stronger and clearer here.

**Figures**

Figure 1. New flowchart is great, but with the small font size hard to read, I had to zoom to 200% on my screen. Please increase font size/redesign flowchart (extend vertically?) to become more readable, also in print-out format

---

## Referee Report (RR2)

Line 8: Therefor**e**

Motivation for the three criteria line 11 abstract?

Line 13: Cross-validation in abstract difficult as it is not clear what is meant with that term, to me it is not a universal known term.

Line 20/21: Last sentence to be amendment to derived from GRACE data over the period xxxx-yyyy. Because it is another period than the 1901-2018 you have to specify it.

Line 26: remove J.

Line 44: establish rather than establishing?

Line 50 often? Or many times?

Line 50: The sentence The …. Eis, 2021). Disturbs the flow of the paragraph. I suggest to leave it out.

Line 51:Unclear where this refers to. It can be either the current manuscript or the work by Eis et al.

Line 55: these models usually lack ice dynamics or geometric scaling rather then "the models .. scaling"

Line 59: the reasoning around energy balance models are hard to optimize is nonsense. You have already 5 parameters here. You could also develop an energy balance model with 5 parameters if you wish. I suggest you simply remove line 55-61 it does not add to the paper and only dismisses energy balance approaches which is not needed and not the purpose of your paper.

Line 70. I don't think it is correct to say that a comprehensive analysis is not yet possible. It has not been done but that does not mean that is impossible to do. It could be done , maybe for a shorter period or a smaller selection of glaciers. So please tune down this too firm statement.

The paragraph starting with line 73 to line 80 should be interchanged with the next.

Line 104: if you explains terms I suggest you also explain what you mean with cross-validation, you use it as if this is trivial and obvious but at this stage in the manuscript I have not yet any clue what you mean with it.

Line 105: as THE optimization..

Figure 1: It is useful but not really embedded in the text try to refer more often to the figure.

Line 205: It looks like the scale factor is simply taken form Bahr with an uncertainty. It seems to be that the scale factor should be derived from the total volume of glaciers as it follows from the Randolph data. If you don't do that it maybe that you over or underestimate the global ice volume from glacier and this then flaws your dV calculation because you can loose more if you have more and vice versa. So I think you have to find the scale factor for a time where you know the area and the global glacier volume (or better the individual glacier volume), being present-day. I therefore would like to see volume and area information I can compare with Randolph information. What is for instance your ice volume for present-day is it comparable to Randolph derived estimates? I believe that is lacking from the manuscript now.

Line 355 how do you know it is correctly, I believe you but I suggest to leave out from the text.

Figure 1 is a useful addition, but it should be more conceptual and better embedded in the text. If you decide to keep it so detailed you need many references to the figure in the text.

Figure 6 could be expanded with literature values for the GMSLR value of glaciers and this study. For the 20$^{th}$ century and a zoom for the satellite period

Table 4 I would prefer the information which is in the footnote to be in a separate column.

---

## Author Response (AR2)

**Point-by-point response to the reviewers**

We thank the two reviewers for the constructive second round of comments on our manuscript. We have done our best to address the concerns raised by the reviewers and here provide a point-by-point reply to each of the reviewers' comments, including an explanation to subsequent changes to the manuscript. ("RC" stands for "reviewer comment", "AR" for "authors response".)

**1   Answers to reviewer 1**

**RC:** *"l8. "Therefor," –typo"*

**AR:** Corrected.

**RC:** *"l29-31. Great addition for background. I would put the newly introduced terms "surface mass balance" and "specific mass balance" in italics"*

**AR:** Done.

**RC:** *"l35. "satellites able to observe Earth's surface did not yet exist for a large portion of the 20th century." Good point, but I think you can be more specific here what you mean by "large portion" (i.e. letting the reader know roughly for how long back in time we have remotely-sensed observations of glaciers)"*

**AR:** The problem is that it depends on what kind of observations you think about. Images (e.g. for outlines) are now being used from spy satellites from the early sixties. We changed the sentence to: "satellites able to observe the Earth's surface only became available well into the second half of the 20th century".

**RC:** *"l180. Do you mean "inaccuracy"or "uncertainty" here?"*

**AR:** We would argue that it is actually inaccuracy, because if the observations are biased, our estimates will be, too

**RC:** *"l199. Would rather use "elevation"than "height"here to be consistent"*

**AR:** Changed to "elevation".

**RC:** *"l200-207: do you mean "scaling factor" not "scale factor" ?"*

**AR:** It is actually called "scale factor" in the literature.

**RC:** *"l212: Not sure what you mean by "random variable" here. Does this mean that it has to be empirically derived?"*

**AR:** Yes. In the subsequent sentence of the manuscript this is also stated as: "[...] and scale factors empirically derived in Bahr (1997) and Bahr et al. (1997)."

    **RC:** *"l214-215. This is important information, how do you estimate the 40% and 100% errors in the volume-area and volume-length scalings, respectively? Does these numbers matter at all for the errors derived in the global model?"*

**AR:** They are very rough estimates based on the scatter of the scaling relation and are taken from Marzeion et al. (2012); we added "[...], as in Marzeion et al. (2012)" to the corresponding sentence. They do matter for the individual model setups' error estimate of, e.g., glacier volume – but they matter only very little for the estimates of volume *change*. Since we show that the error estimates of the individual model setups are substantially too small anyway, especially pre-1950, they matter even less for the total (ensemble) uncertainty.

    **RC:** *"l307. "parameter combinations/sets"–I think it's enough to write either"parameter combinations"or "parameter sets", you don't need both"*

    **AR:** Removed "combinations/".

**RC:** *"l497:Holding the temperature constant resulted in a mass change decrease of 65 %, while the constant lower precipitation increased the glacier mass change by 5 %. I would change to "Holding the temperature constant resulted in 65 % lower mass loss, while the constant lower precipitation increases mass loss by 5 %." I think this is what you mean? (looking at your new Fig A1)"*

    **AR:** Adopted the suggestion.

    **RC:** *"l508: "... applied scaling and relaxation laws," -perhaps include cross-references to the relevant equations/section here."*

    **AR:** Added "(see section 2.2.1)".

**RC:** *"l509-510."is the positive ice-elevation feedback: as a glacier loses mass, it's thickness, and thereby surface elevation decreases, causing it to experience higher temperatures." I think you can write this in a more concise way. I would also call it a "mass balance –elevation feedback"or "surface mass balance –elevation"feedback. Although this feedback is well known, it doesn't hurt to add a reference (e.g. Harrison et al., 2001)."*

    **AR:** We formulated it now as follows: "[...] is the positive mass balance-elevation feedback: as a glacier's surface elevation
decreases due to mass loss, it experiences higher temperatures, because of the atmospheric temperature lapse rate. This in turn enhances the initial mass loss." We already had one reference on this topic in the subsequent sentence, but added the suggested one as well.

    **RC:** *l536-539. Great point, please add a reference if this point has been raised in the literature before."*

**AR:** Although there is some literature on icebergs in fjords, we were not able to find a discussion of the point raised in our manuscript in articles on GRACE-based glacier mass change estimates.

  **RC:** *"l575. "to small"–typo"*

  **AR:** Corrected.

  **RC:** *"l610. "making it unpractical to use them in validation framework we applied."-missing "the""*

  **AR:** Added "the".

  **RC:** *"l619. I don't think using "e.g."in-text reads well, would change sentence to "...the application of a robust initialization*
*method (e.g. Eis et al., 2019;2021)...""*

  **AR:** Adopted the suggestion.

  **RC:** *"l630. "Finally, all ensemble members agree that around the 1930s mass loss rates from glaciers were comparable to those of today."–this is what you find, but something that can easily be misinterpreted by other scientists only reading the*
*Conclusion, as well as by the broader public/news media. I think you should clarify/add the caveat that you discuss at the end of Section 4 (l507-512), specifically the neglected mass balance-elevation feedback, and state that, most likely, mass loss rates are actually greater today than around the 1930s."*

  **AR:** This is a good point, thank you! We added the following now: "This finding is weakened by the lack of an explicit mass balance-elevation feedback in the model we applied, though, and it might hence well be that mass change rates during
the 1930s were actually smaller than today".

  **RC:** *"l632-633. "They were followed by a phase of mass loss deceleration roughly between 1940 and 1980, which has been accelerating since then"–this could be written more clearly. First, it is not clear what "They" refer to (the mass loss rates?). In the last part, "which has been", this needs to be clarified as well. What has been accelerating(the deceleration? the mass*
*loss?). From reading the paper I know what you mean, but I'm being a bit picky here because these lines will be among the most read in the paper, so it's great if they cannot be misinterpreted."*

  **AR:** Further good points, thanks! We now formulated it as follows: "According to our results, the increase in mass loss until the 1930s was followed by a phase of mass loss deceleration until roughly 1980. The glaciers' contribution to sea-level rise has been accelerating again since then [...]".

  **RC:** *"l633-635. Perhaps a personal preference, but I would reduce/remove the use of i.e. in-text (similar to e.g. above), and try to describe what you mean in words. In my opinion this makes the text flow better."*

  **AR:** We let the "[...], i.e. higher altitudes" formulation as it was, hoping that, as it comes at the end of the sentence, it does not disturb the flow of the text in that case. The next sentence, where we also had an "i.e." was reformulated in regard to your next comment.

RC: *"l634. "...this is partly driven by ..."–do you mean here that reduced solid accumulation partly explains the acceleration found since the 1980s? If so this is an important point. Also Fig 8 shows that the amount of precipitation has increased since the 1980s, so I guess Fig 8 suggests that due to air temperatures being warmer, precipitation increasingly falls as rain instead*
*of snow. So, the main driver is still air temperature, right? You don't want people to confuse your finding that air temperature is by far the main driver of global glacier mass loss (cf. your new Fig. A1).I think you can end stronger and clearer here."*

AR: We changed the part you are referring to here to: "Our results also indicate that this acceleration was partly driven by decreasing amounts of solid precipitation at glacier locations from ca. 1980 onward. This implies that the enhanced atmospheric warming not only increased ablation rates, but probably lowered the amount of snow the glaciers received as well,
notwithstanding a slight increase in total precipitation.". This should make the final statement clearer and stronger.

**1.1 Figures**

RC: *"Figure 1. New flowchart is great, but with the small font size hard to read, I had to zoom to 200% on my screen. Please increase font size/redesign flowchart (extend vertically?) to become more readable, also in print-out format"*
AR: We substantially increased the font size.

**2 Answers to reviewer 2**

RC: *"Line 8: Therefore"*
AR: Corrected.

RC: *"Motivation for the three criteria line 11 abstract?"*
AR: The motivation is laid out in section 2.3. Still, we added the following now as the subsequent sentence: "These criteria were chosen in order not to trade lower error estimates by means of the root mean squared error (RMSE) for an unrealistic interannual variability."

RC: *"Line 13: Cross-validation in abstract difficult as it is not clear what is meant with that term, to me it is not a universal known term."*
AR: We think the usage of this term in the abstract is appropriate, since using only "validation" would just make it less specific. To make the kind of used cross-validation method clearer we added to the second sentence where the criticized term appears the following (in italics): "[...] than the uncertainties obtained via the *leave-one-glacier-out* cross-validation, [...]". We

think that methods should usually not be explained in the abstract or introduction, but in the corresponding section of the article.

**RC:** *"Line 20/21: Last sentence to be amendment to derived from GRACE data over the period xxxx-yyyy. Because it is another period than the 1901-2018 you have to specify it."*

**AR:** We added the following to the last sentenced of the Abstract: "[...] , which only cover the years 2002 - 2018 though."

**RC:** *"Line 26: remove J."*

**AR:** Done.

**RC:** *"Line 44: establish rather than establishing?."*

**AR:** We think "establishing" is correct in this context.

**RC:** *"Line 50 often? Or many times?"*

**AR:** Changed to "many times".

**RC:** *"Line 50: The sentence The . . . . Eis, 2021). Disturbs the flow of the paragraph. I suggest to leave it out."*

**AR:** We moved a slightly changed version of this sentence two paragraphs down, where it fits better.

**RC:** *"Line 51: Unclear where this refers to. It can be either the current manuscript or the work by Eis et al.."*

**AR:** We changed "[...] used in this work [...]" to "[...] used in the work presented here [...]".

**RC:** *"Line 55: these models usually lack ice dynamics or geometric scaling rather then "the models .. scaling""*

**AR:** We adopted the suggestion.

**RC:** *"Line 59: the reasoning around energy balance models are hard to optimize is nonsense. You have already 5 parameters here. You could also develop an energy balance model with 5 parameters if you wish. I suggest you simply remove line 55-61 it does not add to the paper and only dismisses energy balance approaches which is not needed and not the purpose of your paper."*

**AR:** We do not dismiss energy balance approaches, but merely state that those add complexity. The number of parameters
usually increases with the number of processes one wants to resolve in a model, if those are not computed from first principles. We added: "[...] not yet done routinely, but might still have the potential to enhance the accuracy of glacier modeling. That is because such models would have the ability to represent the physical mechanisms influencing a glacier in a more detailed, and thus possibly more realistic, fashion." to the end of the paragraph to point out that we do not want to dismiss such an approach, but merely point to the added complexity, which, in turn, might actually enhance such a model's capability to reproduce glacier behavior.

**RC:** *"Line 70. I don't think it is correct to say that a comprehensive analysis is not yet possible. It has not been done but that does not mean that is impossible to do. It could be done , maybe for a shorter period or a smaller selection of glaciers. So please tune down this too firm statement."*

**AR:** We extended the corresponding sentence to (in italics): "This implies that a comprehensive analysis determining which modeling approach might be most appropriate is not yet possible*; at least not for all global glaciers and the whole 20th century*."

**RC:** *"The paragraph starting with line 73 to line 80 should be interchanged with the next."*

**AR:** Done.

**RC:** *"Line 104: if you explains terms I suggest you also explain what you mean with cross-validation, you use it as if this is trivial and obvious but at this stage in the manuscript I have not yet any clue what you mean with it."*

    **AR:** We added (in italics): "[...] *leave-one-glacier-out* cross-validation *(see section 2.2.1 and green box in Fig. 1)* [...]". In
section 2.2.1 the concept is actually (briefly) explained.

**RC:** *"Line 105: as THE optimization.."*
    **AR:** Added "the".

**RC:** *"Figure 1: It is useful but not really embedded in the text try to refer more often to the figure."*
    **AR:** We added some references to the figure to the manuscript.

**RC:** *"Line 205: It looks like the scale factor is simply taken form Bahr with an uncertainty. It seems to be that the scale factor should be derived from the total volume of glaciers as it follows from the Randolph data. If you don't do that it maybe*
*that you over or underestimate the global ice volume from glacier and this then flaws your dV calculation because you can loose more if you have more and vice versa. So I think you have to find the scale factor for a time where you know the area and the global glacier volume (or better the individual glacier volume), being present-day. I therefore would like to see volume and area information I can compare with Randolph information. What is for instance your ice volume for present-day is it comparable to Randolph derived estimates? I believe that is lacking from the manuscript now."*

**AR:** The problem here is that in order to empirically derive the scale factor one has to know the volume of the glaciers the scale factor is fitted to. The RGI does, unfortunately, not contain volume estimates for every glacier; only outlines/area estimates. Thus, we rely on the previously empirically derived scale factors from Bahr (et al.). To section 3.2 we added: "Comparing our results of glacier geometry to a publication that estimated contemporary global glacier volumes (Farinotti et al. 2019), though on the basis of modeling results as well, we find that our global volume estimate differs less than 1 % from their result." Additionally, as stated in the manuscript and Fig. 1, we only directly model glaciers that meet the recorded RGI areas in the individual years of record.

**RC:** *"Line 355 how do you know it is correctly, I believe you but I suggest to leave out from the text."*

**AR:** We calculated it from the validation data as stated in the manuscript (section 2.3). Therefore, we see no good reason to leave it out.

**RC:** *"Figure 1 is a useful addition, but it should be more conceptual and better embedded in the text. If you decide to keep it so detailed you need many references to the figure in the text."*

**AR:** We added some references to the figure to the manuscript.

**RC:** *"Figure 6 could be expanded with literature values for the GMSLR value of glaciers and this study. For the 20th century and a zoom for the satellite period"*

**AR:** We think this would make the figure too crowded and hence unclear. Literature values for comparison are given in Table 4.

**RC:** *"Table 4 I would prefer the information which is in the footnote to be in a separate column."*

**AR:** We put the information that was in the footnote in a separate column now.

---

## Author Response (AR3)

**Response to the editor comments**

We thank the editor for the final constructive comments on our manuscript. We have addressed the points raised by the editor and here provide a brief summary of the changes, including an explanation to subsequent changes to the manuscript. ("EC" stands for "editor comment", "AR" for "authors response".)

**1 Answers to the editor's comments and changes to the final version of the manuscript**

5 **EC:** *"A few of the revisions are not well written. [...]"*

**AR:** We changed the phrasing at places and in the way the editor suggested it, although in one case we are confident that the phrasing was already correct.

**2 Additional minor changes to the manuscript**

10 We corrected some minor typos.

In the last paragraph of section 2.2.2 we changed "[...] *this* then results in an overall positive bias in the global cross-validation result." to "[...], *moving the globally applied $t^*$ to a warmer climate period* then results in an overall positive bias in the global cross-validation result." to make more clear what was meant with "this" in the previous version.

15

In the beginning of section 3.1.1 we adjusted the numbers of modeled glaciers, because in the previous version we did not subtract the glaciers in the Greenland periphery that are marked as 'CL2 (strongly connected)' in the Randolph Glacier Inventory (RGI), which we did exclude from our results in the course of the reviews.

20 In the Conclusions (section 5) we changed the phrasing slightly (in addition to the comments of the editor) to make it clearer.